# Unified Convergence Analysis for Score-Based Diffusion Models with Deterministic Samplers

**Runjia Li[1], Qiwei Di[2], Quanquan Gu[2]**
[1]School of Mathematical Sciences, Peking University, Beijing, China
[2]Department of Computer Science, University of California, Los Angeles, CA 90095, USA
`lirunjia@stu.pku.edu.cn`,`{qiwei2000,` `qgu}@cs.ucla.edu`

## Abstract

Score-based diffusion models have emerged as powerful techniques for generating samples from high-dimensional data distributions. These models involve a two-phase process: first, injecting noise to transform the data distribution into a known prior distribution, and second, sampling to recover the original data distribution from noise. Among the various sampling methods, deterministic samplers stand out for their enhanced efficiency. However, analyzing these deterministic samplers presents unique challenges, as they preclude the use of established techniques such as Girsanov's theorem, which are only applicable to stochastic samplers. Furthermore, existing analysis for deterministic samplers usually focuses on specific examples, lacking a generalized approach for general forward processes and various deterministic samplers. Our paper addresses these limitations by introducing a unified convergence analysis framework. To demonstrate the power of our framework, we analyze the variance-preserving (VP) forward process with the exponential integrator (EI) scheme, achieving iteration complexity of $\widetilde{O}(d^2/\epsilon)$. Additionally, we provide a detailed analysis of Denoising Diffusion Implicit Models (DDIM)-type samplers, which have been underexplored in previous research, achieving polynomial iteration complexity.

## 1 Introduction

Score-based diffusion models (Sohl-Dickstein et al., 2015; Ho et al., 2020; Song et al., 2020a;c) have emerged as a powerful class of generative models, achieving significant success in image generation tasks, such as DALL·E (Ramesh et al., 2022), Stable Diffusion (Rombach et al., 2022), Imagen (Saharia et al., 2022), and SDXL (Podell et al., 2023). Beyond that, these models have also demonstrated effectiveness in diverse applications, including structure-based drug design (Corso et al., 2022; Guan et al., 2024), text generation (Austin et al., 2021; Zheng et al., 2023; Chen et al., 2023c), and reinforcement learning (Wang et al., 2022; Lu et al., 2023). At the core of the score-based diffusion models is a forward Stochastic Differential Equation (SDE) to diffuse the data distribution into a known prior distribution, and a neural network is trained to approximate the score function. Scalable score-matching techniques, such as denoising score matching (Vincent, 2011) and sliced score matching (Song et al., 2020b), enable efficient learning of the score function. Once learned, we can use numerical samplers to simulate the backward process and recover the original data distribution from noise.

To improve the sampling quality and efficiency of diffusion models, it is crucial to use efficient samplers in addition to an accurate score estimator. Early developments of diffusion models, such as Denoising Diffusion Probabilistic Models (DDPM) (Ho et al., 2020) applied stochastic samplers. Later, empirical studies showed that diffusion models with deterministic samplers, such as Denoising Diffusion Implicit Models (DDIM) (Song et al., 2020a) can still generate high-quality samples while achieving better efficiency. For instance, DDIM is more than 10 times faster than DDPM. Beyond DDIM, many novel fast ODE solvers have been developed for diffusion models, further improving the efficiency of the sampling processes (Lu et al., 2022; Zhou et al., 2024).

The remarkable success of diffusion models has inspired extensive research interest in the mathematical analysis of these powerful generative models (Block et al., 2020; De Bortoli et al., 2021; De Bortoli, 2022; Lee et al., 2023; Pidstrigach, 2022; Chen et al., 2022; 2023a;b; Li et al., 2024a;b;c;

Benton et al., 2024; Huang et al., 2024a). Notably, many prior works studied the convergence of diffusion models with stochastic samplers (Chen et al., 2022; 2023a; Benton et al., 2024; Li et al., 2024b). A key tool in their analysis is Girsanov's theorem, which helps control the distance between the distributions of two stochastic processes. However, it relies on the smoothing effect provided by stochasticity and therefore do not apply to deterministic samplers. To address this, (Chen et al., 2024) introduced an additional corrector into the Langevin dynamics, which incorporates randomness to help smooth the distribution. Meanwhile, Chen et al. (2023b) assumed the access to the exact score function and provided a discretization analysis for the probability flow ODE in Kullback–Leibler (KL) divergence. Moreover, under extra conditions, Li et al. (2024b) proved a non-asymptotic convergence rate for a specific deterministic sampler based on the probability flow ODE with elementary analysis. Huang et al. (2024a) considered the Ornstein–Uhlenbeck (OU) forward process with the Runge-Kutta integrator and provided error bounds for both continuous- and discrete-time settings. However, all of these works focus on specific forward processes and samplers. Therefore, a natural question arises:

*Can we develop a unified framework for the convergence analysis of diffusion models with deterministic samplers that accommodates those common forward processes and sampling algorithms?*

In this paper, we provide an affirmative answer to this question. We summarize our contributions as follows:

- We develop a key technical tool (Lemma 3.2), which enables us to bound the time derivative of the total variation (TV) distance between the final states of two ODE processes, through the difference in their drift terms and the divergence. As a direct application, we establish convergence guarantees for the continuous-time reverse ODE in the case of the OU forward process.

- We provide a unified convergence analysis framework for diffusion models with deterministic samplers. For general forward processes and sampling algorithms, our framework decomposes the error of diffusion models into five distinct terms: two arising from score estimation and three from time discretization. This decomposition enables a divide-and-conquer approach, allowing for improved analysis of each error component while maintaining the consistency of the unified framework.

- To demonstrate the generality and effectiveness of our framework, we apply it to two typical diffusion model settings: we achieve $\widetilde{O}(d^2/\epsilon)$ iteration complexity for the Variance Preserving (VP) forward process with exponential integrator (EI) numerical scheme. This theoretical guarantee matches Li et al. (2024b), where a similar but different sampling algorithm is considered. Moreover, we establish polynomial iteration complexity for the Variance Exploding (VE) forward process with the DDIM numerical scheme. To the best of our knowledge, this is the first convergence result for diffusion models employing DDIM-type samplers that can handle estimated scores, whereas prior works, such as Chen et al. (2023b), only considered cases with access to accurate score function.

**Notation:** In this work, we use lowercase letters $a, b$ to represent scalars, lowercase bold letters $\mathbf{x}, \mathbf{y}$ to represent vectors, uppercase italic bold letters $\boldsymbol{X}, \boldsymbol{Y}$ to represent random variables, and uppercase bold letters $\mathbf{A}, \mathbf{B}$ to represent matrices. For a vector $\mathbf{x} \in \mathbb{R}^d$ and matrix $\mathbf{A} \in \mathbb{R}^{d \times d}$, we denote by $\|\mathbf{x}\|$ the Euclidean norm of $\mathbf{x}$ and $\|\mathbf{A}\|_2$ the operator norm of $\mathbf{A}$. We use $f_1 \lesssim f_2$ to denote that there is a universal constant $C$ such that $f_1 \leq C f_2$. For two sequences $\{a_n\}$ and $\{b_n\}$, we write $a_n = O(b_n)$ if there exists an absolute constant $C$ such that $a_n \leq C b_n$, and we use $\widetilde{O}(\cdot)$ to further hide the logarithmic factors. For vector operations, we use $\nabla$ to denote the gradient, $\nabla\cdot$ to denote divergence, and $\nabla^2$ to denote the Jacobian matrix.

## 2 PROBLEM BACKGROUND

The primary objective of diffusion models is to generate new samples given a set of examples drawn from the data distribution. In this section, we introduce the fundamentals of ODE-based diffusion models with deterministic samplers.

### 2.1 ODE-BASED DIFFUSION MODELS

A diffusion model typically consists of a forward process that perturbs the data distribution into noise, followed by a denoising backward process. In general, the forward process can be modeled as an Itô SDE:

$$\mathrm{d}\boldsymbol{X}_t = \mathbf{f}(t, \boldsymbol{X}_t)\mathrm{d}t + g(t)\mathrm{d}\boldsymbol{W}_t, \quad \boldsymbol{X}_0 \sim q_0, \tag{2.1}$$

where $\boldsymbol{W}_t$ is a Brownian motion in $\mathbb{R}^d$, $\mathbf{f}(t, \cdot)$ is called the drift coefficient, $g(t)$ is called the diffusion coefficient (Song et al., 2020c). It begins by sampling $\boldsymbol{X}_0$ from the data distribution $q_0$, and evolves according to the forward process (2.1). The law of $\boldsymbol{X}_t$ is denoted by $q_t(\mathbf{x})$. Under mild regularity conditions on $q_0$, we can construct a family of reverse processes $\left(\boldsymbol{Y}_t^\lambda\right)_{t \in [0, T]}$, which evolve according to the following SDE:

$$\mathrm{d}\boldsymbol{Y}_t^\lambda = -\Big(\mathbf{f}(T - t, \boldsymbol{Y}_t^\lambda) - \frac{1 + \lambda^2}{2} g(T - t)^2 \nabla \log q_{T-t}(\boldsymbol{Y}_t^\lambda)\Big)\mathrm{d}t + \lambda g(T - t)\mathrm{d}\boldsymbol{W}_t\,, \quad \boldsymbol{Y}_0^\lambda \sim q_T. \tag{2.2}$$

These processes hold the same marginal distribution as $q_{T-t}(\mathbf{x})$ at time $t$ (Chen et al., 2023b). As a special case when $\lambda = 0$, the backward process is deterministic, i.e.,

$$\mathrm{d}\boldsymbol{Y}_t = -\Big(\mathbf{f}(T - t, \boldsymbol{Y}_t) - \frac{1}{2} g(T - t)^2 \nabla \log q_{T-t}(\boldsymbol{Y}_t)\Big)\mathrm{d}t\,, \quad \boldsymbol{Y}_0 \sim q_T, \tag{2.3}$$

which is called the probability flow ODE (Song et al., 2020c).

The goal of ODE-based diffusion model is to simulate (2.3). However, we must make several approximations. First, as we do not have access to the true score $\nabla \log q(t, \mathbf{x})$, we learn an approximation $\mathbf{s}_\theta(t, \mathbf{x})$ by minimizing

$$\mathcal{L}(\mathbf{s}_\theta) = \int_0^T \mathbb{E}_{q_t}\left[\|\mathbf{s}_\theta(t, \boldsymbol{X}_t) - \nabla \log q_t(\boldsymbol{X}_t)\|^2\right]\mathrm{d}t. \tag{2.4}$$

When $\mathbf{f}(t, \cdot)$ is affine, the transition is Gaussian, allowing for the closed-form solution of $\nabla \log q(t, \mathbf{x})$. By applying integration by parts, (2.4) can be reformulated into tractable objectives, specifically through the methods of denoising score matching and implicit score matching (Hyvärinen & Dayan, 2005; Vincent, 2011). For more general forward processes, these objectives can be estimated using samples drawn from the forward process. Then, using the estimated score function $\mathbf{s}_\theta$, we have the simulate reverse process:

$$\mathrm{d}\widehat{\boldsymbol{Y}}_t = -\Big(\mathbf{f}(T - t, \widehat{\boldsymbol{Y}}_t) - \frac{1}{2} g(T - t)^2 \mathbf{s}_\theta(T - t, \widehat{\boldsymbol{Y}}_t)\Big)\mathrm{d}t\,, \quad \widehat{\boldsymbol{Y}}_0 \sim q_T. \tag{2.5}$$

Second, since the initial distribution $q_T$ of the reverse process is not directly accessible, we instead initialize the reverse process with a given Gaussian distribution $\pi_d$, which is assumed to be a good approximation of $q_T$.

Third, in practical implementations, the continuous-time process is typically simulated using time discretization. Specifically, we choose time steps $0 = t_0 < t_1 < \cdots < t_N \leq T$, sample $\widehat{\boldsymbol{Y}}_0 \sim \pi_d$ and iteratively calculate $\widehat{\boldsymbol{Y}}_{t_{k+1}}$ from $\widehat{\boldsymbol{Y}}_{t_k}$ using a particular numerical scheme. In this way, we only need to have access to the value of the estimated score function $\mathbf{s}_\theta(t_k, \widehat{\boldsymbol{Y}}_{t_k}), k = 0, 1, \ldots, N$, which can reduce the computational complexity of the sampling process. Common numerical schemes include the Exponential Integrator scheme, the Euler-Maruyama scheme, and the DDIM-type sampler scheme, as we will discuss in Section 4.1.

For non-smooth data distributions, the score function $\nabla \log q_t$ can be unbounded as $t \to 0$, which will happen especially when the data distribution is supported on a lower-dimensional submanifold of $\mathbb{R}^d$. This will lead to difficulty when considering the distance to the data distribution. For this reason, we consider an early stopping scheme, selecting $t_N = T - \delta$ for some small $\delta$. Our analysis focuses on the distance between the final distribution of the sampling process and $q_\delta$. When $\delta$ is sufficiently small, the Wasserstein-p metric between $q_\delta$ and $q_0$ is small. This shows that it is reasonable to approximate $q_\delta$ instead of $q_0$.

In this paper, we propose a unified framework that can handle different time schedules. The choice of the time schedule is crucial for obtaining good results when analyzing the convergence of specific samplers. In this paper, we follow the time schedule used in Benton et al. (2024). This allows us to provide more refined control over the discretization error. We will always assume $T > 1$ and $\delta < 1$. In the first stage when $t \in [0, T - 1]$, we use uniform time steps, with step size $\eta_k = t_{k+1} - t_k \leq \eta$. In the second stage for $t \in [T - 1, T - \delta]$, we assume time steps satisfying $\eta_k \leq \eta(T - t_{k+1})$, which results in an exponential decay at a rate of $(1 + \eta)^{-1}$. Using this schedule, we have

$$\eta_k \leq \eta \min\{1, T - t_{k+1}\}. \tag{2.6}$$

It's worth noting that similar first-uniform-then-exponential schedules have also been utilized in Chen et al. (2023a); Li et al. (2024b;c).

## 2.2 DIFFICULTIES IN CONVERGENCE ANALYSIS OF ODE FLOW

Compared with the convergence analysis of SDEs, the convergence analysis of ODEs is more challenging. For SDE, the prior results either apply Girsanov's Theorem (Chen et al., 2022) or rely on a chain-rule-based argument (Chen et al., 2023a). Using these methods, we can see that for SDEs, when the drift term is slightly perturbed, it is possible to bound the distance between the final iterates or, more strongly, the entire distributions over the trajectories. Therefore, the assumption of the score estimation is sufficient. However, as the following theorem illustrates, this fails in the ODE case.

**Theorem 2.1.** Consider the 1-dimensional OU process $(x_t)_{t \in [0,T]}$ which starts at $N(0,1)$. It satisfies the following SDE

$$\mathrm{d}x_t = -x_t \mathrm{d}t + \sqrt{2}\mathrm{d}W_t, \quad x_0 \sim N(0,1).$$

Let the law of $x_t$ be denoted by $q_t$. Its reverse process $(y_t)_{t \in [0,T]}$ (see (2.3)) can be represented with the following ODE:

$$\mathrm{d}y_t = \Big(y_t + \nabla \log q(T-t, y_t)\Big)\mathrm{d}t, \quad y_0 \sim q_T.$$

For arbitrarily small $\epsilon > 0$, there exists $s_\theta(t,x)$, which is smooth, and anywhere $\epsilon$-close to the true score function, i,e,

$$\big|\mathbf{s}_\theta(T-t, x) - \nabla \log q(T-t, x)\big| \le \epsilon, \quad \forall t, \forall x,$$

such that the corresponding simulated reverse process $(\widehat{y}_t)_{t \in [0,T]}$ (see (2.5)) satisfies

$$\mathrm{TV}(\widehat{y}_T, y_T) \ge \frac{1}{4\pi},$$

which indicates that the TV-distance of the final states is larger than a constant no matter how small the score estimation error is.

**Remark 2.2.** For the construction of the counter-example, we consider a specific scenario where the data distribution is standard Gaussian, and the OU process maintains the distribution. In this case, adding an arbitrarily small perturbation to the drift term of the reverse ODE can result in a sampling probability density function $\widehat{q}(T-t, x) = \frac{1}{\sqrt{2\pi}}\mathrm{e}^{-\frac{x^2}{2}}\big(1 + \frac{t}{2T}\sin(2n\pi x)\big)$ with severe oscillations (n can be arbitrarily large), instead of a Gaussian distribution $q(t,x) = \frac{1}{\sqrt{2\pi}}\mathrm{e}^{-\frac{x^2}{2}}$. For details, please refer to Section B. This demonstrates that the score estimation error assumption is insufficient, even when ignoring the discretization error. For a similar purpose, an example has been given in Li et al. (2024b) for the discrete-time sampling algorithms. Notably, our constructed $\mathbf{s}_\theta$ is smooth while theirs is not. This indicates that adding the score function's smoothness condition alone is insufficient to guarantee the desired properties.

## 3 CONVERGENCE OF CONTINUOUS TIME REVERSE PROCESS

In this section, we focus on the OU forward process, where $f(t, \mathbf{x}) = -\mathbf{x}$ and $g(t) = \sqrt{2}$. Temporarily, we ignore the discretization error and focus on the true reverse process (2.3) and the simulated reverse process (2.5) starting at $\pi_d$. Specifically, we want to study the TV-distance between the following ODEs.

$$\mathrm{d}\mathbf{Y}_t = \Big(\mathbf{Y}_t + \nabla \log q_{T-t}(\mathbf{Y}_t)\Big)\mathrm{d}t, \quad \mathbf{Y}_0 \sim q_T, \tag{3.1}$$

$$\mathrm{d}\widehat{\mathbf{Y}}_t = \Big(\widehat{\mathbf{Y}}_t + \mathbf{s}_\theta(T-t, \widehat{\mathbf{Y}}_t)\Big)\mathrm{d}t, \quad \widehat{\mathbf{Y}}_0 \sim \pi_d. \tag{3.2}$$

We make the following standard assumption regarding the score estimation error.

**Assumption 3.1** (Score estimation error). The estimated score function $\mathbf{s}_\theta(\cdot, \cdot)$ satisfies:

$$\mathbb{E}_{t, \mathbf{Y}_t}\big\|\mathbf{s}_\theta(T-t, \mathbf{Y}_t) - \nabla \log q(T-t, \mathbf{Y}_t)\big\|^2 \le \epsilon_{\mathrm{score}}^2.$$

This assumption guarantees that the drift terms in (3.1) and (3.2) will be close. However, as discussed in Remark 2.2, this condition alone is insufficient for ODE flows. To address this problem, we introduce the following lemma. It states that the time derivative of the TV distance between two ODE flows is determined by the distance between their drift terms and the divergence of these drift terms. Its proof is left to Appendix C.

**Lemma 3.2.** Suppose $\boldsymbol{X}_t$ and $\boldsymbol{Y}_t$ are stochastic processes in $\mathbb{R}^d$ driven by ODEs:

$$
\begin{aligned}
\mathrm{d}\boldsymbol{X}_t &= \mathbf{b}(t, \boldsymbol{X}_t)\,\mathrm{d}t\,, \quad \boldsymbol{X}_0 \sim p_0, \\
\mathrm{d}\boldsymbol{Y}_t &= \mathbf{b}^*(t, \boldsymbol{Y}_t)\,\mathrm{d}t\,, \quad \boldsymbol{Y}_0 \sim q_0.
\end{aligned}
$$

Let $p(t, \mathbf{x})$ be the law of $\boldsymbol{X}_t$ and $q(t, \mathbf{x})$ be the law of $\boldsymbol{Y}_t$. If the drift terms $\mathbf{b}, \mathbf{b}^* : [0, \infty) \times \mathbb{R}^d \to \mathbb{R}^d$ are continuously differentiable with respect to $\mathbf{x} \in \mathbb{R}^d$, then the time-derivative of the total variation distance between $\boldsymbol{X}_t$ and $\boldsymbol{Y}_t$ satisfies the following equation:

$$
\begin{aligned}
\frac{\partial \mathrm{TV}(\boldsymbol{X}_t, \boldsymbol{Y}_t)}{\partial t} &= - \int_{\Omega_t} \big(\nabla \cdot \mathbf{b}(t, \mathbf{x}) - \nabla \cdot \mathbf{b}^*(t, \mathbf{x})\big) q(t, \mathbf{x})\,\mathrm{d}\mathbf{x} \\
&\quad - \int_{\Omega_t} \big(\mathbf{b}(t, \mathbf{x}) - \mathbf{b}^*(t, \mathbf{x})\big) \cdot \nabla \log q(t, \mathbf{x}) q(t, \mathbf{x})\mathrm{d}\mathbf{x},
\end{aligned}
$$

where $\Omega_t := \{\mathbf{x} \in \mathbb{R}^d \mid p(t, \mathbf{x}) > q(t, \mathbf{x})\}$.

In the application of this lemma to diffusion models, we can choose the true reverse ODE and sampling process ODE. Let $p_t$ be the distribution of the sampling process, and $q_t$ be the distribution of the true reverse process. Several works (Chen et al., 2023b; Albergo et al., 2023), considered the time derivative of KL divergence, deriving expressions that involve scores of the sampling processes $\nabla \log p(t, \mathbf{x})$. However, as shown in Theorem B.1, the score of the sampling process can still explode given sufficiently small score estimation error (see Assumption 3.1) and divergence estimation error (see Assumption 3.3). The crucial aspect of this lemma is that it enables us to avoid the occurrence of the sampling process score $\nabla \log p(t, \mathbf{x})$. We apply Gauss's theorem to transfer the integral to $\partial \Omega_t$, where we can replace $p_t$ with $q_t$ because $p_t = q_t$ on $\partial \Omega_t$. Then, we apply Gauss's theorem again to convert it back to a volume integral. This allows us to eliminate the dependence on $\nabla \log p(t, \mathbf{x})$. Motivated by this lemma, we make the following assumption on divergence estimation error.

**Assumption 3.3** (Divergence estimation error). For any $t \in [0, T - \delta]$, the estimated score function $\mathbf{s}_\theta(t, \cdot)$ is second-order continuously differentiable. Moreover, it satisfies:

$$
\mathbb{E}_{t, \boldsymbol{Y}_t} \Big| \nabla \cdot \mathbf{s}_\theta(T - t, \boldsymbol{Y}_t) - \nabla \cdot \nabla \log q(T - t, \boldsymbol{Y}_t) \Big| \le \epsilon_{\mathrm{div}}.
$$

Similar assumptions regarding the difference between the derivatives of true and estimated scores have been made Li et al. (2024b;a;c). In comparison, those studies assumed the closeness of the entire Jacobian matrix, while we only require assumptions on the divergence. Under Assumptions 3.1 and 3.3, we have the following theorem.

**Theorem 3.4.** Let the true and simulate reverse process be defined as (3.1) and (3.2). Under Assumptions 3.1 and 3.3, if we further assume that $\boldsymbol{X}_0$ has finite second-order momentum, then we have:

$$
\mathrm{TV}(\widehat{\boldsymbol{Y}}_{t_N}, \boldsymbol{X}_\delta) \le \mathrm{TV}(\widehat{\boldsymbol{Y}}_0, \boldsymbol{Y}_0) + \sqrt{dT + d\log \frac{1}{\delta}} \epsilon_{\mathrm{score}} + \epsilon_{\mathrm{div}}.
$$

See Appendix D for the proof of this theorem.

**Remark 3.5.** Huang et al. (2024a) proves a similar bound on the TV-distance for continuous-time processes. Their result differs from ours in that, unlike Assumption 3.3, they do not assume the divergence estimation error to be small. Instead, they require the first two derivatives of the estimated score function to be bounded. Their estimation error term scales as $O(d^{3/4}T^{3/4}\delta^{-1}\epsilon_{\mathrm{score}}^{1/2})$, which is strictly worse than our $O(d^{1/2}(T^{1/2} + \log^{1/2}(\delta^{-1}))\epsilon_{\mathrm{score}})$. Moreover, their results rely on extra assumptions regarding the compact support of the data distribution. For a detailed comparison with Huang et al. (2024a), regarding the settings and results, please refer to Appendix A.1.

## 4 UNIFIED ANALYSIS FOR DISCRETE-TIME REVERSE PROCESS

In this section, we conduct convergence analysis for the discrete-time reverse process. Specifically, we consider a sequence of time steps $0 = t_0 < t_1 < ... < t_N \le T$. Starting from $\widehat{\boldsymbol{Y}}_0 \sim \pi_d$, we iteratively apply a deterministic sampler $\{\boldsymbol{T}_k\}_{k=0}^{N-1}$ to generate subsequent iterations. For any $k$, the sampling process can be expressed as:

$$
\widehat{\boldsymbol{Y}}_{t_{k+1}} = \boldsymbol{T}_k(\widehat{\boldsymbol{Y}}_{t_k}),
$$

where $\boldsymbol{T}_k$ acts as a discrete-time simulation of the transition in the reverse ODE process (4.1). We will present our results in three steps. In Section 4.1, we introduce some commonly used numerical schemes for diffusion models. In Section 4.2, we provide a unified framework encompassing these numerical schemes. We then employ an interpolation method to transform the discrete-time sampling into an equivalent continuous-time ODE, enabling us to leverage Lemma 3.2 from the previous section. In Section 4.3, we present the convergence analysis for this general framework.

## 4.1 NUMERICAL SCHEMES

Recall that for general forward process (2.1), the continuous-time reverse process with the estimated score function $\mathbf{s}_\theta$ can be written as

$$\mathrm{d}\widehat{\boldsymbol{Y}}_t = -\Big(\mathbf{f}(T-t, \widehat{\boldsymbol{Y}}_t) - \frac{1}{2}g(T-t)^2\mathbf{s}_\theta(T-t, \widehat{\boldsymbol{Y}}_t)\Big)\mathrm{d}t, \quad \widehat{\boldsymbol{Y}}_0 \sim q_T. \tag{4.1}$$

**Forward Euler Scheme.** The simplest method is the forward Euler scheme, which directly replaces $t \in [t_k, t_{k+1}]$ with the start point $t_k$ in the equation above, i.e.,

$$\mathrm{d}\widehat{\boldsymbol{Y}}_t = -\Big(\mathbf{f}(T-t_k, \widehat{\boldsymbol{Y}}_{t_k}) - \frac{1}{2}g(T-t_k)^2\mathbf{s}_\theta(T-t_k, \widehat{\boldsymbol{Y}}_{t_k})\Big)\mathrm{d}t, \quad \text{for } t_k < t \le t_{k+1}.$$

Or equivalently, we have the following discrete-time sampling algorithm, with $\eta_k = t_{k+1} - t_k$:

$$\widehat{\boldsymbol{Y}}_{t_{k+1}} = \widehat{\boldsymbol{Y}}_{t_k} - \eta_k\Big(\mathbf{f}(T-t_k, \widehat{\boldsymbol{Y}}_{t_k}) - \frac{1}{2}g(T-t_k)^2\mathbf{s}_\theta(T-t_k, \widehat{\boldsymbol{Y}}_{t_k})\Big). \tag{4.2}$$

**Exponential Integrator (EI) Scheme.** When $\mathbf{f}(t, \mathbf{y}) = L\mathbf{y}$ is a linear function, we can apply the Exponential Integrator (EI) scheme by keeping $\widehat{\boldsymbol{Y}}_t$ in the linear part, i.e., the ODE becomes

$$\mathrm{d}\widehat{\boldsymbol{Y}}_t = -\Big(L\widehat{\boldsymbol{Y}}_t - \frac{1}{2}g(T-t_k)^2\mathbf{s}_\theta(T-t_k, \widehat{\boldsymbol{Y}}_{t_k})\Big)\mathrm{d}t, \quad \text{for } t_k < t \le t_{k+1}.$$

In this way, we can integrate the linear part exactly. As a result, we have the following discrete-time sampling algorithm, with $\eta_k = t_{k+1} - t_k$:

$$\widehat{\boldsymbol{Y}}_{t_{k+1}} = \mathrm{e}^{-L\eta_k}\widehat{\boldsymbol{Y}}_{t_k} + \frac{e^{-L\eta_k} - 1}{2L}g(T-t_k)^2\mathbf{s}_\theta(T-t_k, \widehat{\boldsymbol{Y}}_{t_k}). \tag{4.3}$$

**DDIM-type Scheme.** Song et al. (2020a) introduced a deterministic sampler for the probability flow ODE by considering a non-Markovian diffusion process. As interpreted by Chen et al. (2023b), it can be viewed as a two-step process involving a restoration step that provides a rough estimate for a past step and a degradation step that simulates the forward process by progressively adding the estimated noise. Specifically, starting from $\widehat{\boldsymbol{Y}}_{t_k}$, the restoration step provides an estimate of $\boldsymbol{Y}_{t_k+\gamma}$ for some $\gamma > 0$, where $t_{k+1} - t_k \le \gamma$, i.e.

$$\boldsymbol{Y}_{t_k+\gamma} \approx \widehat{\boldsymbol{Y}}_{t_k} - \gamma\Big[\mathbf{f}(T-t_k, \widehat{\boldsymbol{Y}}_{t_k}) - g(T-t_k)^2\mathbf{s}_\theta(T-t_k, \widehat{\boldsymbol{Y}}_{t_k})\Big] = \mathbf{z}.$$

Next, the degradation step simulates the forward process during $t \in [T - t_k - \gamma, T - t_{k+1}]$, which can be expressed as

$$\widehat{\boldsymbol{Y}}_{t_{k+1}} = \mathbf{z} + (t_k + \gamma - t_{k+1})\mathbf{f}(T - t_k - \gamma, \mathbf{z}) + g(T - t_k - \gamma)\sqrt{t_k + \gamma - t_{k+1}}\boldsymbol{\epsilon},$$

where $\boldsymbol{\epsilon}$ represents the noise estimated from $\widehat{\boldsymbol{Y}}_{t_k}$. By substituting the form of $\boldsymbol{\epsilon}$ and making some approximations, we can get the following sampling algorithm, with $\eta_k = t_{k+1} - t_k$:

$$\widehat{\boldsymbol{Y}}_{t_{k+1}} = \widehat{\boldsymbol{Y}}_{t_k} - \eta_k\mathbf{f}\big(T - t_k, \widehat{\boldsymbol{Y}}_{t_k}\big) + l\eta_k\big(1 - \sqrt{1 - 1/l}\big)g(T-t_k)^2\mathbf{s}_\theta(T-t_k, \widehat{\boldsymbol{Y}}_{t_k}), \tag{4.4}$$

where $l = \gamma/(t_{k+1} - t_k)$. Please refer to Chen et al. (2023b) for more details about the DDIM-type sampler. For linear diffusions, we can set $\gamma = T - t_k$. Then, with our selection of time schedule, by (2.6), we have $l \ge (T - t_{k+1})/(t_{k+1} - t_k) \ge \min\{1, T - t_{k+1}\}/(t_{k+1} - t_k) \ge 1/\eta$.

## 4.2 INTERPOLATION METHODS

At the core of our analysis is to apply Lemma 3.2, which analyzes the divergence between two ODEs. However, two main challenges arise. First, the discrete-time nature of the sampling algorithm precludes direct application of the lemma. Second, the sampling process $\widehat{\boldsymbol{Y}}_{t_{k+1}} = \boldsymbol{T}_k(\widehat{\boldsymbol{Y}}_{t_k})$ depends on the position $\widehat{\boldsymbol{Y}}_{t_k}$ at time $t_k$, while the proof of Lemma 3.2 utilizes the Fokker-Planck equation,

which requires the drift term to be a function solely of time and the current position. To solve these problems, we first introduce a unified framework encompassing all the numerical schemes in Section 4.1. Next, we present an interpolation method to transform the sampling process into a continuous-time ODE, enabling the application of Lemma 3.2.

For the numerical schemes defined in (4.2), (4.3) and (4.4), we naturally extend the definition to a continuous interval $t \in [t_k, t_{k+1}]$ by replacing $t_{k+1}$ with $t$. More concrete examples of $F_t$ can be found in Section 5. This yields a continuous-time interpolation operator $F_{t_k \to t}(\cdot)$, or simply $F_t$ when no confusion arises. Moreover, let $\widehat{\boldsymbol{Y}}_t = F_t(\widehat{\boldsymbol{Y}}_{t_k}) = F(t, \widehat{\boldsymbol{Y}}_{t_k})$. It is also equivalent to the following ODE:

$$\mathrm{d}\boldsymbol{Y}_t = \frac{\partial F}{\partial t}(t, \widehat{\boldsymbol{Y}}_{t_k})\mathrm{d}t.$$

Moreover, if we further assume $F_t$ is invertible, which holds for many examples when $\mathbf{s}_\theta$ is Lipschitz and the time step $\eta_k = t_{k+1} - t_k$ is small enough. Then we have the following ODE:

$$\mathrm{d}\widehat{\boldsymbol{Y}}_t = \widehat{\mathbf{b}}(t, \widehat{\boldsymbol{Y}}_t)\mathrm{d}t, \tag{4.5}$$

where $\widehat{\mathbf{b}}(t, \mathbf{x}) = \frac{\partial F}{\partial t}\big(t, F_t^{-1}(\mathbf{x})\big)$.

### 4.3 MAIN RESULTS FOR GENERAL DIFFUSION PROCESSES

Using the interpolation method presented in the last section, we can now provide a general convergence analysis for the discrete-time reverse process. Consider the time step $t \in [t_k, t_{k+1}]$. Recall that for general forward process (2.1), the true reverse process is defined as

$$\mathrm{d}\boldsymbol{Y}_t = \mathbf{b}(t, \boldsymbol{Y}_t)\mathrm{d}t,$$

where $\mathbf{b}(t, \mathbf{x}) = -\big(\mathbf{f}(T - t, \mathbf{x}) - \frac{1}{2}g(T - t)^2 \nabla \log q_{T-t}(\mathbf{x})\big)$. While the simulated reverse process is given by:

$$\mathrm{d}\widehat{\boldsymbol{Y}}_t = \widehat{\mathbf{b}}(t, \widehat{\boldsymbol{Y}}_t)\mathrm{d}t,$$

where $\widehat{\mathbf{b}}(t, \mathbf{x}) = \frac{\partial F}{\partial t}\big(t, F_t^{-1}(\mathbf{x})\big)$.

**Definition 4.1.** At each step $[t_k, t_{k+1}]$, let the interpolation operator of the sampling algorithm be $F_{t_k \to t}$. The estimation-error operator is defined by:

$$\Phi_k(t, \mathbf{x}) = \frac{\partial F}{\partial t}(t, \mathbf{x}) + \mathbf{f}\big(T - t, F_{t_k \to t}(\mathbf{x})\big) - \frac{1}{2}g(T - t)^2 \nabla \log q_{T-t_k}(\mathbf{x}).$$

As we will show in the next section, it reflects the error between true score $\nabla \log q_{T-t_k}(\mathbf{x})$ and $\mathbf{s}_\theta(T - t_k, \mathbf{x})$. Similarly, we can define the divergence-error operator, which reflects the error between $\nabla^2 \log q_{T-t_k}(\mathbf{x})$ and $\nabla \mathbf{s}_\theta(T - t_k, \mathbf{x})$.

**Definition 4.2.** At each step $[t_k, t_{k+1}]$, let the interpolation operator of the sampling algorithm be $F_{t_k \to t}$. The divergence-error operator is defined by

$$\Psi_k(t, \mathbf{x}) = \nabla\Big[\frac{\partial}{\partial t}F\Big](t, \mathbf{x})\mathbf{I} + \nabla_\mathbf{x}\big[\mathbf{f}\big(T - t, F_{t_k \to t}(\mathbf{x})\big)\big] - \frac{1}{2}g(T - t)^2 \nabla^2 \log q_{T-t_k}(\mathbf{x}).$$

Using these definitions, the next theorem shows the convergence for the general diffusion process with any numerical schemes.

**Theorem 4.3.** Consider the true reverse process $(\boldsymbol{Y}_t)_{t \in [0,T]}$ and reverse sampling process $(\widehat{\boldsymbol{Y}}_{t_k})_{k \in [N]}$. Then,

$$\mathrm{TV}(\boldsymbol{Y}_{T-\delta}, \widehat{\boldsymbol{Y}}_{t_N}) \leq \mathrm{TV}(q_T, \pi_d)$$

$$+ \sum_{k=0}^{N-1} \int_{t_k}^{t_{k+1}} \Bigg[ \underbrace{\sqrt{\mathbb{E}\big[\|\Phi_k(t, \boldsymbol{Y}_{t_k})\|^2\big]} \sqrt{\int \Big\|\frac{\nabla \log q(T - t, \mathbf{x})p_{\boldsymbol{Y}_t}(\mathbf{x})}{p_{F_t(\boldsymbol{Y}_{t_k})}(\mathbf{x})}\Big\|^2 p_{F_t(\boldsymbol{Y}_{t_k})}(\mathbf{x})\mathrm{d}\mathbf{x}}}_{\text{(I) Score estimation error}}$$

$$+ \underbrace{\sqrt{\mathbb{E}\big[\mathrm{tr}\big(\Psi_k(t, \boldsymbol{Y}_{t_k})\big)^2\big]} \sqrt{\int \Big(\frac{p_{\boldsymbol{Y}_t}(\mathbf{x})}{p_{F_t(\boldsymbol{Y}_{t_k})}(\mathbf{x})}\Big)^2 p_{F_t(\boldsymbol{Y}_{t_k})}(\mathbf{x})\mathrm{d}\mathbf{x}}}_{\text{(II) Divergence estimation error}}$$

$$+ \frac{1}{2}g(T-t)^2 \int \left| \left( \nabla \log q_{T-t_k}(\mathbf{z}) - \nabla \log q_{T-t}(\mathbf{x}) \right) \cdot \nabla \log q(T-t, \mathbf{x}) \right| q(T-t, \mathbf{x}) \mathrm{d}\mathbf{x}$$

$$\underbrace{\hphantom{+ \frac{1}{2}g(T-t)^2 \int \left| \left( \nabla \log q_{T-t_k}(\mathbf{z}) - \nabla \log q_{T-t}(\mathbf{x}) \right) \cdot \nabla \log q(T-t, \mathbf{x}) \right| q(T-t, \mathbf{x}) \mathrm{d}\mathbf{x}}}_{\text{(III) Score discretization error}}$$

$$+ \frac{1}{2}g(T-t)^2 \int \left| \mathrm{tr}\left( \nabla^2 \log q(T-t_k, \mathbf{z}) - \nabla^2 \log q(T-t, \mathbf{x}) \right) \right| q(T-t, \mathbf{x}) \mathrm{d}\mathbf{x}$$

$$\underbrace{\hphantom{+ \frac{1}{2}g(T-t)^2 \int \left| \mathrm{tr}\left( \nabla^2 \log q(T-t_k, \mathbf{z}) - \nabla^2 \log q(T-t, \mathbf{x}) \right) \right| q(T-t, \mathbf{x}) \mathrm{d}\mathbf{x}}}_{\text{(IV) Divergence discretization error}}$$

$$+ \max_{\mathbf{x}} \left| \mathrm{tr} \left[ \nabla \left[ \frac{\partial}{\partial t} F \right] \left( t, F_t^{-1}(\mathbf{x}) \right) + \nabla_{\mathbf{z}} \left[ \mathbf{f}(T-t, F_t(\mathbf{z})) \right] \right] \left( \nabla F_t^{-1}(\mathbf{x}) - \mathbf{I} \right) \right| \right] \mathrm{d}t,$$

$$\underbrace{\hphantom{+ \max_{\mathbf{x}} \left| \mathrm{tr} \left[ \nabla \left[ \frac{\partial}{\partial t} F \right] \left( t, F_t^{-1}(\mathbf{x}) \right) + \nabla_{\mathbf{z}} \left[ \mathbf{f}(T-t, F_t(\mathbf{z})) \right] \right] \left( \nabla F_t^{-1}(\mathbf{x}) - \mathbf{I} \right) \right|}}_{\text{(V) Bias error}}$$

where $\mathbf{z} = F_t^{-1}(\mathbf{x})$.

Other than the distance between the initial distribution of the reverse process and Gaussian noise, our upper bound of TV distance is divided into five terms.

For (I) and (II), these terms rely on the expectation of the defined estimation-error operator and divergence-error operator over the true reverse process. Moreover, they depend on the density ratios between $p_{\mathbf{Y}_t}(\mathbf{x})$, representing the distribution for the true reverse process at time $t$, and $p_{F_t(\mathbf{Y}_{t_k})}(\mathbf{x})$. As the interpolation operator $F_t$ acts as a simulation of the true reverse process, we expect that the ratio will be close to 1, thus bounded. Consequently, these terms are expected to scale with the score and divergence estimation errors. For (III) and (IV), these terms depend on he distance between score functions and divergence evaluated at $t$ and $t_k$, originating from the time-discretization algorithm. They will decrease as the time step gets smaller. For (V), we observe that $F_t$ is close to the identity when $t \to t_k$. This results in $\nabla F_t^{-1}(\mathbf{x}) - \mathbf{I}$ converging to the zero matrix when $t \to t_k$.

## 5 APPLICATION TO SPECIFIC DETERMINISTIC SAMPLERS

In this section, we apply Theorem 4.3 to analyze specific diffusion processes. We focus on the Variance Preserving (VP) forward process with EI schemes and the Variance Exploding (VE) forward process with the DDIM sampler. While prior work has shown that VP and VE can be connected through reparametrization (Karras et al., 2022), such equivalence does not account for initialization and discretization errors. Therefore, we provide separate detailed analyses for both VP and VE processes. The analysis method can be easily extended to other forward processes and numerical schemes.

We specifically focus on data distributions with compact support, as outlined in the following assumption.

**Assumption 5.1** (Bounded Support of Data). For a constant $R$, the data distribution $q_0$ satisfies:

$$q_0(\mathbf{x}) = 0, \quad \forall \|\mathbf{x}\| > R,$$

or equivalently, $P(\|\boldsymbol{X}_0\| > R) = 0$.

This assumption has also been made in De Bortoli (2022); Chen et al. (2022). In particular, we do not assume the smoothness of the data distribution. Therefore, it includes the setting where the data distribution is supported on a lower-dimensional submanifold of $\mathbb{R}^d$, which, notably, does not possess a smooth density.

Additionally, we make the following assumptions on the estimated score function.

**Assumption 5.2** (Score Estimation Error).

$$\sum_k \eta_k \mathbb{E} \| \mathbf{s}_\theta(T-t_k, \boldsymbol{Y}_{t_k}) - \nabla \log q(T-t_k, \boldsymbol{Y}_{t_k}) \|^2 \le \epsilon_{\text{score}}^2.$$

**Assumption 5.3** (Divergence Estimation Error).

$$\sum_k \eta_k \sqrt{\mathbb{E}\, \mathrm{tr}\left( \nabla \mathbf{s}_\theta(T-t_k, \boldsymbol{Y}_{t_k}) - \nabla^2 \log q(T-t_k, \boldsymbol{Y}_{t_k}) \right)^2} \le \epsilon_{\text{div}}.$$

In addition, to deal with the discretization error for the ODE reverse process, we need the following regularity conditions for $\mathbf{s}_\theta$.

**Assumption 5.4** ($\mathbf{s}_\theta$ is Lipschitz and bounded at 0). For all $t_k$, we have:

$$\| \mathbf{s}_\theta(T-t_k, \mathbf{x}_1) - \mathbf{s}_\theta(T-t_k, \mathbf{x}_2) \| \le L \cdot \|\mathbf{x}_1 - \mathbf{x}_2\|,$$

$$\|\mathbf{s}_\theta(T - t_k, 0)\| \leq c,$$

for some constant $L$ and $c$. Without loss of generality, we assume $L \geq 1$ and $c \leq L$ to simplify later derivation. Similar assumptions on $\mathbf{s}_\theta$ have been made in Huang et al. (2024a). Note that we only need the Lipschitzness and boundness at 0, while they require the boundness of high-order derivatives w.r.t $t$ and $\mathbf{x}$.

### 5.1 VP+EI

In this section, we focus on the VP process where $\mathbf{f}(t, \mathbf{x}) = -\mathbf{x}$ and $g(t) = \sqrt{2}$. As is discussed in Section 4.1, the sampling algorithm is given by:

$$\widehat{\boldsymbol{Y}}_{t_{k+1}} = e^{t_{k+1} - t_k} \widehat{\boldsymbol{Y}}_{t_k} + \left(e^{t_{k+1} - t_k} - 1\right) \mathbf{s}_\theta(T - t_k, \widehat{\boldsymbol{Y}}_{t_k}).$$

At each step $[t_k, t_{k+1}]$, $F_t(\mathbf{x}) = e^{t - t_k}\mathbf{x} + \left(e^{t - t_k} - 1\right)\mathbf{s}_\theta(T - t_k, \mathbf{x})$. Therefore, the estimation-error operator (Definition 4.1) can be computed as

$$\Phi_k(t, \mathbf{x}) = \frac{\partial F}{\partial t}(t, \mathbf{x}) + \mathbf{f}\left(T - t, F_t(\mathbf{x})\right) - \frac{1}{2}g(T - t)^2 \nabla \log q_{T - t_k}(\mathbf{x})$$
$$= s_\theta(T - t_k, \mathbf{x}) - \nabla \log q_{T - t_k}(\mathbf{x}).$$

Similarly, the divergence-error operator (Definition 4.2) can be computed as:

$$\Psi_k(t, \mathbf{x}) = \nabla\left[\frac{\partial}{\partial t}F\right](t, \mathbf{x})\mathbf{I} + \nabla_{\mathbf{x}}\left[\mathbf{f}\left(T - t, F_{t_k \to t}(\mathbf{x})\right)\right] - \frac{1}{2}g(T - t)^2 \nabla^2 \log q_{T - t_k}(\mathbf{x})$$
$$= \nabla \mathbf{s}_\theta(T - t_k, \mathbf{x}) - \nabla^2 \log q_{T - t_k}(\mathbf{x}).$$

Now, using Theorem 4.3, we have the following convergence analysis for VP + EI. See Appendix F for proofs of the following results.

**Theorem 5.5.** Consider a VP forward process with EI numerical scheme. Under Assumptions 5.1, 5.2, 5.3, and 5.4, suppose the time schedule satisfies (2.6). If the step size $\eta$ satisfies $\eta \leq \min\left\{1/(12L^2d^2), 1/(24L^2R^2d)\right\}$, then the following bound holds:

$$\mathrm{TV}(\widehat{\boldsymbol{Y}}_{t_N}, \boldsymbol{X}_\delta) \lesssim \mathrm{TV}(q_T, \pi_d) + \epsilon_{\mathrm{div}} + \sqrt{d}\sqrt{\eta N}\epsilon_{\mathrm{score}} + \eta^2 N\left[LR^4\left(d^2 + \frac{1}{\delta^2}\right) + L^2d + \frac{R^5}{\delta}\right].$$

**Lemma 5.6.** Assumes the data distribution satisfies $\mathrm{Cov}(q_0) = \mathbf{I}_d$. Then for the VP process, we have $\mathrm{TV}(q_T, \pi_d) \lesssim \sqrt{d}e^{-T}$. At VP forward process case, we take $\pi_d \sim N(0, \mathbf{I}_d)$.

**Corollary 5.7.** For all $T \geq 1$, $\delta < 1$ and $N \geq \log(1/\delta)$, there exists $\eta = \Theta((T + \log\frac{1}{\delta})/N)$ and a time schedule satisfying (2.6). Under the same assumptions as Theorem 5.5, we additionally assume that the data distribution satisfies $\mathrm{Cov}(q_0) = \mathbf{I}_d$. When $\delta = 1/d$ and $d \geq R/L + L/R^4$, if we take $T = \log(d/\epsilon^2)/2$ and $N = LR^4\Theta\left(d^2(T + \log(1/\delta))^2/\epsilon\right)$ for some $\epsilon \leq 1/L$, we have $\mathrm{TV}(\widehat{\boldsymbol{Y}}_{t_N}, \boldsymbol{X}_\delta) \lesssim \epsilon$, assuming $\widetilde{O}(\epsilon/\sqrt{d})$ score estimation error and $O(\epsilon)$ divergence error. Hence the diffusion model requires at most $\widetilde{O}(LR^4d^2/\epsilon)$ steps to approximate $q_\delta$ within $\epsilon \leq 1/L$ in TV distance.

**Remark 5.8.** We note that the $\epsilon$-dependence in iteration complexity of our deterministic sampler is $\epsilon^{-1}$, while stochastic ones usually exhibit a slower iteration complexity proportional to $\epsilon^{-2}$(Chen et al., 2023a; 2022; Li et al., 2024b; Benton et al., 2024). This aligns with general observation in diffusion model practice: deterministic samplers demonstrate higher efficiency compared to stochastic samplers.

### 5.2 VE+DDIM

In this section, we focus on the process where $\mathbf{f}(t, \mathbf{x}) = 0$ and $g(t) = 1$, which corresponds to the VE forward process with $\sigma_t^2 = t$. As discussed in Section 4.1, the sampling algorithm is given by:

$$\widehat{\boldsymbol{Y}}_{t_{k+1}} = \widehat{\boldsymbol{Y}}_{t_k} + (t_{k+1} - t_k)l\left(1 - \sqrt{1 - 1/l}\right)\mathbf{s}_\theta(T - t_k, \widehat{\boldsymbol{Y}}_{t_k}), \quad l = \frac{T - t_k}{t_{k+1} - t_k}.$$

Let $c_l = l\left(1 - \sqrt{1 - 1/l}\right) \leq 1$. At each step $[t_k, t_{k+1}]$, $F_t(\mathbf{x}) = \mathbf{x} + c_l(t - t_k)\mathbf{s}_\theta(T - t_k, \mathbf{x})$. Therefore, the estimation-error operator (Definition 4.1) can be computed as

$$\Phi_k(t, \mathbf{x}) = \frac{\partial F}{\partial t}(t, \mathbf{x}) + \mathbf{f}\left(T - t, F_t(\mathbf{x})\right) - \frac{1}{2}g(T - t)^2 \nabla \log q_{T - t_k}(\mathbf{x})$$

$$= c_l s_\theta(T - t_k, \mathbf{x}) - \frac{1}{2} \nabla \log q_{T-t_k}(\mathbf{x}).$$

Similarly, the divergence-error operator (Definition 4.2) can be computed as:

$$\Psi_k(t, \mathbf{x}) = \nabla \Big[ \frac{\partial}{\partial t} F \Big](t, \mathbf{x}) \mathbf{I} + \nabla_{\mathbf{x}} \big[ \mathbf{f}\big(T - t, F_{t_k \to t}(\mathbf{x})\big) \big] - \frac{1}{2} g(T - t)^2 \nabla^2 \log q_{T-t_k}(\mathbf{x})$$

$$= c_l \nabla \mathbf{s}_\theta(T - t_k, \mathbf{x}) - \frac{1}{2} \nabla^2 \log q_{T-t_k}(\mathbf{x}).$$

Now, using Theorem 4.3, we have the following convergence analysis for VE + DDIM. See Appendix G for proofs of the following results.

**Theorem 5.9.** Consider a VE forward process with DDIM numerical scheme. Under Assumptions 5.1, 5.2, 5.3, and 5.4, suppose the time schedule satisfies (2.6). If the step size $\eta$ satisfies $\eta \leq \min\{1/(12L^2Td^2), 1/(24L^2R^2d)\}$, then the following bound holds:

$$\mathrm{TV}(\widehat{\mathbf{Y}}_{T-\delta}, \mathbf{Y}_{T-\delta}) \lesssim \mathrm{TV}(\pi_d, q_T) + \epsilon_{\mathrm{div}} + \sqrt{d}\sqrt{\eta N} \epsilon_{\mathrm{score}} + \eta^2 N \Big[ LR^4 d^2 + L^2 d + \frac{LR^4}{\delta^2} \Big].$$

**Lemma 5.10.** Assume the data distribution satisfies $\mathrm{Cov}(q_0) = \mathbf{I}_d$. Then for the VE forward process, we have $\mathrm{TV}(q_T, \pi_d) \lesssim \sqrt{d}/\sqrt{T}$. At VE forward process case, we take $\pi_d \sim N(0, T\mathbf{I}_d)$.

**Corollary 5.11.** For all $T \geq 1$, $\delta < 1$ and $N \geq \log(1/\delta)$, there exists $\eta = \Theta((T + \log \frac{1}{\delta})/N)$ and a time schedule satisfying (2.6). Under the same assumptions as Theorem 5.5, we additionally assume that the data distribution satisfies $\mathrm{Cov}(q_0) = \mathbf{I}_d$. When $\delta = 1/d$ and $d \geq L/R^4$, if we take $T = d/\epsilon^2$ and $N = LR^4 \Theta\big(d^2(T + \log(1/\delta))^2/\epsilon\big)$ for some $\epsilon \leq 1/L$, we have $\mathrm{TV}(\widehat{\mathbf{Y}}_{t_N}, \mathbf{X}_\delta) \lesssim \epsilon$, assuming sufficiently small score estimation error and divergence error. Hence the diffusion model requires at most $\widetilde{O}(LR^4d^4/\epsilon^5)$ steps to approximate $q_\delta$ within $\epsilon \leq 1/L$ in TV distance.

**Remark 5.12.** Compared to Corollary 5.7, VE SDE has larger iteration complexity, primarily due to the slow $1/\sqrt{T}$ decay in the distance between $q_T$ and $\pi_d$, compared with the exponential decay (Lemma 5.6) for VP SDE.

## 6 CONCLUSION

In this work, we introduce a unified convergence analysis framework for deterministic samplers. We start by presenting a counter-example to illustrate the main challenge in analyzing deterministic samplers compared to stochastic ones. Additionally, we provide a technical lemma that allows us to bound the distance between distributions using score estimation error and divergence error. With this approach, we directly established convergence guarantees for the continuous-time reverse ODE. Moreover, we extend our analysis to the convergence of discrete-time deterministic samplers with a unified framework. Finally, we demonstrate its effectiveness by applying it to two widely adopted sampling methods.

**Limitation and Future Work.** First, for the VP process with EI schemes, our current results have a quadratic dependence on $d$, which leaves room for improvement compared to the $d$-linear state-of-the-art bounds in ODE analysis (Li et al., 2024c). This discrepancy stems from two factors: our different assumption that requires control of the divergence error rather than the full Jacobian error, and our directly estimated Lipschitz constants in the discretization error analysis. We believe that through more delicate methods for analyzing the discretization error and potentially stronger conditions, we could achieve linear dimension dependence within our unified framework. Second, since adding the divergence assumption can guarantee the convergence of the ODE sampler, an interesting future direction may be to design training methods that can obtain both small score error and small divergence error. One potential approach would be to incorporate regularization terms corresponding to divergence error in the loss function, potentially leading to more effective diffusion model training algorithms. Third, for the VE process, we obtain only polynomial bounds due to the slow $1/\sqrt{T}$ decay in the distance between $q_T$ and $\pi_d$. It's important to note that this is not a limitation of the DDIM sampler, as applying DDIM to the VP processes yields results comparable to those obtained with the EI scheme. We leave an improved analysis for the VE forward process as future work. Finally, the discrete-time analysis currently relies on a bounded support assumption for the data distribution, which may be relaxed to less restrictive conditions, such as light-tailed distributions.

ACKNOWLEDGMENTS

We thank the anonymous reviewers and area chair for their helpful comments. QD and QG are supported in part by the NSF grants IIS-2008981, CPS-2312094 and Sloan Research Fellowship. The views and conclusions contained in this paper are those of the authors and should not be interpreted as representing any funding agencies.

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

## A RELATED WORK

**Convergence Analysis of Stochastic Samplers.** Early theoretical studies for diffusion models were either non-quantitative (De Bortoli et al., 2021; Liu et al., 2022; Pidstrigach, 2022), or exhibited exponential dependence on the dimension or other problem parameters (Block et al., 2020; De Bortoli, 2022). Later, Lee et al. (2022) proved the first result with polynomial complexity, with the assumptions $L_2$-score estimate and log-Sobolev inequality (LSI). However, the LSI condition on the data distribution is restrictive, prompting further studies to relax this assumption. Chen et al. (2022) utilized Girsanov's theorem, and proved polynomial convergence bounds, assuming either the Lipschitzness property of the forward process score function or bounded support of the data distribution. Lee et al. (2023) relaxed the smoothness condition to apply only to the data distribution rather than the whole trajectory, and replaced the bounded support assumption with sufficient tail decay. Chen et al. (2023a) proved a result with both the advantages of these two works, achieving a better convergence rate while assuming only the smoothness of the data distribution. Furthermore, they provided results for the non-smooth setting using appropriate early stopping and decreasing step size. Benton et al. (2024) improved the dependency of the dimension $d$ to linear via the stochastic localization method. Compared with the analysis of reverse SDE, Li et al. (2024b) utilized an elementary approach to analyze a DDPM-type stochastic sampler. It also proposed an accelerated stochastic sampler with better iteration complexity. More recently, Huang et al. (2024b) decomposed the entire sampling process into several reverse transition kernel subproblems and proposed novel fast sampling algorithms.

**Convergence Analysis of Deterministic Samplers.** The first non-asymptotic bounds for deterministic samplers were derived by Chen et al. (2023b), which assumes access to the ground truth score function. Subsequently, Chen et al. (2024) proved the first polynomial-time convergence guarantees of probability flow ODE with estimation error, by incorporating an additional corrector Langevin dynamics. While this improved upon prior results, it introduced randomness, making the sampling processes non-deterministic. Li et al. (2024a;b) applied an elementary analysis framework to study the convergence of a specific deterministic sampler with a given learning schedule, requiring an additional assumption on the Jacobian estimation error. Later, Li et al. (2024c) applied the same framework and achieved improved iteration complexity, which is linear in the dimension. Recently, Huang et al. (2024a) examined the Ornstein-Uhlenbeck (OU) forward process, removing the assumption of Jacobian estimation error. However, they introduced higher-order boundedness assumptions on the derivatives of the estimated score function concerning time and space. They proved an upper bound for the total variation between the target and generated data at the continuous-time level while also analyzing the convergence rate for the Runge-Kutta integrator. Further expanding the scope, Gao & Zhu (2024) studied the Wasserstein distance using general forward SDE with log-concavity data assumptions. Additionally, research has started to explore other deterministic generation methods, such as flow matching (Benton et al., 2023).

While we prepared our manuscript, we noticed the concurrent work in (Li et al., 2024c). They analyzed a specific discrete-time sampling algorithm that can be viewed as a deterministic counterpart of DDPM (Ho et al., 2020), achieving a better $\widetilde{O}(d/\epsilon)$ iteration complexity. The improvement arises from their fine-grained analysis of discretization error through an elementary approach. The focus of their paper is orthogonal to ours, as they provide an improved analysis of a specific algorithm, while our work aims to develop a unified analysis framework. We conjecture that their techniques can be adapted into our framework, potentially achieving improved results. Nevertheless, it is beyond the goal of our paper and we will leave it for future work.

### A.1 COMPARISON WITH HUANG ET AL. (2024A)

Huang et al. (2024a) is the most related work to ours. It studied the convergence properties of diffusion models with deterministic samplers based on probability flow ODEs.

Theorem A.1 in Huang et al. (2024a) is similar to our Lemma 3.2, where they applied the characteristic line method for ODEs, while we used the Fokker-Planck equation and the Gauss's theorem twice. However, we use different methods when dealing with the divergence term

$$\nabla \cdot \left[ \left( \mathbf{b}(t, \mathbf{x}) - \mathbf{b}^*(t, \mathbf{x}) \right) q(T - t, \mathbf{x}) \right].$$

We decompose it into score estimation error and divergence estimation error (see Lemma 3.2). In contrast, they applied the Gagliardo-Nirenberg inequality to bound the integral of first-order derivatives using integrals of second-order and zero-order derivatives of $\mathbf{s}_\theta$. This led to different assumptions regarding the convergence analysis of the continuous-time reverse process. In addition to the standard estimation error assumption, we require a small divergence error, while Huang et al.

(2024a) assumed the boundedness of up to the second order derivatives of $\mathbf{s}_\theta$. Furthermore, they required a bounded support assumption, which is not necessary for our continuous-time analysis. As a result, Huang et al. (2024a) proved an upper bound on the TV distance: $O(T^{3/4}d^{3/4}(1/\delta)\epsilon_{\text{score}}^{1/2})$. In comparison, our result is $O(\sqrt{dT + d\log(1/\delta)} \cdot \epsilon_{\text{score}})$, notably only log-dependent on $\delta$.

For the discrete-time analysis, Huang et al. (2024a) studied the $p$-th Runge-Kutta discretization method. When $p = 1$, it becomes the forward Euler method. In comparison, we study the EI scheme. Our analysis can be easily applied to the forward Euler method, yielding similar results. In the discrete-time case, we assumed bounded support of the data distribution and the Lipschitz condition of $\mathbf{s}_\theta$, which was already required in Huang et al. (2024a) in the continuous-time analysis. Additionally, they assumed that the mixed second-order derivatives of $\mathbf{s}_\theta$ with respect to $t$ and $x$ are bounded by a constant $\bar{L}$. For the forward Euler method, Huang et al. (2024a) proved iteration complexity of $\widetilde{O}(\bar{L}^2 R^2 d^2 \epsilon^{-1})$, while our result for EI scheme is $\widetilde{O}(LR^4 d^2 \epsilon^{-1})$ for $\delta = 1/d$. We remark that the constant $\bar{L}$ in their result may be very large. For example, the derivatives of the true score function can depend on higher-order terms of $1/\delta$ (see Lemma H.2). Moreover, we also consider the VE forward process with the DDIM numerical schemes.

## A.2 COMPARISON WITH OTHER WORKS ON DETERMINISTIC SAMPLERS

In this section, we consider different works on convergence analysis of diffusion models with deterministic samplers and provide a detailed comparison in Table 1. The comparison focuses on three key aspects: major assumptions required by these works, the metrics used, and the iteration complexity in terms of dimension $d$ and accuracy $\epsilon$.

| Major Assumptions | Metric | Complexity | Reference |
|---|---|---|---|
| Score Estimation Error [1] 
 Bounded Support | | $\widetilde{O}\left(\frac{d^2}{\epsilon} + \frac{d^3}{\sqrt{\epsilon}}\right)$ | Li et al. (2024b) 
 Theorem 1 |
| $\frac{1}{N}\sum_{t=1}^{N} \mathbb{E}_{\boldsymbol{X} \sim q_t} \left\|\nabla\mathbf{s}_\theta(t, \boldsymbol{X}) - \nabla^2 \log q(t, \boldsymbol{X})\right\| \le \epsilon_{\text{Jacobi}}$ | $\text{TV}(p_1, q_1)$ | $\widetilde{O}\left(\frac{d}{\epsilon}\right)$ | Li et al. (2024c) 
 Theorem 1 |
| Access to Exact Score 
 $\nabla \log q$ and $\nabla^2 \log q$ Lipschitz 
 $\left\|\nabla \log \frac{q_t}{q_s}(\mathbf{x})\right\| \le \beta\|t - s\|^c(1 + \|\mathbf{x}\| + \|\nabla q_t(\mathbf{x})\|)$ | $\text{KL}(p_0, q_0)$ | Polynomial | Chen et al. (2023b) 
 Theorem 4.1 |
| Score Estimation Error 
 $\sup_{\mathbf{x}\in\mathbb{R}^d} \max_{1 \le k+\|\alpha\| \le p+1} \max_{1 \le j \le d} \|\partial_t^k \partial_{\mathbf{x}}^\alpha \mathbf{s}_\theta^{(j)}(t, \mathbf{x})\| \le L$ 
 Bounded Support | $\text{TV}(p_\delta, q_\delta)$ | $O\left(\frac{d^{(p+1)/p}}{\epsilon^{1/p}}\right)$ | Huang et al. (2024a) 
 Theorem 3.10 |
| Bounded Support: Assumption 5.1 
 Score Estimation Error: Assumption 5.2 
 Divergence Error: Assumption 5.3 | $\text{TV}(p_\delta, q_\delta)$ | $\widetilde{O}\left(\frac{d^2}{\epsilon}\right)$ | This Work: 
 Corollary 5.7 |

Table 1: Comparison of convergence analysis for diffusion models with deterministic samplers

## B PROOF OF THEOREM 2.1

*Proof of Theorem 2.1.* Since $x_t$ starts at $N(0, 1)$, we can easily show that $x_t \sim N(0, 1)$ for all $t$. Recall that we use $q(t, x)$ to denote the law of $x_t$, then we have $q(t, x) = \frac{1}{\sqrt{2\pi}}e^{-\frac{x^2}{2}}$ and $\nabla \log q(t, x) = -x$. Define $\widehat{q}(T - t, x) = \frac{1}{\sqrt{2\pi}}e^{-\frac{x^2}{2}}\left(1 + \frac{t}{2T}\sin(2n\pi x)\right)$, and construct our estimated score as:

$$\mathbf{s}_\theta(T - t, x) = \frac{\int_x^\infty \frac{1}{\sqrt{2\pi}}e^{-\frac{y^2}{2}}\sin(2n\pi y)\left(\frac{1}{2T}\right)\mathrm{d}y}{\widehat{q}(T - t, x)} - x,$$

---

[1] This assumption is slightly different from ours. While it assigns uniform weights to estimation errors across all time steps, our approach applies smaller weights to time steps with smaller step sizes.

Without loss of generality, we can assume $x \geq 0$. Then we have:

$$
\begin{aligned}
|\mathbf{s}_\theta(T - t, x) - \nabla \log q(T - t, x)| &= \frac{\left| \int_x^\infty \frac{1}{\sqrt{2\pi}} e^{-\frac{y^2}{2}} \sin(2n\pi y) \left( \frac{1}{2T} \right) \mathrm{d}y \right|}{\frac{1}{\sqrt{2\pi}} e^{-\frac{x^2}{2}} \left( 1 + \frac{T-t}{2T} \sin(2n\pi x) \right)} \\
&\leq \frac{\frac{1}{\sqrt{2\pi}} e^{-\frac{x^2}{2}} \frac{1}{2T} \sup_{N \geq x} \left| \int_x^N \sin(2n\pi y) \mathrm{d}y \right|}{\frac{1}{\sqrt{2\pi}} e^{-\frac{x^2}{2}} \frac{1}{2}} \\
&\leq \frac{1}{T} \frac{1}{n\pi}.
\end{aligned}
$$

Taking $n \geq 1/(T\pi\epsilon)$, we have $|\mathbf{s}_\theta(T - t, x) - \nabla \log q(T - t, x)| \leq \epsilon$. Moreover, since we have

$$
\frac{\partial \widehat{q}(T - t, x)}{\partial t} = -\frac{\partial}{\partial x} \left( \left( \mathbf{s}_\theta(T - t, x) + x \right) \cdot \widehat{q}(T - t, x) \right), \quad \widehat{q}(T - 0, x) \sim N(0, 1),
$$

which satisfies the Fokker-Planck equation. Therefore, we know that $\widehat{q}(t, x)$ is the law of $\widehat{y}_{T-t}$. Hence the law of $\widehat{y}_T$ is

$$
\widehat{q}_0(x) = \frac{1}{\sqrt{2\pi}} e^{-\frac{x^2}{2}} \left( 1 + \frac{1}{2} \sin(2n\pi x) \right).
$$

The TV distance between $y_T$ and $\widehat{y}_T$ is:

$$
\begin{aligned}
\mathrm{TV}(y_T, \widehat{y}_T) &= \frac{1}{2} \int_\mathbb{R} \left| \frac{1}{\sqrt{2\pi}} e^{-\frac{x^2}{2}} \left( 1 + \frac{1}{2} \sin(2n\pi x) \right) - \frac{1}{\sqrt{2\pi}} e^{-\frac{x^2}{2}} \right| \mathrm{d}x \\
&= \frac{1}{4} \int_\mathbb{R} \frac{1}{\sqrt{2\pi}} e^{-\frac{x^2}{2}} |\sin(2n\pi x)| \mathrm{d}x.
\end{aligned} \tag{B.1}
$$

To calculate the last integral, we consider the Fourier expansion

$$
|\sin(2n\pi x)| = \frac{2}{\pi} - \frac{4}{\pi} \sum_{k \geq 1} \frac{\cos 4kn\pi x}{4k^2 - 1}.
$$

For any $k \geq 1$, we have:

$$
\begin{aligned}
\left| \int_\mathbb{R} e^{-\frac{x^2}{2}} \cos(4kn\pi x) \mathrm{d}x \right| &= 2 \left| \int_0^\infty e^{-\frac{x^2}{2}} \cos(4kn\pi x) \mathrm{d}x \right| \\
&\leq 2 \sup_{N \geq 0} \left| \int_0^N \cos(4kn\pi x) \mathrm{d}x \right| \\
&\leq \frac{1}{2kn\pi}.
\end{aligned}
$$

Therefore, we can plug these integrals into B.1 and obtain

$$
\begin{aligned}
\mathrm{TV}(y_T, \widehat{y}_T) &\geq \frac{1}{4} \left[ \frac{2}{\pi} - \frac{4}{\pi} \sum_{k \geq 1} \frac{1}{4k^2 - 1} \left| \int_\mathbb{R} \frac{1}{\sqrt{2\pi}} e^{-\frac{x^2}{2}} \cos(4kn\pi x) \mathrm{d}x \right| \right] \\
&\geq \frac{1}{4} \left[ \frac{2}{\pi} - \frac{4}{\pi} \sum_{k \geq 1} \frac{1}{4k^2 - 1} \frac{1}{\sqrt{2\pi} 2kn\pi} \right] \\
&\geq \frac{1}{4} \left[ \frac{2}{\pi} - \frac{2}{\pi^2 \sqrt{2\pi} n} \sum_{k \geq 1} \frac{1}{3k^3} \right].
\end{aligned}
$$

By taking $n$ large enough, we can easily show that $\mathrm{TV}(y_T, \widehat{y}_T) \geq \frac{1}{4\pi}$. This completes the proof of Theorem 2.1. $\qquad \square$

**Theorem B.1.** Consider the same 1-dimensional OU process $(x_t)_{t \in [0,T]}$ and its reverse process $(y_t)_{t \in [0,T]}$ defined in Theorem 2.1. Recall that we denote the law of $x_t$ by $q_t$. Then for arbitrarily small $\epsilon > 0$ and arbitrarily big $N > 0$, there exists $s_\theta(t, x)$, which is smooth, anywhere $\epsilon$-close to

the true score function, and its divergence is anywhere $\epsilon$-close to the divergence of the true score function, i.e.,

$$\left|\mathbf{s}_\theta(T-t,x) - \nabla \log q(T-t,x)\right| \le \epsilon, \quad \forall t,\, \forall x,$$
$$\left|\nabla \cdot \mathbf{s}_\theta(T-t,x) - \nabla \cdot \nabla \log q(T-t,x)\right| \le \epsilon, \quad \forall t,\, \forall x,$$

such that the law $p_t$ of the corresponding simulated reverse process $(\widehat{y}_t)_{t \in [0,T]}$ (see (2.5)) satisfies

$$\sup |\nabla \log p_T(x)| \ge N,$$

which indicates that the score of the sampling process can be unbounded no matter how small the score estimation error and divergence error is.

*Proof of Theorem B.1.* Substitute the oscillate term $\sin(2n\pi x)$ in the proof of Theorem 2.1 by $\frac{1}{n(x^2+1)}\sin(2n^2\pi x)$. In this case, we still have $q(t,x) = \frac{1}{\sqrt{2\pi}}e^{-\frac{x^2}{2}}$ and $\nabla \log q(t,x) = -x$. Now $\widehat{q}(T-t,x)$ becomes $\frac{1}{\sqrt{2\pi}}e^{-\frac{x^2}{2}}\left(1 + \frac{t}{2T}\frac{1}{n(x^2+1)}\sin(2n^2\pi x)\right)$, and now our estimated score becomes:

$$\mathbf{s}_\theta(T-t,x) = \frac{\int_x^\infty \frac{1}{\sqrt{2\pi}}e^{-\frac{y^2}{2}}\frac{1}{n(x^2+1)}\sin(2n^2\pi x)\left(\frac{1}{2T}\right)\mathrm{d}y}{\widehat{q}(T-t,x)} - x,$$

Without loss of generality, we can assume $x \ge 0$. Then we have:

$$\left|\mathbf{s}_\theta(T-t,x) - \nabla \log q(T-t,x)\right| = \frac{\left|\int_x^\infty \frac{1}{\sqrt{2\pi}}e^{-\frac{y^2}{2}}\frac{1}{n(y^2+1)}\sin(2n^2\pi y)\left(\frac{1}{2T}\right)\mathrm{d}y\right|}{\frac{1}{\sqrt{2\pi}}e^{-\frac{x^2}{2}}\left(1 + \frac{T-t}{2T}\frac{1}{n(y^2+1)}\sin(2n^2\pi y)\right)}$$

$$\le \frac{\frac{1}{\sqrt{2\pi}}e^{-\frac{x^2}{2}}\frac{1}{2T}\frac{1}{(y^2+1)}\sup_{N \ge x}\left|\int_x^N \frac{1}{n}\sin(2n^2\pi y)\mathrm{d}y\right|}{\frac{1}{\sqrt{2\pi}}e^{-\frac{x^2}{2}}\frac{1}{2}}$$

$$\le \frac{1}{T}\frac{1}{n^3\pi}. \tag{B.2}$$

Further more, we consider the divergence estimation error. Since this example is one-dimensional, taking divergence is the same as taking a derivative with respect to $x$. Hence we have:

$$\left|\nabla \cdot \mathbf{s}_\theta(T-t,x) - \nabla \cdot \nabla \log q(T-t,x)\right| = \left|\frac{\partial}{\partial x}\frac{\int_x^\infty \frac{1}{\sqrt{2\pi}}e^{-\frac{y^2}{2}}\frac{1}{n(y^2+1)}\sin(2n^2\pi y)\left(\frac{1}{2T}\right)\mathrm{d}y}{\widehat{q}(T-t,x)}\right|$$

$$\le \left|\frac{\frac{1}{\sqrt{2\pi}}e^{-\frac{x^2}{2}}\frac{1}{n(x^2+1)}\sin(2n^2\pi x)\left(\frac{1}{2T}\right)\cdot\widehat{q}(T-t,x)}{\widehat{q}(T-t,x)^2}\right|$$

$$+\left|\frac{\int_x^\infty \frac{1}{\sqrt{2\pi}}e^{-\frac{y^2}{2}}\frac{1}{n(y^2+1)}\sin(2n^2\pi y)\left(\frac{1}{2T}\right)\mathrm{d}y\cdot\frac{\partial}{\partial x}\widehat{q}(T-t,x)}{\widehat{q}(T-t,x)^2}\right|. \tag{B.3}$$

Firstly, we have

$$\frac{1}{2}\frac{1}{\sqrt{2\pi}}e^{-\frac{x^2}{2}} \le |\widehat{q}(T-t,x)| \le \frac{1}{\sqrt{2\pi}}e^{-\frac{x^2}{2}}.$$

Moreover, we know that

$$\left|\frac{\partial}{\partial x}\widehat{q}(T-t,x)\right| = \left|\frac{1}{\sqrt{2\pi}}e^{-\frac{x^2}{2}}\left(-x\left(1 + \frac{t}{2T}\frac{1}{n(x^2+1)}\sin(2n^2\pi x)\right)\right.\right.$$

$$\left.\left.+\frac{t}{2T}\frac{1}{n}\frac{2n^2\pi\cos(2n^2\pi x)(x^2+1) - 2x\sin(2n^2\pi x)}{(x^2+1)^2}\right)\right| \tag{B.4}$$

$$\le \frac{1}{\sqrt{2\pi}}e^{-\frac{x^2}{2}}\left(|x| + \frac{1}{4Tn} + n\pi + \frac{1}{2Tn}\right)$$

$$\le \frac{1}{\sqrt{2\pi}}e^{-\frac{x^2}{2}}\left(|x| + 2n\pi\right).$$

Since $\frac{1}{\sqrt{2\pi}}e^{-\frac{y^2}{2}}\frac{1}{n(y^2+1)}$ is strictly monotonically decreasing when $y \geq x \geq 0$, we have

$$\left| \int_x^\infty \frac{1}{\sqrt{2\pi}} e^{-\frac{y^2}{2}} \frac{1}{n(y^2+1)} \sin(2n^2\pi y) \left(\frac{1}{2T}\right) dy \right|$$

$$\leq \frac{1}{\sqrt{2\pi}} e^{-\frac{x^2}{2}} \frac{1}{n(x^2+1)} \frac{1}{2T} \sup_{N \geq x} \left| \int_x^N \sin(2n^2\pi y) dy \right|$$

$$\leq \frac{1}{\sqrt{2\pi}} e^{-\frac{x^2}{2}} \frac{1}{x^2+1} \frac{1}{n^3\pi} \frac{1}{2T}.$$

Substituting back to B.3, we have

$$|\nabla \cdot \mathbf{s}_\theta(T-t,x) - \nabla \cdot \nabla \log q(T-t,x)| \leq \frac{1}{2Tn} + 4\frac{1}{x^2+1}\frac{1}{n^3\pi}\frac{1}{2T}(|x|+2n\pi)$$

$$\leq \frac{1}{2Tn} + \frac{1}{Tn^3\pi} + \frac{4}{Tn^2}. \tag{B.5}$$

For any $\epsilon > 0$, using (B.2) and (B.5), for $n$ large enough, the score estimation error and the divergence estimation error are less than $\epsilon$ for every $t$ and $x$. Same as in the proof of Theorem 2.1, using Fokker-Planck equation, we know that $\widehat{q}$ is the law of the sampling process. However, by (B.4), the score of the sampling process is clearly unbounded (there is a coefficient proportional to $n$, and our $n$ is arbitrary large). $\qquad\square$

## C    PROOF OF LEMMA 3.2

*Proof of Lemma 3.2.* We begin by computing total variance distance between $\boldsymbol{X}_t$ and $\boldsymbol{Y}_t$ for any $t \in [0,T]$:

$$\mathrm{TV}(\boldsymbol{X}_t, \boldsymbol{Y}_t) = \int_{\Omega_t} p(t,\mathbf{x}) - q(t,\mathbf{x}) \, d\mathbf{x}, \tag{C.1}$$

where $\Omega_t := \{\mathbf{x} \in \mathbb{R}^d \mid p(t,\mathbf{x}) > q(t,\mathbf{x})\}$. Now we proceed to compute the time-derivative of (C.1). By Theorem K.1, we have

$$\frac{\partial \mathrm{TV}(\boldsymbol{X}_t, \boldsymbol{Y}_t)}{\partial t} = \underbrace{\int_{\Omega_t} \frac{\partial}{\partial t}\big(p(t,\mathbf{x}) - q(t,\mathbf{x})\big) \, d\mathbf{x}}_{I_1} + \underbrace{\int_{\partial\Omega_t} \big(p(t,\mathbf{x}) - q(t,\mathbf{x})\big) \mathbf{v}(t,\mathbf{x}) \cdot \mathbf{n}(t,\mathbf{x}) \, dS}_{I_2}.$$
$$\tag{C.2}$$

For term $I_1$, by Fokker-Planck equation, we have

$$\frac{\partial}{\partial t}\big(p(t,\mathbf{x}) - q(t,\mathbf{x})\big) = -\nabla \cdot \big(\mathbf{b}(t,\mathbf{x})p(t,\mathbf{x}) - \mathbf{b}^*(t,\mathbf{x})q(t,\mathbf{x})\big).$$

Therefore, we have

$$I_1 = \int_{\Omega_t} -\nabla \cdot \big(\mathbf{b}(t,\mathbf{x})p(t,\mathbf{x}) - \mathbf{b}^*(t,\mathbf{x})q(t,\mathbf{x})\big) d\mathbf{x}.$$

For term $I_2$, by the definition of $\Omega_t := \{\mathbf{x} \in \mathbb{R}^d \mid p(t,\mathbf{x}) > q(t,\mathbf{x})\}$, we have $p(t,\mathbf{x}) = q(t,\mathbf{x})$ on $\partial\Omega_t$. Hence $I_2 = 0$.

Therefore, by substituting $I_1$ and $I_2$ into (C.2), we obtain

$$\frac{\partial \mathrm{TV}(\boldsymbol{X}_t, \boldsymbol{Y}_t)}{\partial t} = \int_{\Omega_t} -\nabla \cdot \big(\mathbf{b}(t,\mathbf{x})p(t,\mathbf{x}) - \mathbf{b}^*(t,\mathbf{x})q(t,\mathbf{x})\big) d\mathbf{x}$$

$$= \int_{\partial\Omega_t} -\big(\mathbf{b}(t,\mathbf{x})p(t,\mathbf{x}) - \mathbf{b}^*(t,\mathbf{x})q(t,\mathbf{x})\big) \cdot \mathbf{n}(t,\mathbf{x}) dS$$

$$= \int_{\partial\Omega_t} -\big(\mathbf{b}(t,\mathbf{x})q(t,\mathbf{x}) - \mathbf{b}^*(t,\mathbf{x})q(t,\mathbf{x})\big) \cdot \mathbf{n}(t,\mathbf{x}) dS$$

$$= \int_{\Omega_t} -\nabla \cdot \Big(\big(\mathbf{b}(t,\mathbf{x}) - \mathbf{b}^*(t,\mathbf{x})\big)q(t,\mathbf{x})\Big) d\mathbf{x}$$

$$= -\int_{\Omega_t} \big(\nabla \cdot \mathbf{b}(t, \mathbf{x}) - \nabla \cdot \mathbf{b}^*(t, \mathbf{x})\big) q(t, \mathbf{x}) \, \mathrm{d}\mathbf{x} - \int_{\Omega_t} \big(\mathbf{b}(t, \mathbf{x}) - \mathbf{b}^*(t, \mathbf{x})\big) \cdot \nabla q(t, \mathbf{x}) \, \mathrm{d}\mathbf{x},$$

where in the second equation we use Gauss's theorem to compute the integration of divergence, the third equation uses the fact that $p(t, \mathbf{x}) = q(t, \mathbf{x})$ on $\partial \Omega_t$, the forth equation holds by Gauss's theorem again. This completes the proof. $\qquad\square$

## D    PROOF OF THEOREM 3.4

In this section, we provide a proof of Theorem 3.4. The core of the theorem's proof is Lemma 3.2, which allows us to represent the time-derivative of TV distribution between distribution by score estimation error and divergence estimation error.

*Proof of Theorem 3.4.* Recall $\mathbf{Y}_t$ and $\widehat{\mathbf{Y}}_t$ are ODE flows determined by (3.1) and (3.2). Define $\mathbf{b}(t, \mathbf{x}) = \mathbf{x} + \nabla \log q(T - t, \mathbf{x})$ and $\widehat{\mathbf{b}}(t, \mathbf{x}) = \mathbf{x} + \mathbf{s}_\theta(T - t, \mathbf{x})$, then we have:

$$\mathrm{d}\mathbf{Y}_t = \mathbf{b}(t, \mathbf{Y}_t)\mathrm{d}t, \quad \mathbf{Y}_0 \sim q_T.$$
$$\mathrm{d}\widehat{\mathbf{Y}}_t = \widehat{\mathbf{b}}(t, \widehat{\mathbf{Y}}_t)\mathrm{d}t, \quad \widehat{\mathbf{Y}}_0 \sim \pi_d.$$

By Lemma 3.2, recall that the law of $\mathbf{Y}_t$ is denoted by $q(T - t, \mathbf{x})$, we have:

$$
\begin{aligned}
\frac{\partial \mathrm{TV}(\widehat{\mathbf{Y}}_t, \mathbf{Y}_t)}{\partial t} &= -\int_{\Omega_t} \Big(\nabla \cdot \widehat{\mathbf{b}}(t, \mathbf{x}) - \nabla \cdot \mathbf{b}(t, \mathbf{x})\Big) q(T - t, \mathbf{x}) \, \mathrm{d}\mathbf{x} \\
&\quad - \int_{\Omega_t} \Big(\widehat{\mathbf{b}}(t, \mathbf{x}) - \mathbf{b}(t, \mathbf{x})\Big) \cdot \nabla \log q(T - t, \mathbf{x}) q(T - t, \mathbf{x}) \mathrm{d}\mathbf{x} \\
&= \underbrace{\int_{\Omega_t} \Big(\nabla \cdot \mathbf{s}_\theta(T - t, \mathbf{x}) - \nabla \cdot \nabla \log q(T - t, \mathbf{x})\Big) q(T - t, \mathbf{x}) \mathrm{d}\mathbf{x}}_{I_1} \\
&\quad - \underbrace{\int_{\Omega_t} \big(\mathbf{s}_\theta(T - t, \mathbf{x}) - \nabla \log q(T - t, \mathbf{x})\big) \cdot \nabla \log q(T - t, \mathbf{x}) q(T - t, \mathbf{x}) \mathrm{d}\mathbf{x}}_{I_2}.
\end{aligned}
\tag{D.1}
$$

For $I_1$, we know that:

$$
\begin{aligned}
|I_1| &\le \int_{\Omega_t} \Big|\nabla \cdot \mathbf{s}_\theta(T - t, \mathbf{x}) - \nabla \cdot \nabla \log q(T - t, \mathbf{x})\Big| q(T - t, \mathbf{x}) \mathrm{d}\mathbf{x} \\
&\le \mathbb{E}\Big|\nabla \cdot \mathbf{s}_\theta(T - t, \mathbf{Y}_t) - \nabla \cdot \nabla \log q(T - t, \mathbf{Y}_t)\Big|.
\end{aligned}
\tag{D.2}
$$

For $I_2$, using the Cauchy-Schwartz inequality, we know that:

$$
\begin{aligned}
|I_2| &\le \sqrt{\int_{\Omega_t} \big\|\mathbf{s}_\theta(T - t, \mathbf{x}) - \nabla \log q(T - t, \mathbf{x})\big\|^2 q(T - t, \mathbf{x}) \mathrm{d}\mathbf{x}} \\
&\quad \cdot \sqrt{\int_{\Omega_t} \big\|\nabla \log q(T - t, \mathbf{x})\big\|^2 q(T - t, \mathbf{x}) \mathrm{d}\mathbf{x}} \\
&\le \sqrt{\mathbb{E}\big\|\mathbf{s}_\theta(T - t, \mathbf{Y}_t) - \nabla \log q(T - t, \mathbf{Y}_t)\big\|^2} \cdot \sqrt{\mathbb{E}\big\|\nabla \log q(T - t, \mathbf{Y}_t)\big\|^2}.
\end{aligned}
\tag{D.3}
$$

Substituting (D.2) and (D.3) back to (D.1), and taking the integral with respect to $t$ on $[0, T - \delta]$, we have:

$$
\begin{aligned}
\mathrm{TV}(\widehat{\mathbf{Y}}_{T-\delta}, \mathbf{X}_\delta) &\le \mathrm{TV}(\widehat{\mathbf{Y}}_0, \mathbf{Y}_0) + \int_0^{T-\delta} \big(|I_1| + |I_2|\big) \mathrm{d}t \\
&\le \mathrm{TV}(\widehat{\mathbf{Y}}_0, \mathbf{Y}_0) + \int_0^{T-\delta} \mathbb{E}\Big|\nabla \cdot \mathbf{s}_\theta(T - t, \mathbf{Y}_t) - \nabla \cdot \nabla \log q(T - t, \mathbf{Y}_t)\Big| \mathrm{d}t \\
&\quad + \int_0^{T-\delta} \sqrt{\mathbb{E}\big\|\mathbf{s}_\theta(T - t, \mathbf{Y}_t) - \nabla \log q(T - t, \mathbf{Y}_t)\big\|^2} \cdot \sqrt{\mathbb{E}\big\|\nabla \log q(T - t, \mathbf{Y}_t)\big\|^2} \mathrm{d}t
\end{aligned}
$$

$$\leq \mathrm{TV}(\widehat{\boldsymbol{Y}}_0, \boldsymbol{Y}_0) + \int_0^{T-\delta} \mathbb{E}\Big|\nabla \cdot \mathbf{s}_\theta(T-t, \boldsymbol{Y}_t) - \nabla \cdot \nabla \log q(T-t, \boldsymbol{Y}_t)\Big| \mathrm{d}t$$

$$+ \sqrt{\int_0^{T-\delta} \mathbb{E}\big\|\mathbf{s}_\theta(T-t, \boldsymbol{Y}_t) - \nabla \log q(T-t, \boldsymbol{Y}_t)\big\|^2 \mathrm{d}t} \cdot \sqrt{\int_0^{T-\delta} \mathbb{E}\big\|\nabla \log q(T-t, \boldsymbol{Y}_t)\big\|^2 \mathrm{d}t}.$$

$$(\mathrm{D.4})$$

Since $\boldsymbol{X}_0$ has finite second-order momentum, using Lemma K.2, we have that:

$$\mathbb{E}_Q\big\|\nabla \log q(T-t, \boldsymbol{Y}_t)\big\|^2 \leq d\big(1 - \mathrm{e}^{-2(T-t)}\big)^{-1}.$$

Thus we know that:

$$\begin{aligned}
\int_0^{T-\delta} \mathbb{E}\big\|\nabla \log q(T-t, \boldsymbol{Y}_t)\big\|^2 \mathrm{d}t &\leq \int_0^{T-\delta} \frac{d}{1 - \mathrm{e}^{-2(T-t)}} \mathrm{d}t \\
&= d \int_\delta^T \frac{1}{1 - \mathrm{e}^{-2t}} \mathrm{d}t \\
&= d\Big(\frac{1}{2} \log\big(\mathrm{e}^{2T} - 1\big) - \frac{1}{2} \log\big(\mathrm{e}^{2\delta} - 1\big)\Big) \\
&= O\Big(dT + d\log\frac{1}{\delta}\Big).
\end{aligned}$$

$$(\mathrm{D.5})$$

Substituting (D.5) into (D.4) and using Assumptions 3.1 and 3.3, we can conclude with

$$\mathrm{TV}(\widehat{\boldsymbol{Y}}_{T-\delta}, \boldsymbol{X}_\delta) \lesssim \mathrm{TV}(\widehat{\boldsymbol{Y}}_0, \boldsymbol{Y}_0) + \sqrt{dT + d\log\frac{1}{\delta}}\epsilon_{\mathrm{score}} + \epsilon_{\mathrm{div}}.$$

$$\qquad\qquad\qquad\qquad\qquad\qquad\qquad\qquad\qquad\qquad\qquad\qquad\qquad\qquad\qquad\quad \square$$

## E  PROOF OF THEOREM 4.3

*Proof of Theorem 4.3.* In the time step $t \in [t_k, t_{k+1}]$, we can compute the time derivative of the total variation distance between $\widehat{Y}_t$ and $\boldsymbol{Y}_t$ using Lemma 3.2:

$$\frac{\partial \mathrm{TV}(\widehat{\boldsymbol{Y}}_t, \boldsymbol{Y}_t)}{\partial t} = -\underbrace{\int_{\Omega_t} \big(\nabla \cdot \widehat{\mathbf{b}}(t, \mathbf{x}) - \nabla \cdot \mathbf{b}(t, \mathbf{x})\big) q(T-t, \mathbf{x}) \mathrm{d}\mathbf{x}}_{I_1}$$

$$-\underbrace{\int_{\Omega_t} \big(\widehat{\mathbf{b}}(t, \mathbf{x}) - \mathbf{b}(t, \mathbf{x})\big) \cdot \nabla q(T-t, \mathbf{x}) \mathrm{d}\mathbf{x}}_{I_2}. \qquad (\mathrm{E.1})$$

Recall that $\mathbf{b}(t, \mathbf{x}) = -\big(\mathbf{f}(T-t, \mathbf{x}) - \frac{1}{2}g(T-t)^2 \nabla \log q_{T-t}(\mathbf{x})\big)$, $\widehat{\mathbf{b}}(t, \mathbf{x}) = \frac{\partial F}{\partial t}\big(t, F_t^{-1}(\mathbf{x})\big)$.

For $I_1$, we decompose $\mathrm{tr}[\nabla\widehat{\mathbf{b}}(t, \mathbf{x}) - \nabla\mathbf{b}(t, \mathbf{x})]$ as follows:

$$\begin{aligned}
\mathrm{tr}[\nabla\widehat{\mathbf{b}}(t, \mathbf{x}) - \nabla\mathbf{b}(t, \mathbf{x})] &= \Big[\nabla\Big[\frac{\partial}{\partial t}F\Big]\big(t, F_t^{-1}(\mathbf{x})\big)\nabla F_t^{-1}(\mathbf{x}) + \nabla\mathbf{f}(T-t, \mathbf{x}) \\
&\quad - \frac{1}{2}g(T-t)^2 \nabla^2 \log q_{T-t_k}\big(F_t^{-1}(\mathbf{x})\big)\Big] \\
&\quad - \frac{1}{2}g(T-t)^2 \Big[\nabla^2 \log q_{T-t}(\mathbf{x}) - \nabla^2 \log q_{T-t_k}\big(F_t^{-1}(\mathbf{x})\big)\Big] \\
&= \Psi_k(t, \mathbf{z}) + \Big(\nabla\Big[\frac{\partial}{\partial t}F\Big](t, \mathbf{z}) + \nabla_{\mathbf{z}}\big[\mathbf{f}\big(T-t, F_t(\mathbf{z})\big)\big]\Big)\big(\nabla F_t^{-1}(\mathbf{x}) - \mathbf{I}\big) \\
&\quad - \frac{1}{2}g(T-t)^2 \Big[\nabla^2 \log q_{T-t}(\mathbf{x}) - \nabla^2 \log q_{T-t_k}(\mathbf{z})\Big],
\end{aligned}$$

where $\mathbf{z} = F_t^{-1}(\mathbf{x})$. Therefore, we have

$$|I_1| \leq \int_{\Omega_t} \Big|\mathrm{tr}\big[\nabla\widehat{\mathbf{b}}(t, \mathbf{x}) - \nabla\mathbf{b}(t, \mathbf{x})\big]\Big| q(T-t, \mathbf{x}) \mathrm{d}\mathbf{x}$$

$$\leq \int |\operatorname{tr}\Psi_k(t,\mathbf{z})|q(T-t,\mathbf{x})\mathrm{d}\mathbf{x}$$

$$+ \int \left|\operatorname{tr}\left[\nabla\left[\frac{\partial}{\partial t}F\right](t,\mathbf{z}) + \nabla_{\mathbf{z}}\left[\mathbf{f}(T-t,F_t(\mathbf{z}))\right]\right](\nabla F_t^{-1}(\mathbf{x}) - \mathbf{I})\right|q(T-t,\mathbf{x})\mathrm{d}\mathbf{x}$$

$$+ \int \frac{1}{2}g(T-t)^2\left|\operatorname{tr}\left[\nabla^2\log q_{T-t}(\mathbf{x}) - \nabla^2\log q_{T-t_k}(\mathbf{z})\right]\right|q(T-t,\mathbf{x})\mathrm{d}\mathbf{x}$$

$$\overset{(i)}{\leq} \left(\int \left[\operatorname{tr}\Psi_k(t,\mathbf{z})\right]^2 p_{F_t(\mathbf{Y}_{t_k})}(\mathbf{x})\mathrm{d}\mathbf{x}\right)^{\frac{1}{2}} \left(\int \left(\frac{p_{\mathbf{Y}_t}(\mathbf{x})}{p_{F_t(\mathbf{Y}_{t_k})}(\mathbf{x})}\right)^2 p_{F_t(\mathbf{Y}_{t_k})}(\mathbf{x})\mathrm{d}\mathbf{x}\right)^{\frac{1}{2}}$$

$$+ \max_{\mathbf{x}}\left|\operatorname{tr}\left[\nabla\left[\frac{\partial}{\partial t}F\right](t,F_t^{-1}(\mathbf{x})) + \nabla_{\mathbf{z}}\left[\mathbf{f}(T-t,F_t(\mathbf{z}))\right]\right](\nabla F_t^{-1}(\mathbf{x}) - \mathbf{I})\right|$$

$$+ \int \frac{1}{2}g(T-t)^2\left|\operatorname{tr}\left[\nabla^2\log q_{T-t}(\mathbf{x}) - \nabla^2\log q_{T-t_k}(\mathbf{z})\right]\right|q(T-t,\mathbf{x})\mathrm{d}\mathbf{x}$$

$$\overset{(ii)}{=} \left(\mathbb{E}\left[\left(\operatorname{tr}\Psi_k(t,\mathbf{Y}_{t_k})\right)^2\right]\right)^{\frac{1}{2}} \left(\int \left(\frac{p_{\mathbf{Y}_t}(\mathbf{x})}{p_{F_t(\mathbf{Y}_{t_k})}(\mathbf{x})}\right)^2 p_{F_t(\mathbf{Y}_{t_k})}(\mathbf{x})\mathrm{d}\mathbf{x}\right)^{\frac{1}{2}}$$

$$+ \max_{\mathbf{x}}\left|\operatorname{tr}\left[\nabla\left[\frac{\partial}{\partial t}F\right](t,F_t^{-1}(\mathbf{x})) + \nabla_{\mathbf{z}}\left[\mathbf{f}(T-t,F_t(\mathbf{z}))\right]\right](\nabla F_t^{-1}(\mathbf{x}) - \mathbf{I})\right|$$

$$+ \int \frac{1}{2}g(T-t)^2\left|\operatorname{tr}\left[\nabla^2\log q_{T-t}(\mathbf{x}) - \nabla^2\log q_{T-t_k}(\mathbf{z})\right]\right|q(T-t,\mathbf{x})\mathrm{d}\mathbf{x}. \tag{E.2}$$

where $(i)$ holds due to the Cauchy-Schwarz inequality. $(ii)$ holds due to the Jacobian transformation $p_{F_t(\mathbf{Y}_{t_k})}(\mathbf{x})\mathrm{d}\mathbf{x} = p_{\mathbf{Y}_{t_k}}(\mathbf{z})\mathrm{d}\mathbf{z}$ for $\mathbf{z} = F_t^{-1}(\mathbf{x})$.

For $I_2$, we decompose $\widehat{\mathbf{b}}(t,\mathbf{x}) - \mathbf{b}(t,\mathbf{x})$ as follows:

$$\widehat{\mathbf{b}}(t,\mathbf{x}) - \mathbf{b}(t,\mathbf{x}) = \left[\frac{\partial F}{\partial t}\left(t,F_t^{-1}(\mathbf{x})\right) + \mathbf{f}(T-t,\mathbf{x}) - \frac{1}{2}g(T-t)^2\nabla\log q_{T-t_k}\left(F_t^{-1}(\mathbf{x})\right)\right]$$

$$- \frac{1}{2}g(T-t)^2\left[\nabla\log q_{T-t}(\mathbf{x}) - \nabla\log q_{T-t_k}\left(F_t^{-1}(\mathbf{x})\right)\right]$$

$$= \left[\frac{\partial F}{\partial t}(t,\mathbf{z}) + \mathbf{f}\left(T-t,F_t(\mathbf{z})\right) - \frac{1}{2}g(T-t)^2\nabla\log q_{T-t_k}(\mathbf{z})\right]$$

$$- \frac{1}{2}g(T-t)^2\left[\nabla\log q_{T-t}(\mathbf{x}) - \nabla\log q_{T-t_k}\left(F_t^{-1}(\mathbf{x})\right)\right],$$

$$= \Phi_k(t,\mathbf{z}) - \frac{1}{2}g(T-t)^2\left[\nabla\log q_{T-t}(\mathbf{x}) - \nabla\log q_{T-t_k}\left(F_t^{-1}(\mathbf{x})\right)\right],$$

where $\mathbf{z} = F_t^{-1}(\mathbf{x})$. Therefore, we have

$$|I_2| \leq \int_{\Omega_t} \left|\Phi_k(t,\mathbf{z})\cdot\nabla q(T-t,\mathbf{x})\right|\mathrm{d}\mathbf{x}$$

$$+ \frac{1}{2}g(T-t)^2\int_{\Omega_t}\left|\left[\nabla\log q_{T-t}(\mathbf{x}) - \nabla\log q_{T-t_k}\left(F_t^{-1}(\mathbf{x})\right)\right]\cdot\nabla q(T-t,\mathbf{x})\right|\mathrm{d}\mathbf{x}$$

$$\overset{(i)}{\leq} \left(\int \|\Phi_k(t,\mathbf{z})\|^2 p_{F_t(\mathbf{Y}_{t_k})}(\mathbf{x})\mathrm{d}\mathbf{x}\right)^{\frac{1}{2}} \left(\int \left\|\frac{\nabla\log q(T-t,\mathbf{x})p_{\mathbf{Y}_t}(\mathbf{x})}{p_{F_t(\mathbf{Y}_{t_k})}(\mathbf{x})}\right\|^2 p_{F_t(\mathbf{Y}_{t_k})}(\mathbf{x})\mathrm{d}\mathbf{x}\right)^{\frac{1}{2}}$$

$$+ \frac{1}{2}g(T-t)^2\int_{\Omega_t}\left|\left[\nabla\log q_{T-t}(\mathbf{x}) - \nabla\log q_{T-t_k}\left(F_t^{-1}(\mathbf{x})\right)\right]\cdot\nabla q(T-t,\mathbf{x})\right|\mathrm{d}\mathbf{x}$$

$$\overset{(ii)}{\leq} \left(\mathbb{E}\left[\|\Phi_k(t,\mathbf{Y}_{t_k})\|^2\right]\right)^{\frac{1}{2}} \left(\int \left\|\frac{\nabla\log q(T-t,\mathbf{x})p_{\mathbf{Y}_t}(\mathbf{x})}{p_{F_t(\mathbf{Y}_{t_k})}(\mathbf{x})}\right\|^2 p_{F_t(\mathbf{Y}_{t_k})}(\mathbf{x})\mathrm{d}\mathbf{x}\right)^{\frac{1}{2}}$$

$$+ \frac{1}{2}g(T-t)^2\int_{\Omega_t}\left|\left[\nabla\log q_{T-t}(\mathbf{x}) - \nabla\log q_{T-t_k}\left(F_t^{-1}(\mathbf{x})\right)\right]\cdot\nabla q(T-t,\mathbf{x})\right|\mathrm{d}\mathbf{x}. \tag{E.3}$$

where $(i)$ holds due to the Cauchy-Schwarz inequality. $(ii)$ holds due to the Jacobian transformation $p_{F_t(\mathbf{Y}_{t_k})}(\mathbf{x})\mathrm{d}\mathbf{x} = p_{\mathbf{Y}_{t_k}}(\mathbf{z})\mathrm{d}\mathbf{z}$ for $\mathbf{z} = F_t^{-1}(\mathbf{x})$. Substituting (E.2) and (E.3) into (E.1) and taking the integral over $t$, we can complete the proof of Theorem 4.3. $\qquad\square$

## F ANALYSIS OF VP+EI

In this section, we focus on the case where $f(\mathbf{x}) = -\mathbf{x}$ and $g(t) = \sqrt{2}$, i.e. the VP-SDE. Specifically, the forward process $\boldsymbol{X}_t$ satisfies:

$$\mathrm{d}\boldsymbol{X}_t = -\boldsymbol{X}_t\mathrm{d}t + \sqrt{2}\mathrm{d}\boldsymbol{W}_t, \quad \boldsymbol{X}_0 \sim q_0, \tag{F.1}$$

and the true reverse process $\boldsymbol{Y}_t$ satisfies:

$$\mathrm{d}\boldsymbol{Y}_t = \Big(\boldsymbol{Y}_t + \nabla \log q_{T-t}(\boldsymbol{Y}_t)\Big)\mathrm{d}t, \quad \boldsymbol{Y}_0 \sim q_T.$$

Then, the forward process has the following closed-form expression:

$$\boldsymbol{Y}_t = \boldsymbol{X}_{T-t} = \mathrm{e}^{-(T-t)}\boldsymbol{X}_0 + \sqrt{1 - \mathrm{e}^{-2(T-t)}}\boldsymbol{Z}, \quad \boldsymbol{Z} \sim N(0, \mathbf{I}_d). \tag{F.2}$$

Recall the exponential integrator provides a discretized version of the reverse process:

$$\widehat{\boldsymbol{Y}}_{t_{k+1}} = \mathrm{e}^{\eta_k}\widehat{\boldsymbol{Y}}_{t_k} + \frac{1}{2}\big(\mathrm{e}^{\eta_k} - 1\big)\frac{1}{2}\mathbf{s}_\theta(T - t_k, \widehat{\boldsymbol{Y}}_{t_k}).$$

Therefore, we can define the following interpolation operator:

$$F_t(\mathbf{z}) = \mathrm{e}^{t-t_k}\mathbf{z} + \big(\mathrm{e}^{t-t_k} - 1\big)\mathbf{s}_\theta(T - t_k, \mathbf{z}), \tag{F.3}$$

satisfying $F_{t_{k+1}}(\widehat{\boldsymbol{Y}}_{t_k}) = \widehat{\boldsymbol{Y}}_{t_{k+1}}$. Under Assumption 5.4, we know $\mathbf{s}_\theta(t, \cdot)$ is Lipschitz. Therefore, suppose $\eta \le 1/L$, then we have $1/2 \le \|\nabla F_t\| \le 3/2$. In particular, $F_t$ is invertible.

Denote $\mathrm{e}^{t-t_k}$ by $a$. Since $t - t_k \le \eta \le 1$, we have $a \le 1 + 2(t - t_k)$. Then we know that:

$$\begin{aligned}
\|\mathbf{z}\| &\ge a\|F_t^{-1}(\mathbf{z})\| - (a-1)\Big(L\|F_t^{-1}(\mathbf{z})\| + c\Big) \\
&\ge \|F_t^{-1}(\mathbf{z})\| - 2(t - t_k)\Big(L\|F_t^{-1}(\mathbf{z})\| + c\Big) \\
&\ge \frac{1}{2}\|F_t^{-1}(\mathbf{z})\| - \frac{1}{2},
\end{aligned}$$

where the last inequality holds when we assume that $t - t_k \le \eta \le \min\{1/4L, 1/4c\}$. Thus we know that:

$$\|F_t^{-1}(\mathbf{z})\| \le 2\|\mathbf{z}\| + 1. \tag{F.4}$$

Before we start the proof, we introduce several important lemmas used in our proof. Let $c_1 = (1 + d/2)4L^2$ and $c_2 = (1 + d/2)(L + c)^2$. Using our assumption on $\eta$, we know that

$$\eta \le \min\left\{\frac{1}{16d}, \frac{1}{2c_1 d}, \frac{1}{c_2}, \frac{1}{4R^2 c_1}\right\}. \tag{F.5}$$

The following two lemmas are related to the ratios $p_{\boldsymbol{Y}_t}(\mathbf{x})/p_{F_t(\boldsymbol{Y}_{t_k})}$.

**Lemma F.1.** Let $\boldsymbol{Y}_t$ and $F_t(\mathbf{z})$ be defined in (F.2) and (F.3). Under Assumptions 5.4 and 5.1, suppose our time schedule satisfies (2.6). Then, for $\eta \le \min\{\frac{1}{2(L+1)d}, \frac{1}{16d}, \frac{1}{c_1 d}, \frac{1}{2c_1}, \frac{1}{c_2}, \frac{1}{R^2 c_1}\}$, we have

$$\frac{p_{\boldsymbol{Y}_t}(\mathbf{x})}{p_{F_t(\boldsymbol{Y}_{t_k})}(\mathbf{x})} \lesssim \mathrm{e}^{(t-t_k)c_1\|\mathbf{x}\|^2 + (t-t_k)c_2}.$$

Moreover, we have

$$\int_{\Omega_t} \left(\frac{p_{\boldsymbol{Y}_t}(\mathbf{x})}{p_{F_t(\boldsymbol{Y}_{t_k})}(\mathbf{x})}\right)^2 p_{F_t(\boldsymbol{Y}_{t_k})}(\mathbf{x})\mathrm{d}\mathbf{x} \lesssim 1.$$

**Lemma F.2.** Let $\boldsymbol{Y}_t$ and $F_t(\mathbf{z})$ be defined in (F.2) and (F.3). Under Assumptions 5.1 and 5.4, suppose our time schedule satisfies (2.6). Then for $\eta \le \min\{\frac{1}{2(L+1)d}, \frac{1}{16d}, \frac{1}{2c_1 d}, \frac{1}{2c_1}, \frac{1}{c_2}, \frac{1}{4R^2 c_1}\}$, we have:

$$\int \left\|\frac{\nabla q(T-t, \mathbf{x})}{p_{F_t(\boldsymbol{Y}_{t_k})}(\mathbf{x})}\right\|^2 p_{F_t(\boldsymbol{Y}_{t_k})}(\mathbf{x})\mathrm{d}\mathbf{x} \lesssim \frac{d}{\min\{T-t, 1\}}.$$

The following two lemmas show the upper bound of the derivatives of the true score function with respect to $t$ and $\mathbf{x}$ separately.

**Lemma F.3.** When $X_t$ is defined as in (F.1), under Assumption 5.1, we have:

$$\left| \frac{\partial}{\partial t} \Big( \operatorname{tr} \big( \nabla^2 \log q(t, \mathbf{x}) \big) \Big) \right| \lesssim \frac{d}{\min\{t, 1\}^2} + \frac{R^2 (\|\mathbf{x}\| + R)^2}{\min\{t, 1\}^4},$$

$$\left\| \frac{\partial}{\partial t} \Big( \nabla \log q(t, \mathbf{x}) \Big) \right\| \lesssim \frac{\|\mathbf{x}\| + R}{\min\{t, 1\}^2} + \frac{(\|\mathbf{x}\| + R)^2 R^2}{\min\{t, 1\}^2} + \frac{(\|\mathbf{x}\| + R)^3}{\min\{t, 1\}^3}.$$

**Lemma F.4.** When $X_t$ is defined as in (F.1), under Assumption 5.1, we have:

$$\|\nabla \log q(t, \mathbf{x})\| \lesssim \frac{\|\mathbf{x}\| + R}{\min\{t, 1\}},$$

$$\big\| \nabla^2 \log q(t, \mathbf{x}) \big\|_2 \leq \frac{1}{\min\{t, 1\}} + \frac{R^2}{\min\{t, 1\}^2},$$

$$\big\| \nabla \operatorname{tr} \big( \nabla^2 \log q(t, \mathbf{x}) \big) \big\| \leq \frac{R^3}{\min\{t, 1\}^3}.$$

The proofs of the above lemmas can be found in later sections.

### F.1 PROOF OF THEOREM 5.5

*Proof of Theorem 5.5.* From the discussion in Section 5.1, we know that:

$$\Phi_k(t, \mathbf{x}) = s_\theta(T - t_k, \mathbf{x}) - \nabla \log q_{T-t_k}(\mathbf{x}), \tag{F.6}$$

$$\Psi_k(t, \mathbf{x}) = \nabla \mathbf{s}_\theta(T - t_k, \mathbf{x}) - \nabla^2 \log q_{T-t_k}(\mathbf{x}), \tag{F.7}$$

in this specific case. Recall that $g(t) = \sqrt{2}$ and $\mathbf{f}(T - t, \mathbf{x}) = -\mathbf{x} = -F_t(\mathbf{z})$. From (F.3) we have:

$$\frac{\partial}{\partial t} F(t, \mathbf{x}) = F_t(\mathbf{x}) + \mathbf{s}_\theta(T - t_k, \mathbf{x}).$$

Thus we know that

$$\nabla \Big[ \frac{\partial}{\partial t} F \Big] \big( t, F_t^{-1}(\mathbf{x}) \big) = \big[ \nabla F_t \big](\mathbf{z}) + \nabla \mathbf{s}_\theta(T - t_k, \mathbf{z}). \tag{F.8}$$

By Theorem 4.3, we know that:

$$\begin{aligned}
\mathrm{TV}(\boldsymbol{Y}_{T-\delta}, \widehat{\boldsymbol{Y}}_{t_N}) &\leq \mathrm{TV}(q_T, \pi_d) \\
&+ \sum_{k=0}^{N-1} \int_{t_k}^{t_{k+1}} \bigg[ \underbrace{\sqrt{\mathbb{E}\big[\|\Phi_k(t, \boldsymbol{Y}_{t_k})\|^2\big]} \sqrt{\int \Big\| \frac{\nabla \log q(T - t, \mathbf{x}) p_{\boldsymbol{Y}_t}(\mathbf{x})}{p_{F_t(\boldsymbol{Y}_{t_k})}(\mathbf{x})} \Big\|^2 p_{F_t(\boldsymbol{Y}_{t_k})}(\mathbf{x}) \mathrm{d}\mathbf{x}}}_{J_1:\ \text{Score estimation error}} \\
&+ \underbrace{\sqrt{\mathbb{E}\big[\operatorname{tr}\big(\Psi_k(t, \boldsymbol{Y}_{t_k})\big)^2\big]} \sqrt{\int \Big( \frac{p_{\boldsymbol{Y}_t}(\mathbf{x})}{p_{F_t(\boldsymbol{Y}_{t_k})}(\mathbf{x})} \Big)^2 p_{F_t(\boldsymbol{Y}_{t_k})}(\mathbf{x}) \mathrm{d}\mathbf{x}}}_{J_2:\ \text{Divergence estimation error}} \\
&+ \underbrace{\int \big|\big(\nabla \log q_{T-t_k}(\mathbf{z}) - \nabla \log q_{T-t}(\mathbf{x})\big) \cdot \nabla \log q(T - t, \mathbf{x})\big| q(T - t, \mathbf{x}) \mathrm{d}\mathbf{x}}_{J_3:\ \text{Score discretization error}} \\
&+ \underbrace{\int \big|\operatorname{tr}\big(\nabla^2 \log q(T - t_k, \mathbf{z}) - \nabla^2 \log q(T - t, \mathbf{x})\big)\big| q(T - t, \mathbf{x}) \mathrm{d}\mathbf{x}}_{J_4:\ \text{Divergence discretization error}} \\
&+ \underbrace{\max_{\mathbf{x}} \Big| \operatorname{tr}\Big[ \nabla \mathbf{s}_\theta(T - t_k, \mathbf{z}) \Big] \big( \nabla F_t^{-1}(\mathbf{x}) - \mathbf{I} \big) \Big|}_{J_5:\ \text{Bias error}} \bigg] \mathrm{d}t. 
\end{aligned} \tag{F.9}$$

**Bounding the Score Estimation Error $J_1$.** We can easily verify that our $\eta$ is small enough to satisfies the condition of Lemma F.2. Therefore, using Lemma F.2 and (F.6), we have=

$$|J_1| \lesssim \sqrt{\frac{d}{\min\{T-t, 1\}}} \sqrt{\mathbb{E}_Q \left\| \mathbf{s}_\theta(T - t_k, \boldsymbol{Y}_{t_k}) - \nabla \log q_{T-t_k}(\boldsymbol{Y}_{t_k}) \right\|^2}. \qquad (\text{F.10})$$

**Bounding the Divergence Estimation Error $J_2$.**
Similarly, our $\eta$ is small enough to satisfies the condition of Lemma F.1. Using Lemma F.1 and (F.7), we have

$$|J_2| \lesssim \sqrt{\mathbb{E}_Q \operatorname{tr} \left( \nabla \mathbf{s}_\theta(T - t_k, \boldsymbol{Y}_{t_k}) - \nabla^2 \log q(T - t_k, \boldsymbol{Y}_{t_k}) \right)^2}. \qquad (\text{F.11})$$

**Bounding the Bias Term $J_5$.** From (F.3), we know that:

$$\nabla F_t^{-1}(\mathbf{x}) = \left[ (\nabla F_t)\left( F_t^{-1}(\mathbf{x}) \right) \right]^{-1}$$
$$= \left[ \mathrm{e}^{t-t_k} \mathbf{I}_d + \left( \mathrm{e}^{t-t_k} - 1 \right) \nabla \mathbf{s}_\theta \left( T - t_k, F_t^{-1}(\mathbf{x}) \right) \right]^{-1}.$$

Using Assumption 5.4 and $t - t_k \leq \eta \leq 1$, we have:

$$\| (\nabla F_t)\left( F_t^{-1}(\mathbf{x}) \right) - \mathbf{I}_d \|_2 \leq \left( \mathrm{e}^{t-t_k} - 1 \right) \| \mathbf{I}_d + \nabla \mathbf{s}_\theta \left( T - t_k, F_t^{-1}(\mathbf{x}) \right) \|_2$$
$$\leq 2(t - t_k)(1 + L).$$

Let $\mathbf{A} = (\nabla F_t)\left( F_t^{-1}(\mathbf{x}) \right) - \mathbf{I}_d$. Then we have $\|\mathbf{A}\|_2 \leq 2(t - t_k)(1 + L)$. Thus we know that:

$$\| \nabla \left( F_t^{-1}(\mathbf{x}) \right) - \mathbf{I}_d \|_2 = \| (\mathbf{I}_d + \mathbf{A})^{-1} - \mathbf{I}_d \|_2$$
$$\overset{(i)}{=} \left\| \sum_{n=1}^{\infty} (-1)^n \mathbf{A}^n \right\|_2$$
$$\overset{(ii)}{\leq} \sum_{n=1}^{\infty} \|\mathbf{A}\|_2^n$$
$$\overset{(iii)}{\leq} 4(t - t_k)(1 + L), \qquad (\text{F.12})$$

here $(i)$ holds because of the series expansion for $\|\mathbf{A}\|_2 < 1$. $(ii)$ holds due to the Cauchy-Schwarz inequality. $(iii)$ holds due to our assumption $\eta \leq 1/4(1 + L)$. Then we know that:

$$|J_5| = \max_{\mathbf{x}} \left| \operatorname{tr} \left[ \nabla \mathbf{s}_\theta(T - t_k, \mathbf{z}) \left( \nabla \left( F_t^{-1}(\mathbf{x}) \right) - \mathbf{I}_d \right) \right] \right|$$
$$\overset{(i)}{\leq} d \max \left\| \nabla \mathbf{s}_\theta(T - t_k, \mathbf{z}) \left( \nabla \left( F_t^{-1}(\mathbf{x}) \right) - \mathbf{I}_d \right) \right\|_2$$
$$\leq d \max \| \nabla \mathbf{s}_\theta(T - t_k, \mathbf{z}) \|_2 \| \nabla \left( F_t^{-1}(\mathbf{x}) \right) - \mathbf{I}_d \|_2$$
$$\overset{(ii)}{\leq} 4dL(t - t_k)(1 + L),$$

where $(i)$ holds due to $\operatorname{tr}(\mathbf{A}) \leq d\|\mathbf{A}\|_2$. $(ii)$ holds due to Assumption 5.4 and (F.12). Thus we have:

$$|J_5| \lesssim (t - t_k)dL^2. \qquad (\text{F.13})$$

**Bounding the Moments of $Y_t$.**
Before we start estimating $J_3$ and $J_4$, we first do some preparation. Since we have $\boldsymbol{Y}_t = \boldsymbol{X}_{T-t} = \mathrm{e}^{-(T-t)} \boldsymbol{X}_0 + \sqrt{1 - \mathrm{e}^{-2(T-t)}} \boldsymbol{Z}$, here $\boldsymbol{Z} \sim N(0, \mathbf{I}_d)$. Our goal is to bound the moments of $\boldsymbol{Y}_t$. When $T - t < 1$, we know that $\mathrm{e}^{-(T-t)} = \Theta(1)$ and $\sqrt{1 - \mathrm{e}^{-2(T-t)}} = O(\sqrt{T-t})$. Thus we have:

$$\mathbb{E}\|\boldsymbol{Y}_t\|^2 = \mathrm{e}^{-2(T-t)}\mathbb{E}\|\boldsymbol{X}_0\|^2 + \left(1 - \mathrm{e}^{-2(T-t)}\right)\mathbb{E}\|\boldsymbol{Z}\|^2 + \mathbb{E}\mathrm{e}^{-(T-t)}\sqrt{1 - \mathrm{e}^{-2(T-t)}}\langle \boldsymbol{X}_0, \boldsymbol{Z} \rangle$$
$$\lesssim \mathbb{E}\|\boldsymbol{X}_0\|^2 + (T - t)\mathbb{E}\|\boldsymbol{Z}\|^2$$
$$= R^2 + (T - t)d. \qquad (\text{F.14})$$

where the first inequality holds due to the independence of $\boldsymbol{X}_0$ and $\boldsymbol{Z}$. When $T - t \geq 1$, we have:

$$\mathbb{E}\|\boldsymbol{Y}_t\|^2 \lesssim \mathrm{e}^{-2(T-t)}\mathbb{E}\|\boldsymbol{X}_0\|^2 + \mathbb{E}\|\boldsymbol{Z}\|^2$$

$$= e^{-2(T-t)}R^2 + d. \tag{F.15}$$

Combine (F.14) and (F.15) together, we know that:

$$\mathbb{E}\|\boldsymbol{Y}_t\|^2 \lesssim R^2 + \min\{T-t, 1\}d. \tag{F.16}$$

After computing the second-order moment, using the inequality $\left(\mathbb{E}\|\boldsymbol{Y}_t\|\right)^2 \leq \mathbb{E}\|\boldsymbol{Y}_t\|^2$, we can bound the first-order moment as follows:

$$\mathbb{E}\|\boldsymbol{Y}_t\| \lesssim R + \sqrt{\min\{T-t, 1\}}\sqrt{d}. \tag{F.17}$$

Moreover, we consider the fourth-order moment using similar methods. When $T - t < 1$, we have $e^{-(T-t)} = \Theta(1)$ and $\sqrt{1 - e^{-2(T-t)}} = O(\sqrt{T-t})$. Thus we know that:

$$\begin{aligned}
\mathbb{E}\|\boldsymbol{Y}_t\|^4 &\lesssim \mathbb{E}\|\boldsymbol{X}_0\|^4 + (T-t)^2 \mathbb{E}\|\boldsymbol{Z}\|^4 \\
&\lesssim R^4 + (T-t)^2 d^2,
\end{aligned} \tag{F.18}$$

where the first inequality holds due to $\|\mathbf{x} - \mathbf{y}\|^4 \leq (\|\mathbf{x}\| + \|\mathbf{y}\|) \leq C(\|\mathbf{x}\|^4 + \|\mathbf{y}\|^4)$ for some constant $C$. Same as what we did earlier, when $T - t \geq 1$, we have:

$$\begin{aligned}
\mathbb{E}\|\boldsymbol{Y}_t\|^4 &\lesssim e^{-4(T-t)}\mathbb{E}\|\boldsymbol{X}_0\|^4 + \mathbb{E}\|\boldsymbol{Z}\|^4 \\
&\lesssim e^{-3(T-t)}R^4 + d^2.
\end{aligned} \tag{F.19}$$

Putting (F.18) and (F.19) together, we have:

$$\mathbb{E}\|\boldsymbol{Y}_t\|^4 \lesssim R^4 + \min\{T-t, 1\}^2 d^2. \tag{F.20}$$

After computing the fourth-order momentum, using the inequality $\mathbb{E}\|\boldsymbol{Y}_t\|^3 \leq \sqrt{\mathbb{E}\|\boldsymbol{Y}_t\|^2} \cdot \sqrt{\mathbb{E}\|\boldsymbol{Y}_t\|^4}$, we know that:

$$\mathbb{E}\|\boldsymbol{Y}_t\|^3 \lesssim R^3 + \min\{T-t, 1\}^{3/2} d^{3/2}. \tag{F.21}$$

**Bounding Divergence Discretization Error** $J_4$. Let $\mathbf{z} = F_t^{-1}(\mathbf{x})$. We start by bounding $\|\mathbf{z} - \mathbf{x}\|$. We know that

$$\begin{aligned}
\|\mathbf{z} - \mathbf{x}\| &= \|F_t^{-1}(\mathbf{x}) - \mathbf{x}\| \\
&= \left(e^{t-t_k} - 1\right)\left\|F_t^{-1}(\mathbf{x}) + \mathbf{s}_\theta(T - t_k, F_t^{-1}(\mathbf{x}))\right\| \\
&\leq 2(t - t_k)\left\|F_t^{-1}(\mathbf{x}) + \mathbf{s}_\theta(T - t_k, F_t^{-1}(\mathbf{x}))\right\| \\
&\leq 2(t - t_k)(L + 1)\left\|F_t^{-1}(\mathbf{x})\right\| + (t - t_k)c,
\end{aligned}$$

where the first inequality holds due to $t - t_k < 1$. The second inequality holds due to Assumption 5.4. From (F.4) we know that $\|F_t^{-1}(\mathbf{x})\| \leq 2\|\mathbf{x}\| + 1$. Hence we have:

$$\begin{aligned}
\|\mathbf{z} - \mathbf{x}\| &\lesssim (t - t_k)(L + 1)\|\mathbf{x}\| + (t - t_k)(L + c + 1) \\
&\overset{(i)}{\lesssim} (t - t_k)L(\|\mathbf{x}\| + 1).
\end{aligned} \tag{F.22}$$

Using Lemma F.3 and Lemma F.4, we have

$$\left|\int_{\Omega_t} \operatorname{tr}\left(\nabla^2 \log q(T - t_k, \mathbf{z}) - \nabla^2 \log q(T - t, \mathbf{x})\right) q(T - t, \mathbf{x})\mathrm{d}\mathbf{x}\right|$$

$$\lesssim \int_{\Omega_t}\left((t - t_k)\left(\frac{d}{\min\{T - t, 1\}^2} + \frac{R^2(\|\mathbf{x}\| + R)^2}{\min\{T - t, 1\}^4}\right) + \frac{R^3}{\min\{T - t, 1\}^3}\|\mathbf{x} - \mathbf{z}\|\right) q(T - t, \mathbf{x})\mathrm{d}\mathbf{x}$$

$$\overset{(i)}{\lesssim} \int_{\Omega_t}\left((t - t_k)\left(\frac{d}{\min\{T - t, 1\}^2} + \frac{R^2(\|\mathbf{x}\| + R)^2}{\min\{T - t, 1\}^4}\right)\right.$$

$$\left. + \frac{R^3}{\min\{T - t, 1\}^3}(t - t_k)L(\|\mathbf{x}\| + 1)\right) q(T - t, \mathbf{x})\mathrm{d}\mathbf{x}$$

$$\lesssim (t - t_k)\frac{d}{\min\{T - t, 1\}^2} + (t - t_k)\frac{R^2}{\min\{T - t, 1\}^4}\mathbb{E}(\|\boldsymbol{Y}_t\|^2 + R^2)$$

$$+ (t - t_k) \frac{R^3}{\min\{T - t, 1\}^3} L \cdot \mathbb{E}(\|\boldsymbol{Y}_t\| + 1)$$

$$\overset{(ii)}{\lesssim} (t - t_k) \frac{d}{\min\{T - t, 1\}^2} + (t - t_k) \frac{R^4 + R^2 \min\{T - t, 1\}d}{\min\{T - t, 1\}^4}$$

$$+ (t - t_k) \frac{LR^3}{\min\{T - t, 1\}^3} (R + \sqrt{T - t}\sqrt{d} + 1),$$

where $(i)$ holds due to (F.22). $(ii)$ holds due to (F.16) and (F.17). Assuming $R \geq 1$, we can further simplify its form:

$$|J_4| \lesssim (t - t_k)d \frac{R^2}{\min\{T - t, 1\}^3} + (t - t_k)(1 + L\min\{T - t, 1\}) \frac{R^4}{\min\{T - t, 1\}^4}$$

$$+ (t - t_k)\sqrt{d} \frac{LR^3}{\min\{T - t, 1\}^{5/2}}$$

$$\lesssim (t - t_k)d \frac{R^2}{\min\{T - t, 1\}^3} + (t - t_k) \frac{LR^4}{\min\{T - t, 1\}^4} + (t - t_k)\sqrt{d} \frac{LR^3}{\min\{T - t, 1\}^{5/2}}.$$

$$(\text{F.23})$$

**Bounding Score Discretization Error $J_3$.**

According to Lemma F.4, we know that:

$$\|\nabla \log q(T - t, \mathbf{x})\| \lesssim \frac{\|\mathbf{x}\| + R}{\min\{T - t, 1\}},$$

$$\left\|\nabla^2 \log q(T - t, \mathbf{x})\right\|_2 \lesssim \frac{1}{\min\{T - t, 1\}} + \frac{R^2}{\min\{T - t, 1\}^2} \overset{(i)}{\lesssim} \frac{R^2}{\min\{T - t, 1\}^2}.$$

Here $(i)$ is because we assumed that $R \geq 1$. Using Lemma F.3, we have:

$$\left\|\frac{\partial}{\partial t}\Big(\nabla \log q(T - t, \mathbf{x})\Big)\right\| \lesssim \frac{\|\mathbf{x}\| + R}{\min\{T - t, 1\}^2} + \frac{(\|\mathbf{x}\| + R)^2 R^2}{\min\{T - t, 1\}^2} + \frac{(\|\mathbf{x}\| + R)^3}{\min\{T - t, 1\}^3}$$

$$\lesssim \frac{(\|\mathbf{x}\| + R)^2 R^2}{\min\{T - t, 1\}^2} + \frac{(\|\mathbf{x}\| + R)^3}{\min\{T - t, 1\}^3}.$$

Thus we know that:

$$\left|\int_{\Omega_t} \Big(\nabla \log q_{T - t_k}(\mathbf{z}) - \nabla \log q_{T - t}(\mathbf{x})\Big)\nabla \log q(T - t, \mathbf{x})q(T - t, \mathbf{x})\mathrm{d}\mathbf{x}\right|$$

$$\lesssim \int_{\Omega_t} (t - t_k)\Big(\frac{(\|\mathbf{x}\| + R)^2 R^2}{\min\{T - t, 1\}^2} + \frac{(\|\mathbf{x}\| + R)^3}{\min\{T - t, 1\}^3}\Big)\frac{\|\mathbf{x}\| + R}{\min\{T - t, 1\}}q(T - t, \mathbf{x})\mathrm{d}\mathbf{x}$$

$$+ \int_{\Omega_t} \frac{R^2}{\min\{T - t, 1\}^2}\|\mathbf{x} - \mathbf{z}\|\frac{\|\mathbf{x}\| + R}{\min\{T - t, 1\}}q(T - t, \mathbf{x})\mathrm{d}\mathbf{x}$$

$$\lesssim (t - t_k)\frac{R^2}{\min\{T - t, 1\}^3}\mathbb{E}(\|\boldsymbol{Y}_t\| + R)^3 + (t - t_k)\frac{1}{\min\{T - t, 1\}^4}\mathbb{E}(\|\boldsymbol{Y}_t\| + R)^4$$

$$+ (t - t_k)\int_{\Omega_t} \frac{R^2}{\min\{T - t, 1\}^2}L(\|\mathbf{x}\| + 1)\frac{\|\mathbf{x}\| + R}{\min\{T - t, 1\}}q(T - t, \mathbf{x})\mathrm{d}\mathbf{x}$$

$$\overset{(i)}{\lesssim} (t - t_k)\frac{R^2}{\min\{T - t, 1\}^3}(R^3 + \min\{T - t, 1\}^{3/2}d^{3/2})$$

$$+ (t - t_k)\frac{1}{\min\{T - t, 1\}^4}(R^4 + \min\{T - t, 1\}^2 d^2)$$

$$+ (t - t_k)\frac{LR^2}{\min\{T - t, 1\}^3}(R^2 + \min\{T - t, 1\}d), \qquad (\text{F.24})$$

where we use (F.20) and (F.21) to bound the third and fourth moments of $\boldsymbol{Y}_t$. $(i)$ holds due to:

$$\int_{\Omega_t} (\|\mathbf{x}\| + 1)(\|\mathbf{x}\| + R)q(T - t, \mathbf{x})\mathrm{d}\mathbf{x} \lesssim \mathbb{E}(\|\boldsymbol{Y}_t\| + R)^2 \lesssim R^2 + \min\{T - t, 1\}d.$$

Reorganizing terms in (F.24), we have:

$$|J_3| \lesssim (t - t_k) \Big[ \frac{R^5}{\min\{T - t, 1\}^3} + d^{3/2} \frac{R^2}{\min\{T - t, 1\}^{3/2}} + \frac{R^4}{\min\{T - t, 1\}^4}$$

$$+ d^2 \frac{1}{\min\{T - t, 1\}^2} + \frac{LR^4}{\min\{T - t, 1\}^3} + d \frac{LR^2}{\min\{T - t, 1\}^2} \Big]. \tag{F.25}$$

**Putting $J_3$ and $J_4$ Together.**

By (F.23) and (F.25), and since we assume $R \geq 1$ and $L \geq 1$, we have:

$$|J_3| + |J_4| \lesssim (t - t_k) \, d \underbrace{\frac{R^2}{\min\{T - t, 1\}^3}}_{C_1} + (t - t_k) \underbrace{\frac{LR^4}{\min\{T - t, 1\}^4}}_{C_2} + (t - t_k) \sqrt{d} \underbrace{\frac{LR^3}{\min\{T - t, 1\}^{5/2}}}_{C_3}$$

$$+ (t - t_k) \Big[ \underbrace{\frac{R^5}{\min\{T - t, 1\}^3}}_{C_4} + d^{3/2} \underbrace{\frac{R^2}{\min\{T - t, 1\}^{3/2}}}_{C_5} + \underbrace{\frac{R^4}{\min\{T - t, 1\}^4}}_{C_6}$$

$$+ d^2 \underbrace{\frac{1}{\min\{T - t, 1\}^2}}_{C_7} + \underbrace{\frac{LR^4}{\min\{T - t, 1\}^3}}_{C_8} + d \underbrace{\frac{LR^2}{\min\{T - t, 1\}^2}}_{C_9} \Big]$$

$$\overset{(ii)}{\lesssim} (t - t_k) \Big[ C_1 + C_2 + C_3 + C_4 + C_5 + C_7 + C_9 \Big]$$

$$= (t - t_k) \Big[ d^2 \frac{1}{\min\{T - t, 1\}^2} + d^{3/2} \frac{R^2}{\min\{T - t, 1\}^{3/2}} + d \Big( \frac{LR^2 \min\{T - t, 1\} + R^2}{\min\{T - t, 1\}^3} \Big)$$

$$+ \sqrt{d} \frac{LR^3}{\min\{T - t, 1\}^{5/2}} + \frac{R^5 \min\{T - t, 1\} + LR^4}{\min\{T - t, 1\}^4} \Big]. \tag{F.26}$$

Here $(ii)$ is because $C_6 \leq C_2$ and $C_8 \leq C_2$.

**Combining Everything Together.**

Plugging (F.10) (F.11) (F.13) (F.26) back to (F.9), and taking the integral, we can obtain the following inequality:

$$\text{TV}(\widehat{\boldsymbol{Y}}_{T-\delta}, \boldsymbol{Y}_{T-\delta}) \lesssim \text{TV}(\pi_d, q_T)$$

$$+ \underbrace{\sum_k (t_{k+1} - t_k) \sqrt{\mathbb{E}_Q \, \text{tr} \left( \nabla \mathbf{s}_\theta(T - t_k, \boldsymbol{Y}_{t_k}) - \nabla^2 \log q(T - t_k, \boldsymbol{Y}_{t_k}) \right)^2}}_{K_1}$$

$$+ \underbrace{\sum_k (t_{k+1} - t_k) \sqrt{\frac{d}{\min\{T - t_{k+1}, 1\}}} \sqrt{\mathbb{E}_Q \left\| \mathbf{s}_\theta(T - t_k, \boldsymbol{Y}_{t_k}) - \nabla \log q_{T - t_k}(\boldsymbol{Y}_{t_k}) \right\|^2}}_{K_2}$$

$$+ \underbrace{\sum_k (t_{k+1} - t_k)^2 dL^2}_{K_3}$$

$$+ \sum_k (t_{k+1} - t_k)^2 \Big[ d^2 \frac{1}{\min\{T - t, 1\}^2} + d^{3/2} \frac{R^2}{\min\{T - t, 1\}^{3/2}} + d \Big( \frac{LR^2 \min\{T - t, 1\} + R^2}{\min\{T - t, 1\}^3} \Big)$$

$$+ \sqrt{d} \frac{LR^3}{\min\{T - t, 1\}^{5/2}} + \frac{R^5 \min\{T - t, 1\} + LR^4}{\min\{T - t, 1\}^4} \Big].$$

Using Assumption 5.3, we have $K_1 = \epsilon_{\text{div}}$. For $K_2$, using the Cauchy-Schwartz inequality, we have

$$\sum_k (t_{k+1} - t_k) \sqrt{\frac{d}{\min\{T - t_{k+1}, 1\}}} \sqrt{\mathbb{E}_Q \left\| \mathbf{s}_\theta(T - t_k, \boldsymbol{Y}_{t_k}) - \nabla \log q_{T - t_k}(\boldsymbol{Y}_{t_k}) \right\|^2}$$

$$\leq \sqrt{\sum_k (t_{k+1} - t_k) \frac{d}{\min\{T - t_{k+1}, 1\}}} \sqrt{\sum_k (t_{k+1} - t_k) \mathbb{E}_Q \left\| \mathbf{s}_\theta(T - t_k, \boldsymbol{Y}_{t_k}) - \nabla \log q_{T - t_k}(\boldsymbol{Y}_{t_k}) \right\|^2}$$

$$\leq \sqrt{\eta N d} \epsilon_{\text{score}}.$$

For $K_3$, we know that:

$$\sum_k (t_{k+1} - t_k)^2 d L^2 \leq \eta^2 N d L^2.$$

For $K_4$, using our assumption on time schedule $t_{k+1} - t_k \leq \eta \left( T - t_{k+1} \right)$, we have:

$$K_4 \lesssim \eta^2 N \Big[ d^2 + d^{3/2} R^2 + L R^2 d + R^2 d \frac{1}{\delta}$$

$$+ L R^3 \sqrt{\frac{d}{\delta}} + R^5 \frac{1}{\delta} + L R^4 \frac{1}{\delta^2} \Big]$$

$$\overset{(ii)}{\lesssim} \eta^2 N \Big[ L R^4 d^2 + d^{3/2} R^2 + L R^2 d + R^5 \frac{1}{\delta} + L R^4 \frac{1}{\delta^2} \Big]$$

$$\overset{(iii)}{\lesssim} \eta^2 N \Big[ L R^4 d^2 + R^5 \frac{1}{\delta} + L R^4 \frac{1}{\delta^2} \Big],$$

where $(ii)$ holds due to $R^2 d \frac{1}{\delta} \leq d^2 + \frac{L R^4}{\delta^2}$ and $2 L R^3 \sqrt{\frac{d}{\delta}} \leq L R^2 d + L R^4 \frac{1}{\delta^2}$. $(iii)$ holds due to $2 d^{3/2} R^2 \leq L R^4 d^2 + L R^2 d$ and $2 L R^2 d \leq L R^4 d^2 + L R^4 \frac{1}{\delta^2}$. Putting $K_1$, $K_2$, $K_3$ and $K_4$ together, we have

$$\text{TV}(\widehat{\boldsymbol{Y}}_{T-\delta}, \boldsymbol{Y}_{T-\delta}) \lesssim \text{TV}(q_T, \pi_d) + \epsilon_{\text{div}} + \sqrt{d} \sqrt{\eta N} \epsilon_{\text{score}}$$

$$+ \eta^2 N \Big[ L R^4 (d^2 + \frac{1}{\delta^2}) + L^2 d + \frac{R^5}{\delta} \Big].$$

This completes the proof of Theorem 5.5. $\qquad\square$

### F.2 Proof of Lemma 5.6 and Corollary 5.7

*Proof of Lemma 5.6.* By proposition 4 in Benton et al. (2024), we have:

$$\text{KL}(q_T \| \pi_d) \lesssim d e^{-2T}, \text{ for } T \geq 1.$$

Then by Pinsker's inequality, we know that:

$$\text{TV}(q_T, \pi_d) \leq \sqrt{\frac{1}{2} \text{KL}(q_T \| \pi_d)} \lesssim \sqrt{d} e^{-T}.$$

This completes the proof. $\qquad\square$

*Proof of Corollary 5.7.* Please refer to Benton et al. (2024)'s Appendix D for a detailed derivation of the existence of such time schedule. Since we take $T = \log(d/\epsilon^2)/2$ and $N = L R^4 \Theta \big( d^2 (T + \log(1/\delta))^2 / \epsilon \big)$, using $\epsilon \leq 1/L$, we can easily show that $\eta \leq \min\{1/(12 L^2 d^2), 1/(24 L^2 R^2 d)\}$. Hence by Theorem 5.5 and Lemma 5.6, we have:

$$\text{TV}(\widehat{\boldsymbol{Y}}_{T-\delta}, \boldsymbol{Y}_{T-\delta}) \lesssim \sqrt{d} e^{-T} + \epsilon_{\text{div}} + \sqrt{d} \sqrt{\eta N} \epsilon_{\text{score}} + \eta^2 N \Big[ L R^4 \Big( d^2 + \frac{1}{\delta^2} \Big) + L^2 d + \frac{R^5}{\delta} \Big].$$

Since we take $\delta = 1/d$ and assumed that $d \geq R/L + L/R^4$, we can further simply the bound of TV to:

$$\text{TV}(\widehat{\boldsymbol{Y}}_{T-\delta}, \boldsymbol{Y}_{T-\delta}) \lesssim \sqrt{d} e^{-T} + \epsilon_{\text{div}} + \sqrt{d} \sqrt{\eta N} \epsilon_{\text{score}} + \eta^2 N L R^4 d^2.$$

Since $T = \log(d/\epsilon^2)/2$ and $N = L R^4 \Theta \big( d^2 (T + \log(1/\delta))^2 / \epsilon \big)$, we know that

$$\sqrt{d} e^{-T} \leq \epsilon,$$

$$\eta^2 N L R^4 d^2 = \frac{1}{N} (T + \log(1/\delta))^2 L R^4 d^2 \leq \epsilon.$$

Recall that we assume sufficiently small score estimation and divergence error, this completes the proof of Corollary 5.7 $\qquad\square$

## G   ANALYSIS OF VE+DDIM

In this section, we focus on the case where $f(\mathbf{x}) = 0$ and $g(t) = 1$, i.e. taking $\sigma(t)^2 = t$ in VE-SDE. Specifically, the forward process $\boldsymbol{X}_t$ satisfies:

$$\mathrm{d}\boldsymbol{X}_t = \mathrm{d}\boldsymbol{W}_t, \quad \boldsymbol{X}_0 \sim q_0, \tag{G.1}$$

and the true reverse process $\boldsymbol{Y}_t$ satisfies:

$$\mathrm{d}\boldsymbol{Y}_t = \frac{1}{2}\nabla \log q_{T-t}(\boldsymbol{Y}_t)\mathrm{d}t, \quad \boldsymbol{Y}_0 \sim q_T. \tag{G.2}$$

Then, the forward process has the following closed-form expression:

$$\boldsymbol{Y}_t = \boldsymbol{X}_{T-t} = \boldsymbol{X}_0 + \sqrt{T-t}\boldsymbol{Z}, \quad \boldsymbol{Z} \sim N(0, \mathbf{I}_d). \tag{G.3}$$

Recall that we interpret DDIM as a numerical scheme, and it provides the following discretized version of the reverse process:

$$\widehat{\boldsymbol{Y}}_{t_{k+1}} = \widehat{\boldsymbol{Y}}_{t_k} + (t_{k+1} - t_k)l\left(1 - \sqrt{1 - \frac{1}{l}}\right)\mathbf{s}_\theta(T - t_k, \widehat{\boldsymbol{Y}}_{t_k}), \quad l = \frac{T - t_k}{t_{k+1} - t_k}. \tag{G.4}$$

Therefore, denoting $l(1 - \sqrt{1 - 1/l})$ by $c_l$, we have $c_l \leq 1$. Then we can define the following interpolation operator:

$$F_t(\mathbf{z}) = \mathbf{z} + (t - t_k)c_l \mathbf{s}_\theta(T - t_k, \mathbf{z}), \tag{G.5}$$

satisfying $F_{t_{k+1}}(\widehat{\boldsymbol{Y}}_{t_k}) = \widehat{\boldsymbol{Y}}_{t_{k+1}}$. Under Assumption 5.4, we know $\mathbf{s}_\theta(t, \cdot)$ is Lipschitz. Therefore, suppose $\eta \leq 1/L$, then we have $1/2 \leq \|\nabla F_t\| \leq 3/2$. In particular, $F_t$ is invertible.

Moreover, using Assumption 5.4 and $F_t^{-1}(\mathbf{x}) - \mathbf{x} = -(t - t_k)c_l \cdot \mathbf{s}_\theta\big(T - t_k, F_t^{-1}(\mathbf{x})\big)$, when $\eta \leq \frac{1}{L+c}$, we have

$$\begin{aligned}
\|F_t^{-1}(\mathbf{x}) - \mathbf{x}\| &= (t - t_k)c_l\|\mathbf{s}_\theta\big(T - t_k, F_t^{-1}(\mathbf{x})\big)\| \\
&\leq (t - t_k)L\|F_t^{-1}(\mathbf{x})\| + (t - t_k)c \\
&\leq \|\mathbf{x}\| + 1.
\end{aligned} \tag{G.6}$$

Thus we know that

$$\|F_t^{-1}(\mathbf{x})\| \leq 2\|\mathbf{x}\| + 1. \tag{G.7}$$

Before we start the proof, we introduce several important lemmas used in our proof. Let $c_1 = (1 + d/2)4L^2$ and $c_2 = (1 + d/2)(L + c)^2$. Using our assumption on $\eta$, we know that

$$\eta \leq \min\left\{\frac{1}{4c_1 dT}, \frac{1}{c_2}, \frac{1}{4R^2 c_1}\right\}. \tag{G.8}$$

The following two lemmas are related to the ratios $p_{\boldsymbol{Y}_t}(\mathbf{x})/p_{F_t(\boldsymbol{Y}_{t_k})}$.

**Lemma G.1.** Let $\boldsymbol{Y}_t$ and $F_t(\mathbf{z})$ be defined in (G.2) and (G.5). Under Assumption 5.4 (Lipschitz condition on $\mathbf{s}_\theta$), Assumption 5.1 (bounded support) and suppose the time schedule satisfies (2.6), assuming $\eta \leq \min\{\frac{1}{d}, \frac{1}{L+c}, \frac{1}{4c_1(T-t)}, \frac{1}{c_1 R^2}, \frac{1}{c_2}\}$, we have

$$\frac{p_{\boldsymbol{Y}_t}(\mathbf{x})}{p_{F_t(\boldsymbol{Y}_{t_k})}(\mathbf{x})} \lesssim \mathrm{e}^{(t-t_k)c_1\|\mathbf{x}\|^2 + (t-t_k)c_2},$$

where $c_1 = (1 + d/2)4L^2$ and $c_2 = (1 + d/2)(L + c)^2$. Moreover, we have

$$\int_{\Omega_t}\left(\frac{p_{\boldsymbol{Y}_t}(\mathbf{x})}{p_{F_t(\boldsymbol{Y}_{t_k})}(\mathbf{x})}\right)^2 p_{F_t(\boldsymbol{Y}_{t_k})}(\mathbf{x})\mathrm{d}\mathbf{x} \lesssim 1.$$

**Lemma G.2.** Let $\boldsymbol{Y}_t$ and $F_t(\mathbf{z})$ be defined in (G.2) and (G.5). Under Assumption 5.4, 5.1 and 2.6, suppose $\eta \leq \min\{\frac{1}{c_2}, \frac{1}{c_1 R^2}, \frac{1}{2(d+2)c_1(T-t)}\}$, where $c_1 = (1 + d/2)4L^2$ and $c_2 = (1 + d/2)(L + c)^2$. then, we have:

$$\int_{\Omega_t}\left\|\frac{\nabla q(T - t, \mathbf{x})}{p_{F_t(\boldsymbol{Y}_{t_k})}(\mathbf{x})}\right\|^2 p_{F_t(\boldsymbol{Y}_{t_k})}(\mathbf{x})\mathrm{d}\mathbf{x} \lesssim \frac{d}{T - t}.$$

The following two lemmas show the upper bound of the derivatives of the true score function of the VE forward process concerning $t$ and $\mathbf{x}$ separately.

**Lemma G.3.** When $X_t$ is defined as in (G.1), then under Assumption 5.1, we have:

$$\left\| \frac{\partial}{\partial t}\Big( \nabla \log q(t, \mathbf{x}) \Big) \right\| \lesssim \frac{\|\mathbf{x}\| + R}{t^2} + \frac{(\|\mathbf{x}\| + R)^3}{t^3}.$$

Moreover, we have:

$$\left| \frac{\partial}{\partial t}\Big( \operatorname{tr}\big( \nabla^2 \log q(t, \mathbf{x}) \big) \Big) \right| \lesssim \frac{d}{t^2} + \frac{R^2(\|\mathbf{x}\| + R)^2}{t^4}.$$

**Lemma G.4.** When $X_t$ is defined as in (G.1), then under Assumption 5.1, we have:

$$\|\nabla \log q(t, \mathbf{x})\| \lesssim \frac{\|\mathbf{x}\| + R}{t},$$

$$\big\|\nabla^2 \log q(t, \mathbf{x})\big\|_2 \le \frac{1}{t} + \frac{R^2}{t^2},$$

$$\big\|\nabla \operatorname{tr}\big(\nabla^2 \log q(t, \mathbf{x})\big)\big\| \le \frac{R^3}{t^3}.$$

The proofs of the above lemmas can be found in later sections.

### G.1 PROOF OF THEOREM 5.9

*Proof of Theorem 5.9.* From the discussion in Section 5.2, we know that:

$$\Phi_k(t, \mathbf{x}) = c_l \mathbf{s}_\theta(T - t_k, \mathbf{x}) - \frac{1}{2}\nabla \log q_{T-t_k}(\mathbf{x}), \tag{G.9}$$

$$\Psi_k(t, \mathbf{x}) = c_l \nabla \mathbf{s}_\theta(T - t_k, \mathbf{x}) - \frac{1}{2}\nabla^2 \log q_{T-t_k}(\mathbf{x}). \tag{G.10}$$

The interpolation operator can be expressed as:

$$F_t(\mathbf{x}) = \mathbf{x} + c_l(t - t_k)\mathbf{s}_\theta(T - t_k, \mathbf{x}), \tag{G.11}$$

in this specific case. Recall that $g(t) = 1$ and $\mathbf{f}(T - t, \mathbf{x}) = 0$. Denote $F_t^{-1}(\mathbf{x})$ by $\mathbf{z}$. From (G.11) we have:

$$\frac{\partial}{\partial t}F(t, \mathbf{x}) = c_l \mathbf{s}_\theta(T - t_k, \mathbf{x}).$$

Thus we know that

$$\nabla\Big[\frac{\partial}{\partial t}F\Big]\big(t, F_t^{-1}(\mathbf{x})\big) = c_l \nabla \mathbf{s}_\theta(T - t_k, \mathbf{z}). \tag{G.12}$$

By Theorem 4.3, we know that:

$$\mathrm{TV}(\boldsymbol{Y}_{T-\delta}, \widehat{\boldsymbol{Y}}_{t_N}) \le \mathrm{TV}(q_T, \pi_d)$$

$$+ \sum_{k=0}^{N-1} \int_{t_k}^{t_{k+1}} \bigg[ \underbrace{\sqrt{\mathbb{E}\big[\|\Phi_k(t, \boldsymbol{Y}_{t_k})\|^2\big]}\sqrt{\int \bigg\| \frac{\nabla \log q(T - t, \mathbf{x})p_{\boldsymbol{Y}_t}(\mathbf{x})}{p_{F_t(\boldsymbol{Y}_{t_k})}(\mathbf{x})} \bigg\|^2 p_{F_t(\boldsymbol{Y}_{t_k})}(\mathbf{x})\mathrm{d}\mathbf{x}}}_{J_1:\ \text{Score estimation error}}$$

$$+ \underbrace{\sqrt{\mathbb{E}\big[\operatorname{tr}\big(\Psi_k(t, \boldsymbol{Y}_{t_k})\big)^2\big]}\sqrt{\int \Big(\frac{p_{\boldsymbol{Y}_t}(\mathbf{x})}{p_{F_t(\boldsymbol{Y}_{t_k})}(\mathbf{x})}\Big)^2 p_{F_t(\boldsymbol{Y}_{t_k})}(\mathbf{x})\mathrm{d}\mathbf{x}}}_{J_2:\ \text{Divergence estimation error}}$$

$$+ \underbrace{\frac{1}{2}\int \big|\big(\nabla \log q_{T-t_k}(\mathbf{z}) - \nabla \log q_{T-t}(\mathbf{x})\big) \cdot \nabla \log q(T - t, \mathbf{x})\big|q(T - t, \mathbf{x})\mathrm{d}\mathbf{x}}_{J_3:\ \text{Score discretization error}}$$

$$+ \underbrace{\frac{1}{2}\int \big|\operatorname{tr}\big(\nabla^2 \log q(T - t_k, \mathbf{z}) - \nabla^2 \log q(T - t, \mathbf{x})\big)\big|q(T - t, \mathbf{x})\mathrm{d}\mathbf{x}}_{J_4:\ \text{Divergence discretization error}}$$

$$+ \underbrace{\max_{\mathbf{x}} \left| \operatorname{tr} \left[ c_l \nabla \mathbf{s}_\theta (T - t_k, \mathbf{z}) \right] \left( \nabla F_t^{-1}(\mathbf{x}) - \mathbf{I} \right) \right|}_{J_5: \text{Bias}} \right] \mathrm{d}t. \tag{G.13}$$

Since we assumed $\eta \leq \min\{1/(12L^2 T d^2), 1/(24L^2 R^2 d)\}$, we can easily verify that $\eta$ is small enough to satisfy the condition of Lemma G.1 and Lemma G.2.

**Bounding the score Estimation Error $J_1$.** For the first square root in $J_1$, we know that:

$$\sqrt{\mathbb{E}\left[\left\|\Phi_k(t, \mathbf{Y}_{t_k})\right\|^2\right]} = \sqrt{\mathbb{E}\left[\left\|c_l \mathbf{s}_\theta(T - t_k, \mathbf{Y}_{t_k}) - \frac{1}{2}\nabla \log q_{T-t_k}(\mathbf{Y}_{t_k})\right\|^2\right]}$$

$$\leq \sqrt{\mathbb{E}\left[\left[\left\|\frac{1}{2}\mathbf{s}_\theta(T - t_k, \mathbf{Y}_{t_k}) - \frac{1}{2}\nabla \log q_{T-t_k}(\mathbf{Y}_{t_k})\right\| + (c_l - \frac{1}{2})\left\|\mathbf{s}_\theta(T - t_k, \mathbf{Y}_{t_k})\right\|\right]^2\right]}$$

$$\lesssim \sqrt{\mathbb{E}\left[\left\|\frac{1}{2}\mathbf{s}_\theta(T - t_k, \mathbf{Y}_{t_k}) - \frac{1}{2}\nabla \log q_{T-t_k}(\mathbf{Y}_{t_k})\right\|^2\right]} + \sqrt{\mathbb{E}\left[(c_l - \frac{1}{2})^2\left\|\mathbf{s}_\theta(T - t_k, \mathbf{Y}_{t_k})\right\|^2\right]}. \tag{G.14}$$

Using Assumption 5.4, we have $\left\|\mathbf{s}_\theta(T - t_k, \mathbf{Y}_{t_k})\right\| \leq L\|\mathbf{Y}_{t_k}\| + c$. Since $\mathbf{Y}_{t_k} = \mathbf{X}_0 + \sqrt{T - t_k}\mathbf{Z}$, we have:

$$\mathbb{E}\left\|\mathbf{Y}_{t_k}\right\|^2 = \mathbb{E}\left\|\mathbf{X}_0\right\|^2 + (T - t_k)\mathbb{E}\left\|\mathbf{Z}\right\|^2$$
$$\leq R^2 + (T - t_k)d.$$

Moreover, recall that we take $l = (T - t_k)/(t_{k+1} - t_k) \geq 1/\eta$. We can show that $0 \leq c_l - \frac{1}{2} \leq \eta$. Hence we have:

$$\sqrt{\mathbb{E}\left[(c_l - \frac{1}{2})^2\left\|\mathbf{s}_\theta(T - t_k, \mathbf{Y}_{t_k})\right\|^2\right]} \lesssim (c_l - \frac{1}{2})\sqrt{\mathbb{E}\left[L^2\left\|\mathbf{Y}_{t_k}\right\|^2 + c^2\right]}$$

$$\leq \eta\sqrt{L^2 R^2 + L^2(T - t_k)d + c^2}. \tag{G.15}$$

For the second term, since our $\eta$ satisfies the condition of Lemma G.2 we can apply Lemma G.2 and obtain

$$\int_{\Omega_t} \left\|\frac{\nabla \log q(T - t, \mathbf{x})p_{\mathbf{Y}_t}(\mathbf{x})}{p_{F_t(\mathbf{Y}_{t_k})}(\mathbf{x})}\right\|^2 p_{F_t(\mathbf{Y}_{t_k})}(\mathbf{x})\mathrm{d}\mathbf{x} \lesssim \frac{d}{T - t}. \tag{G.16}$$

Combining (G.15), (G.14) and (G.16), we have:

$$J_1 \lesssim \sqrt{\frac{d}{T - t}}\sqrt{\mathbb{E}\left[\left\|\mathbf{s}_\theta(T - t_k, \mathbf{Y}_{t_k}) - \nabla \log q_{T-t_k}(\mathbf{Y}_{t_k})\right\|^2\right]}$$

$$+ \eta\sqrt{\frac{d}{T - t}}\sqrt{L^2 R^2 + L^2(T - t_k)d + c^2}. \tag{G.17}$$

**Bounding the Divergence Estimation Error $J_2$.** For the first term in $J_2$, we know that:

$$\sqrt{\mathbb{E}\left[\operatorname{tr}\left(c_l \nabla \mathbf{s}_\theta(T - t_k, \mathbf{Y}_{t_k}) - \frac{1}{2}\nabla^2 \log q_{T-t_k}(\mathbf{Y}_{t_k})\right)^2\right]}$$

$$\lesssim \sqrt{\mathbb{E}\left[\operatorname{tr}\left(\frac{1}{2}\nabla \mathbf{s}_\theta(T - t_k, \mathbf{Y}_{t_k}) - \frac{1}{2}\nabla^2 \log q_{T-t_k}(\mathbf{Y}_{t_k})\right)^2 + \left(c_l - \frac{1}{2}\right)^2\left(\operatorname{tr}\nabla \mathbf{s}_\theta(T - t_k, \mathbf{Y}_{t_k})\right)^2\right]}$$

$$\lesssim \sqrt{\mathbb{E}\left[\operatorname{tr}\left(\nabla \mathbf{s}_\theta(T - t_k, \mathbf{Y}_{t_k}) - \nabla^2 \log q_{T-t_k}(\mathbf{Y}_{t_k})\right)^2\right]} + \underbrace{(c_l - \frac{1}{2})\sqrt{\mathbb{E}\left[\left(\operatorname{tr}\nabla \mathbf{s}_\theta(T - t_k, \mathbf{Y}_{t_k})\right)^2\right]}}_{K_1}. \tag{G.18}$$

Using Assumption 5.4, we know that $\|\nabla \mathbf{s}_\theta(T - t_k, \mathbf{x})\|_2 \leq L$, thus $|\operatorname{tr}\nabla \mathbf{s}_\theta(T - t_k, \mathbf{x})| \leq dL$. Same as we did with $J_1$, we have $0 \leq c_l - \frac{1}{2} \leq \eta$. Moreover, $K_1$ can be further bounded by

$$\left(c_l - \frac{1}{2}\right)\sqrt{\mathbb{E}\left[\left(\operatorname{tr}\nabla \mathbf{s}_\theta(T - t_k, \mathbf{Y}_{t_k})\right)^2\right]} \leq \eta dL. \tag{G.19}$$

For the second term, since our $\eta$ satisfies the condition of Lemma G.1, we can apply it and obtain

$$\int_{\Omega_t} \left( \frac{p_{\mathbf{Y}_t}(\mathbf{x})}{p_{F_t(\mathbf{Y}_{t_k})}(\mathbf{x})} \right)^2 p_{F_t(\mathbf{Y}_{t_k})}(\mathbf{x})\mathrm{d}\mathbf{x} \lesssim 1. \tag{G.20}$$

Combining (G.18), (G.19) and (G.20) together, we have:

$$J_2 \lesssim \sqrt{\mathbb{E}_Q \operatorname{tr} \left( \nabla \mathbf{s}_\theta(T - t_k, \mathbf{Y}_{t_k}) - \nabla^2 \log q(T - t_k, \mathbf{Y}_{t_k}) \right)^2} + \eta dL. \tag{G.21}$$

**Bounding the Bias Term $J_5$.** We know that:

$$\nabla F_t^{-1}(\mathbf{x}) = \left[ (\nabla F_t)\left(F_t^{-1}(\mathbf{x})\right) \right]^{-1}$$
$$= \left[ \mathbf{I}_d + c_l(t - t_k)\nabla \mathbf{s}_\theta\left(T - t_k, F_t^{-1}(\mathbf{x})\right) \right]^{-1}.$$

Using Assumption 5.4 and $t - t_k \leq \eta \leq 1$, we have:

$$\|(\nabla F_t)\left(F_t^{-1}(\mathbf{x})\right) - \mathbf{I}_d\|_2 \leq (t - t_k)\|c_l \nabla \mathbf{s}_\theta\left(T - t_k, F_t^{-1}(\mathbf{x})\right)\|_2$$
$$\leq (t - t_k)(1 + L).$$

Here we use $c_l \leq 1$. Let $\mathbf{A} = (\nabla F_t)\left(F_t^{-1}(\mathbf{x})\right) - \mathbf{I}_d$. Then we have $\|\mathbf{A}\|_2 \leq (t - t_k)(1 + L)$. Thus we know that:

$$\|\nabla\left(F_t^{-1}(\mathbf{x})\right) - \mathbf{I}_d\|_2 = \|(\mathbf{I}_d + \mathbf{A})^{-1} - \mathbf{I}_d\|_2$$
$$\overset{(i)}{=} \left\| \sum_{n=1}^{\infty} (-1)^n \mathbf{A}^n \right\|_2$$
$$\overset{(ii)}{\leq} \sum_{n=1}^{\infty} \|\mathbf{A}\|_2^n$$
$$\overset{(iii)}{\leq} 2(t - t_k)(1 + L),$$

here $(i)$ holds because of the series expansion for $\|\mathbf{A}\|_2 < 1$. $(ii)$ holds due to the Cauchy-Schwarz inequality. $(iii)$ holds due to our assumption $\eta \leq 1/2(1 + L)$. Then we know that:

$$J_5 = \max \left| \operatorname{tr} \left( c_l \nabla \mathbf{s}_\theta(T - t_k, \mathbf{z})\left(\nabla\left(F_t^{-1}(\mathbf{x})\right) - \mathbf{I}_d\right) \right) \right|$$
$$\overset{(i)}{\leq} d \max \|\nabla \mathbf{s}_\theta(T - t_k, \mathbf{z})\left(\nabla\left(F_t^{-1}(\mathbf{x})\right) - \mathbf{I}_d\right)\|_2$$
$$\leq d \max \|\nabla \mathbf{s}_\theta(T - t_k, \mathbf{z})\|_2 \|\nabla\left(F_t^{-1}(\mathbf{x})\right) - \mathbf{I}_d\|_2$$
$$\leq dL(t - t_k)2(1 + L),$$

here $(i)$ holds due to $\operatorname{tr}(\mathbf{A}) \leq d\|\mathbf{A}\|_2$ and $c_l \leq 1$. Thus we have:

$$|J_5| \lesssim (t - t_k)dL^2. \tag{G.22}$$

**Estimating $\mathbf{z} - \mathbf{x}$.** By Assumption 5.4, we have:

$$\|\mathbf{z} - \mathbf{x}\| = \|F_t^{-1}(\mathbf{x}) - \mathbf{x}\|$$
$$= (t - t_k)\|\mathbf{s}_\theta(T - t_k, F_t^{-1}(\mathbf{x}))\|$$
$$\leq (t - t_k)L\|F_t^{-1}(\mathbf{x})\| + (t - t_k)c.$$

Since we assume that $t - t_k \leq \eta \leq \min\{\frac{1}{4L}, \frac{1}{4c}\}$. From (G.7) we know that $\|F_t^{-1}(\mathbf{x})\| \leq 2\|\mathbf{x}\| + 1$. Hence we have:

$$\|\mathbf{z} - \mathbf{x}\| \lesssim (t - t_k)L(2\|\mathbf{x}\| + 1) + (t - t_k)c$$
$$\overset{(i)}{\lesssim} (t - t_k)L(\|\mathbf{x}\| + 1). \tag{G.23}$$

**Bounding the Moments of $Y_t$.**

Before we start estimating $J_3$ and $J_4$, we do some preparation work. Our goal is to estimate the moments of $Y_t$. Since we have $Y_t = X_{T-t} = X_0 + \sqrt{T-t}Z$, here $Z \sim N(0, \mathbf{I}_d)$. We have

$$\mathbb{E}\|Y_t\|^2 = \mathbb{E}\|X_0\|^2 + (T-t)\mathbb{E}\|Z\|^2$$
$$\leq R^2 + (T-t)d. \tag{G.24}$$

After computing the second order momentum, by the inequality $\left(\mathbb{E}\|Y_t\|\right)^2 \leq \mathbb{E}\|Y_t\|^2$, we know that:

$$\mathbb{E}\|Y_t\| \lesssim R + \sqrt{T-t}\sqrt{d}. \tag{G.25}$$

Moreover, we consider the fourth order momentum as we will need to estimate it later: we have

$$\mathbb{E}\|Y_t\|^4 \lesssim \mathbb{E}\|X_0\|^4 + (T-t)^2\mathbb{E}\|Z\|^4$$
$$\lesssim R^4 + (T-t)^2 d^2. \tag{G.26}$$

After computing the fourth order momentum, by the inequality $\mathbb{E}\|Y_t\|^3 \leq \sqrt{\mathbb{E}\|Y_t\|^2} \cdot \sqrt{\mathbb{E}\|Y_t\|^4}$, we know that:

$$\mathbb{E}\|Y_t\|^3 \lesssim R^3 + (T-t)^{3/2}d^{3/2}. \tag{G.27}$$

**Bounding Divergence Discretization Error $J_4$.**

Using Lemma G.3 and Lemma G.4, we know that

$$\left|\frac{\partial}{\partial t}\Big(\operatorname{tr}\big(\nabla^2 \log q(T-t, \mathbf{x})\big)\Big)\right| \lesssim \frac{d}{(T-t)^2} + \frac{R^2(\|\mathbf{x}\| + R)^2}{(T-t)^4},$$

$$\left\|\nabla\operatorname{tr}\big(\nabla^2 \log q(t, \mathbf{x})\big)\right\| \leq \frac{R^3}{(T-t)^3}.$$

Thus we know that:

$$\left|\int_{\Omega_t} \operatorname{tr}\Big(\nabla^2 \log q(T-t_k, \mathbf{z}) - \nabla^2 \log q(T-t, \mathbf{x})\Big)q(T-t, \mathbf{x})\mathrm{d}\mathbf{x}\right|$$

$$\lesssim \int_{\Omega_t}\left((t-t_k)\big(\frac{d}{(T-t)^2} + \frac{R^2(\|\mathbf{x}\| + R)^2}{(T-t)^4}\big) + \frac{R^3}{(T-t)^3}\|\mathbf{x} - \mathbf{z}\|\right)q(T-t, \mathbf{x})\mathrm{d}\mathbf{x}$$

$$\overset{(i)}{\lesssim} \int_{\Omega_t}\left((t-t_k)\big(\frac{d}{(T-t)^2} + \frac{R^2(\|\mathbf{x}\| + R)^2}{(T-t)^4}\big) + \frac{R^3}{(T-t)^3}(t-t_k)L(\|\mathbf{x}\| + 1)\right)q(T-t, \mathbf{x})\mathrm{d}\mathbf{x}$$

$$\lesssim (t-t_k)\frac{d}{(T-t)^2} + (t-t_k)\frac{R^2}{(T-t)^4}\mathbb{E}(\|Y_t\|^2 + R^2) + (t-t_k)\frac{R^3}{(T-t)^3}L \cdot \mathbb{E}(\|Y_t\| + 1)$$

$$\overset{(ii)}{\lesssim} (t-t_k)\frac{d}{(T-t)^2} + (t-t_k)\frac{R^4 + R^2(T-t)d}{(T-t)^4} + (t-t_k)\frac{LR^3}{(T-t)^3}(R + \sqrt{T-t}\sqrt{d} + 1).$$

Here $(i)$ is due to (G.23) and $(ii)$ is due to (G.24) and (G.25). Assuming $R \geq 1$, we can further simplify its form:

$$|J_4| \lesssim (t-t_k)\left[d\frac{1}{(T-t)^2} + d\frac{R^2}{(T-t)^3} + (1 + L(T-t))\frac{R^4}{(T-t)^4} + \sqrt{d}\frac{LR^3}{(T-t)^{5/2}}\right]. \tag{G.28}$$

**Bounding the Score Discretization Error $J_3$.**

Using Lemma G.4, we know that:

$$\|\nabla \log q(t, \mathbf{x})\| \lesssim \frac{\|\mathbf{x}\| + R}{t},$$

$$\left\|\nabla^2 \log q(t, \mathbf{x})\right\|_2 \leq \frac{1}{t} + \frac{R^2}{t^2}.$$

Moreover, by Lemma G.3, we have:

$$\left\|\frac{\partial}{\partial t}\Big(\nabla \log q(t, \mathbf{x})\Big)\right\| \lesssim \frac{\|\mathbf{x}\| + R}{t^2} + \frac{(\|\mathbf{x}\| + R)^3}{t^3}.$$

Thus we know that:

$$\left| \int_{\Omega_t} \Big( \nabla \log q_{T-t_k}(\mathbf{z}) - \nabla \log q_{T-t}(\mathbf{x}) \Big) \nabla \log q(T-t, \mathbf{x}) q(T-t, \mathbf{x}) \mathrm{d}\mathbf{x} \right|$$

$$\lesssim \int_{\Omega_t} (t-t_k) \Big( \frac{\|\mathbf{x}\|+R}{(T-t)^2} + \frac{(\|\mathbf{x}\|+R)^3}{(T-t)^3} \Big) \frac{\|\mathbf{x}\|+R}{T-t} q(T-t, \mathbf{x}) \mathrm{d}\mathbf{x}$$

$$+ \int_{\Omega_t} \Big( \frac{1}{T-t} + \frac{R^2}{(T-t)^2} \Big) \|\mathbf{x} - \mathbf{z}\| \frac{\|\mathbf{x}\|+R}{T-t} q(T-t, \mathbf{x}) \mathrm{d}\mathbf{x}$$

$$\lesssim (t-t_k) \frac{1}{(T-t)^3} \mathbb{E}(\|\mathbf{Y}_t\| + R)^2 + (t-t_k) \frac{1}{(T-t)^4} \mathbb{E}(\|\mathbf{Y}_t\| + R)^4$$

$$+ (t-t_k) \int_{\Omega_t} \Big( \frac{1}{T-t} + \frac{R^2}{(T-t)^2} \Big) L(\|\mathbf{x}\| + 1) \frac{\|\mathbf{x}\|+R}{T-t} q(T-t, \mathbf{x}) \mathrm{d}\mathbf{x}$$

$$\overset{(i)}{\lesssim} (t-t_k) \frac{1}{(T-t)^3} (R^2 + (T-t)d) + (t-t_k) \frac{1}{(T-t)^4} (R^4 + (T-t)^2 d^2)$$

$$+ (t-t_k) \frac{L}{(T-t)^2} (R^2 + (T-t)d) + (t-t_k) \frac{LR^2}{(T-t)^3} (R^2 + (T-t)d). \qquad (G.29)$$

Here we use (G.26) and (G.24) to bound the second and fourth momentum of $\mathbf{Y}_t$, and $(i)$ is due to:

$$\int_{\Omega_t} (\|\mathbf{x}\| + 1)(\|\mathbf{x}\| + R) q(T-t, \mathbf{x}) \mathrm{d}\mathbf{x} \lesssim \mathbb{E}(\|\mathbf{Y}_t\| + R)^2 \lesssim R^2 + (T-t)d.$$

Reorganizing terms in (G.29), since we assume that $L \geq 1$ and $R \geq 1$, we have:

$$|J_3| \lesssim (t-t_k) \Big[ d^2 \frac{1}{(T-t)^2} + d \frac{1}{(T-t)^2} + d \frac{LR^2}{(T-t)^2} + d \frac{L}{T-t}$$

$$+ \frac{R^4}{(T-t)^4} + \frac{R^2}{(T-t)^3} + \frac{LR^2}{(T-t)^2} + \frac{LR^4}{(T-t)^3} \Big]$$

$$\lesssim (t-t_k) \Big[ d^2 \frac{1}{(T-t)^2} + d \frac{LR^2}{(T-t)^2} + d \frac{L}{T-t} + \frac{R^4}{(T-t)^4} + \frac{LR^4}{(T-t)^3} + \frac{LR^2}{(T-t)^2} \Big].$$
$$(G.30)$$

**Putting $J_3$ and $J_4$ Together.**
Combining (G.30) and (G.28) together, we have

$$|J_3| + |J_4| \lesssim (t-t_k) \Big[ \underbrace{d^2 \frac{1}{(T-t)^2}}_{C_1} + \underbrace{d \frac{LR^2}{(T-t)^2}}_{C_2} + \underbrace{d \frac{L}{T-t}}_{C_3} + \underbrace{\frac{R^4}{(T-t)^4}}_{C_4} + \underbrace{\frac{LR^4}{(T-t)^3}}_{C_5} + \underbrace{\frac{LR^2}{(T-t)^2}}_{C_6} \Big]$$

$$+ (t-t_k) \Big[ \underbrace{d \frac{1}{(T-t)^2}}_{C_7} + \underbrace{d \frac{R^2}{(T-t)^3}}_{C_8} + \underbrace{(1+L(T-t)) \frac{R^4}{(T-t)^4}}_{C_9} + \underbrace{\sqrt{d} \frac{LR^3}{(T-t)^{5/2}}}_{C_{10}} \Big]$$

$$\overset{(i)}{\lesssim} (t-t_k) \Big[ C_1 + C_2 + C_3 + C_4 + C_5 + C_8 + C_10 \Big]$$

$$= (t-t_k) \Big[ d^2 \frac{1}{(T-t)^2} + d \frac{LR^2}{(T-t)^2} + d \frac{L}{T-t} + \frac{R^4}{(T-t)^4}$$

$$+ \frac{LR^4}{(T-t)^3} + d \frac{R^2}{(T-t)^3} + \sqrt{d} \frac{LR^3}{(T-t)^{5/2}} \Big], \qquad (G.31)$$

where $(i)$ is due to $C_7 \leq C_2$, $C_9 = C_4 + C_5$, $C_6 \leq C_2$.
**Combining Everything Together.**
Plugging (G.17), (G.21), (G.22) and (G.31) into (G.13), and taking the integral, we can obtain the following inequality

$$\mathrm{TV}(\widehat{\mathbf{Y}}_{T-\delta}, \mathbf{Y}_{T-\delta}) \lesssim \mathrm{TV}(\pi_d, q_T)$$

$$+ \underbrace{\sum_k (t_{k+1} - t_k) \sqrt{\mathbb{E}_Q \operatorname{tr} \left( \nabla \mathbf{s}_\theta(T - t_k, \boldsymbol{Y}_{t_k}) - \nabla^2 \log q(T - t_k, \boldsymbol{Y}_{t_k}) \right)^2}}_{R_1}$$

$$+ \underbrace{\sum_k (t_{k+1} - t_k) \eta dL}_{R_2}$$

$$+ \underbrace{\sum_k (t_{k+1} - t_k) \sqrt{\frac{d}{T - t_{k+1}}} \sqrt{\mathbb{E}_Q \left\| \mathbf{s}_\theta(T - t_k, \boldsymbol{Y}_{t_k}) - \nabla \log q_{T - t_k}(\boldsymbol{Y}_{t_k}) \right\|^2}}_{R_3}$$

$$+ \underbrace{\sum_k (t_{k+1} - t_k) \sqrt{\frac{d}{T - t_{k+1}}} \eta \sqrt{L^2 R^2 + L^2 (T - t_k) d + c^2}}_{R_4}$$

$$+ \underbrace{\sum_k (t_{k+1} - t_k)^2 dL^2}_{R_5}$$

$$+ \sum_k (t_{k+1} - t_k)^2 \Big[ d^2 \frac{1}{(T - t)^2} + d \frac{LR^2}{(T - t)^2} + d \frac{L}{T - t} + \frac{R^4}{(T - t)^4}$$

$$+ \frac{LR^4}{(T - t)^3} + d \frac{R^2}{(T - t)^3} + \sqrt{d} \frac{LR^3}{(T - t)^{5/2}} \Big].$$

We further denote the last term by $R_6$. We know that $R_1$ is $\epsilon_{\mathrm{div}}$. For $R_3$, by Cauchy-Shwartz inequality, we have:

$$R_3 \le \sqrt{\sum_k (t_{k+1} - t_k) \frac{d}{T - t_{k+1}}} \sqrt{\sum_k (t_{k+1} - t_k) \mathbb{E}_Q \left\| \mathbf{s}_\theta(T - t_k, \boldsymbol{Y}_{t_k}) - \nabla \log q_{T - t_k}(\boldsymbol{Y}_{t_k}) \right\|^2}$$

$$\le \sqrt{\eta N d} \cdot \epsilon_{\mathrm{score}}.$$

Since we assume $L \ge 1$, for $R_2 + R_5$, we have

$$\sum_k (t_{k+1} - t_k) \eta dL + \sum_k (t_{k+1} - t_k)^2 dL^2 \lesssim \eta^2 N dL^2.$$

For the $R_4$, we have

$$\sum_k (t_{k+1} - t_k) \sqrt{\frac{d}{T - t_{k+1}}} \eta \sqrt{L^2 R^2 + L^2 (T - t_k) d + c^2}$$

$$\overset{(i)}{\lesssim} \eta \sum_k (t_{k+1} - t_k) \Big[ \sqrt{\frac{d}{T - t_{k+1}}} \sqrt{L^2 R^2} + \sqrt{L^2 (T - t_k) d} \Big]$$

$$\overset{(ii)}{\lesssim} \eta \sum_k \eta \sqrt{d} LR + (t - t_k) Ld$$

$$\le \eta^2 N (\sqrt{d} LR + Ld),$$

here we omit $c$ in $(i)$, and $(ii)$ is because $\frac{T - t_k}{T - t_{k+1}} \le 1 + \eta \le 2$. Hence we know that

$$R_2 + R_4 + R_5 \lesssim \eta^2 N \Big[ L^2 d + LR\sqrt{d} \Big].$$

Finally, for the $R_6$, we know that:

$$R_6 \le \eta^2 N \Big[ d^2 + LR^2 d + Ld + \frac{R^4}{\delta^2} + \frac{LR^4}{\delta} + \frac{R^2}{\delta} d + LR^3 \sqrt{\frac{d}{\delta}} \Big]$$

$$\lesssim \eta^2 N \Big[ d^2 + LR^2 d + \frac{R^2}{\delta} d + LR^3 \sqrt{\frac{d}{\delta}} + \frac{R^4}{\delta^2} + \frac{LR^4}{\delta} \Big].$$

Putting together, we know that

$$\mathrm{TV}(\widehat{\boldsymbol{Y}}_{T-\delta}, \boldsymbol{Y}_{T-\delta}) \lesssim \mathrm{TV}(\pi_d, q_T) + \epsilon_{\mathrm{div}} + \sqrt{\eta N d} \cdot \epsilon_{\mathrm{score}}$$

$$+ \eta^2 N \Big[ d^2 + LR^2 d + L^2 d + \frac{R^2}{\delta} d + LR^3 \sqrt{\frac{d}{\delta}} + \frac{R^4}{\delta^2} + \frac{LR^4}{\delta} \Big]$$

$$\overset{(iii)}{\lesssim} \mathrm{TV}(\pi_d, q_T) + \epsilon_{\mathrm{div}} + \sqrt{\eta N d} \cdot \epsilon_{\mathrm{score}} + \eta^2 N \Big[ LR^4 d^2 + L^2 d + \frac{LR^4}{\delta^2} \Big],$$

here $(iii)$ is due to $2LR^2 d \leq LR^4 d^2 + \frac{LR^4}{\delta^2}$, $2\frac{R^2}{\delta} d \leq LR^4 d^2 + \frac{LR^4}{\delta^2}$ and $LR^3 \sqrt{\frac{d}{\delta}} \leq LR^4 d^2 + \frac{LR^4}{\delta^2}$ (and $\delta < 1$ as well as $R \geq 1$ and $L \geq 1$). This completes the proof of Theorem 5.9. $\qquad\square$

### G.2 PROOF OF LEMMA 5.10 AND COROLLARY 5.11

*Proof of Lemma 5.10.* Since $q_{t|0}(\mathbf{x}_t | \mathbf{x}_0) = \mathcal{N}(\mathbf{x}_t; \mathbf{x}_0, t I_d)$, we have

$$\mathrm{KL}(q_{t|0}(\cdot | x_0) \| N(0, T\mathbf{I}_d)) = \frac{\|\mathbf{x}_0\|^2}{2T}. \tag{G.32}$$

By the convexity of the KL divergence,

$$\mathrm{KL}(q_T \| N(0, T\mathbf{I}_d)) = \mathrm{KL}\left( \int_{\mathbb{R}^d} q_{T|0}(\cdot | x_0) q_0(\mathrm{d}x_0) \| N(0, T\mathbf{I}_d) \right)$$

$$\leq \int_{\mathbb{R}^d} \mathrm{KL}(q_{T|0}(\cdot | x_0) \| \pi_d) q_0(\mathrm{d}x_0)$$

$$= \frac{1}{2T} \mathbb{E}_{q_0} \left[ \|X_0\|^2 \right]$$

$$= \frac{d}{2T}. \tag{G.33}$$

where we have used that $\mathbb{E}_{q_0}[\|\boldsymbol{X}_0\|^2] = d$, since $\mathrm{Cov}(q_0) = \boldsymbol{I}_d$. Applying Pinsker's inequality to (G.33), we have:

$$\mathrm{TV}\left( \boldsymbol{X}_0 + N(0, T\mathbf{I}_d), N(0, T\mathbf{I}_d) \right) \leq \frac{\sqrt{d}}{\sqrt{2T}}. \tag{G.34}$$

This completes the proof of Lemma 5.10. $\qquad\square$

*Proof of Corollary 5.11.* Please refer to Benton et al. (2024)'s Appendix D for a detailed derivation of the existence of such time schedule. Since we take $T = d/\epsilon^2$ and $N = LR^4 \Theta(d^2(T + \log(1/\delta))^2/\epsilon)$, using $\epsilon \leq 1/L$, we can easily show that $\eta \leq \min\{1/(12L^2 T d^2), 1/(24L^2 R^2 d)\}$. Hence by Theorem 5.5 and Lemma 5.6, we have:

$$\mathrm{TV}(\widehat{\boldsymbol{Y}}_{T-\delta}, \boldsymbol{Y}_{T-\delta}) \lesssim \frac{\sqrt{d}}{\sqrt{T}} + \epsilon_{\mathrm{div}} + \sqrt{d}\sqrt{\eta N}\epsilon_{\mathrm{score}} + \eta^2 N \Big[ LR^4 d^2 + L^2 d + LR^4 \frac{1}{\delta^2} \Big].$$

Since we take $\delta = 1/d$ and assume that $d \geq L/R^4$, we can further simply the bound of TV to:

$$\mathrm{TV}(\widehat{\boldsymbol{Y}}_{T-\delta}, \boldsymbol{Y}_{T-\delta}) \lesssim \frac{\sqrt{d}}{\sqrt{T}} + \epsilon_{\mathrm{div}} + \sqrt{d}\sqrt{\eta N}\epsilon_{\mathrm{score}} + \eta^2 N LR^4 d^2.$$

Since $T = d/\epsilon^2$ and $N = LR^4 \Theta(d^2(T + \log(1/\delta))^2/\epsilon)$, we know that

$$\frac{\sqrt{d}}{\sqrt{T}} \leq \epsilon,$$

$$\eta^N LR^4 d^2 = \frac{1}{N}(T + \log(1/\delta))^2 LR^4 d^2 \leq \epsilon.$$

Recall that we assume sufficiently small score estimation and divergence error, this completes the proof of Corollary 5.11. $\qquad\square$

## H  LIPSCHITZNESS OF SCORE FUNCTIONS

In this section, we consider the forward process by gradually adding noise to data distribution, i.e.,

$$\boldsymbol{X}_t = f(t)\boldsymbol{X}_{\text{data}} + g(t)\boldsymbol{Z}, \tag{H.1}$$

where $\boldsymbol{Z} \sim \mathcal{N}(0, I)$ is a standard normal random variable independent of $\boldsymbol{X}_{\text{data}}$, and $f(t)$ and $g(t)$ are scaling functions determined by the forward process that modulate the influence of the data and noise over time. Both VE and VP forward processes can be represented in this form. The distribution of $\boldsymbol{X}_t$ is denoted by $q(t, \mathbf{x})$. We will show the Lipschitzness property of $\nabla \log q(t, \mathbf{x})$ under Assumption 5.1.

### H.1  GENERAL RESULTS

The following lemma regards the upper bound of the derivatives concerning $\mathbf{x}$.

**Lemma H.1.** Suppose Assumption 5.1 holds. Let $\boldsymbol{X}_t$ be the forward process defined in (H.1), and its distribution is denoted by $q(t, \mathbf{x})$. We have the following inequalities:

$$\|\nabla \log q(t, \mathbf{x})\| \leq \frac{\|\mathbf{x}\| + f(t)R}{g(t)^2},$$

$$\left\|\nabla^2 \log q(t, \mathbf{x})\right\|_2 \leq \frac{1}{g(t)^2}\Big(1 + \frac{f(t)^2}{g(t)^2}2R^2\Big),$$

$$\left\|\nabla \text{tr}\big(\nabla^2 \log q(t, \mathbf{x})\big)\right\| \leq \frac{6f(t)^3}{g(t)^6}R^3.$$

*Proof of Lemma H.1.* We prove the lemma by gradually computing the higher-order derivatives of the score function step by step. A key point in our proof is that we bringing the terms involving $\mathbf{x}$ outside of the integral and canceling them, resulting in an expression where $\mathbf{x}$ only appears in the function's evaluation. Firstly, we derive the form of our score function $\nabla \ln q(t, \mathbf{x})$.

**Expression of the Score Function.**
To start with, using the expression of the forward process $\boldsymbol{X}_t$ in (H.1), we represent $q(t, \mathbf{x})$ as the convolution of the data distribution and Gaussian distribution. Specifically, we have:

$$
\begin{aligned}
q(t, \mathbf{x}) &= \int_{\mathbb{R}^d} p_{f(t)\boldsymbol{X}_0}(\mathbf{y}) p_{g(t)\boldsymbol{Z}}(\mathbf{x} - \mathbf{y}) \mathrm{d}\mathbf{y} \\
&= \int_{\mathbb{R}^d} \frac{1}{f(t)} p_{\boldsymbol{X}_0}\Big(\frac{\mathbf{y}}{f(t)}\Big) p_{g(t)\boldsymbol{Z}}(\mathbf{x} - \mathbf{y}) \mathrm{d}\mathbf{y} \\
&= \int_{\mathbb{R}^d} p_{\boldsymbol{X}_0}(\mathbf{y}) p_{g(t)\boldsymbol{Z}}\big(\mathbf{x} - f(t)\mathbf{y}\big) \mathrm{d}\mathbf{y} \\
&= \int_{\mathbb{R}^d} q_0(\mathbf{y}) p_{\boldsymbol{Z}}\Big(\frac{\mathbf{x} - f(t)\mathbf{y}}{g(t)}\Big) \frac{1}{g(t)} \mathrm{d}\mathbf{y} \\
&= \int_{\mathbb{R}^d} q_0(\mathbf{y}) \phi\Big(\frac{\mathbf{x} - f(t)\mathbf{y}}{g(t)}\Big) \frac{1}{g(t)} \mathrm{d}\mathbf{y}, \tag{H.2}
\end{aligned}
$$

where we represent the probability density function of standard Gaussian distribution as $\phi(\mathbf{x}) = (\mathrm{e}^{-\|\mathbf{x}\|^2/2})/(\sqrt{2\pi})^d$. Besides, in the second and fourth equations, we use the change of variable formula of the density function, i.e., $p_{a\boldsymbol{X}}(\mathbf{x}) = p_{\boldsymbol{X}}(\mathbf{x}/a)/a$, where $\boldsymbol{X}$ is a random variable and $a$ is a constant. By directly taking the gradient, we have the following property of $\phi(\cdot)$:

$$\nabla \phi(\mathbf{x}) = -\phi(\mathbf{x})\mathbf{x}.$$

Moreover, by the chain rule, we can calculate the gradient with respect to $\mathbf{x}$:

$$
\begin{aligned}
\nabla_{\mathbf{x}}\Big[\phi\Big(\frac{\mathbf{x} - f(t)\mathbf{y}}{g(t)}\Big)\Big] &= -\frac{1}{g(t)}\Big[\phi\Big(\frac{\mathbf{x} - f(t)\mathbf{y}}{g(t)}\Big)\frac{\mathbf{x} - f(t)\mathbf{y}}{g(t)}\Big] \\
&= -\frac{\mathbf{x} - f(t)\mathbf{y}}{g(t)^2}\phi\Big(\frac{\mathbf{x} - f(t)\mathbf{y}}{g(t)}\Big). \tag{H.3}
\end{aligned}
$$

Thus, we can express the score function as follows:

$$\nabla \log q(t, \mathbf{x}) = \frac{\nabla q(t, \mathbf{x})}{q(t, \mathbf{x})}$$

$$\overset{(i)}{=} \frac{\nabla \int_{\mathbb{R}^d} p_{\text{data}}(\mathbf{y})\phi\big(\frac{\mathbf{x}-f(t)\mathbf{y}}{g(t)}\big)\frac{1}{g(t)}\mathrm{d}\mathbf{y}}{\int_{\mathbb{R}^d} p_{\text{data}}(\mathbf{y})\phi\big(\frac{\mathbf{x}-f(t)\mathbf{y}}{g(t)}\big)\frac{1}{g(t)}\mathrm{d}\mathbf{y}}$$

$$= \frac{\int_{\mathbb{R}^d} p_{\text{data}}(\mathbf{y})\nabla_{\mathbf{x}}\big[\phi\big(\frac{\mathbf{x}-f(t)\mathbf{y}}{g(t)}\big)\big]\mathrm{d}\mathbf{y}}{\int_{\mathbb{R}^d} p_{\text{data}}(\mathbf{y})\phi\big(\frac{\mathbf{x}-f(t)\mathbf{y}}{g(t)}\big)\mathrm{d}\mathbf{y}}$$

$$\overset{(ii)}{=} \frac{\int_{\mathbb{R}^d} p_{\text{data}}(\mathbf{y})\big(-\frac{\mathbf{x}-f(t)\mathbf{y}}{g(t)^2}\big)\phi\big(\frac{\mathbf{x}-f(t)\mathbf{y}}{g(t)}\big)\mathrm{d}\mathbf{y}}{\int_{\mathbb{R}^d} p_{\text{data}}(\mathbf{y})\,\phi\big(\frac{\mathbf{x}-f(t)\mathbf{y}}{g(t)}\big)\mathrm{d}\mathbf{y}}$$

$$= -\frac{1}{g(t)^2}\frac{\int_{\mathbb{R}^d} p_{\text{data}}(\mathbf{y})\big(\mathbf{x}-f(t)\mathbf{y}\big)\phi\big(\frac{\mathbf{x}-f(t)\mathbf{y}}{g(t)}\big)\mathrm{d}\mathbf{y}}{\int_{\mathbb{R}^d} p_{\text{data}}(\mathbf{y})\phi\big(\frac{\mathbf{x}-f(t)\mathbf{y}}{g(t)}\big)\mathrm{d}\mathbf{y}}, \tag{H.4}$$

where $(i)$ holds due to (H.2), and $(ii)$ holds due to (H.3).

Under Assumption 5.1, we have $p_{\text{data}}(\mathbf{y}) = 0$ for $\|\mathbf{y}\| > R$. Thus, we have

$$\|\nabla \log q(t,\mathbf{x})\| = \frac{1}{g(t)^2}\frac{\big\|\int_{\mathbb{R}^d} p_{\text{data}}(\mathbf{y})\big(\mathbf{x}-f(t)\mathbf{y}\big)\phi\big(\frac{\mathbf{x}-f(t)\mathbf{y}}{g(t)}\big)\mathrm{d}\mathbf{y}\big\|}{\int_{\mathbb{R}^d} p_{\text{data}}(\mathbf{y})\phi\big(\frac{\mathbf{x}-f(t)\mathbf{y}}{g(t)}\big)\mathrm{d}\mathbf{y}}$$

$$\le \frac{1}{g(t)^2}\frac{\int_{\mathbb{R}^d} p_{\text{data}}(\mathbf{y})\|\mathbf{x}-f(t)\mathbf{y}\|\phi\big(\frac{\mathbf{x}-f(t)\mathbf{y}}{g(t)}\big)\mathrm{d}\mathbf{y}}{\int_{\mathbb{R}^d} p_{\text{data}}(\mathbf{y})\phi\big(\frac{\mathbf{x}-f(t)\mathbf{y}}{g(t)}\big)\mathrm{d}\mathbf{y}}$$

$$\le \frac{1}{g(t)^2}\frac{\int_{\mathbb{R}^d} p_{\text{data}}(\mathbf{y})\big(\|\mathbf{x}\|+f(t)\|\mathbf{y}\|\big)\phi\big(\frac{\mathbf{x}-f(t)\mathbf{y}}{g(t)}\big)\mathrm{d}\mathbf{y}}{\int_{\mathbb{R}^d} p_{\text{data}}(\mathbf{y})\phi\big(\frac{\mathbf{x}-f(t)\mathbf{y}}{g(t)}\big)\mathrm{d}\mathbf{y}}$$

$$\le \frac{\|\mathbf{x}\|+f(t)R}{g(t)^2}.$$

This shows that, under Assumption 5.1, the norm of the score function can be bounded by a linear function.

**First-order Derivative of the Score Function.**

We proceed by computing the Jacobian matrix of $\nabla \log q(t,\mathbf{x})$. Directly calculating the gradient of (H.6), we have

$$\nabla^2 \log q(t,\mathbf{x}) = \nabla\bigg[-\frac{1}{g(t)^2}\frac{\int_{\mathbb{R}^d} p_{\text{data}}(\mathbf{y})(\mathbf{x}-f(t)\mathbf{y})\phi\big(\frac{\mathbf{x}-f(t)\mathbf{y}}{g(t)}\big)\mathrm{d}\mathbf{y}}{\int_{\mathbb{R}^d} p_{\text{data}}(\mathbf{y})\phi\big(\frac{\mathbf{x}-f(t)\mathbf{y}}{g(t)}\big)\mathrm{d}\mathbf{y}}\bigg]$$

$$= -\frac{1}{g(t)^2}\frac{1}{\Phi(\mathbf{x})^2}\bigg[\underbrace{\nabla\bigg(\int_{\mathbb{R}^d} p_{\text{data}}(\mathbf{y})(\mathbf{x}-f(t)\mathbf{y})\phi\big(\frac{\mathbf{x}-f(t)\mathbf{y}}{g(t)}\big)\mathrm{d}\mathbf{y}\bigg)\Phi(\mathbf{x})}_{I_1}$$

$$\underbrace{-\int_{\mathbb{R}^d} p_{\text{data}}(\mathbf{y})(\mathbf{x}-f(t)\mathbf{y})\phi\big(\frac{\mathbf{x}-f(t)\mathbf{y}}{g(t)}\big)\mathrm{d}\mathbf{y}\bigg(\nabla\int_{\mathbb{R}^d} p_{\text{data}}(\mathbf{y})\phi\big(\frac{\mathbf{x}-f(t)\mathbf{y}}{g(t)}\big)\mathrm{d}\mathbf{y}\bigg)^{\top}}_{I_2}\bigg], \tag{H.5}$$

where we use the shorthand expression $\Phi(x) = \int_{\mathbb{R}^d} p_{\text{data}}(\mathbf{y})\phi\big(\frac{\mathbf{x}-f(t)\mathbf{y}}{g(t)}\big)\mathrm{d}\mathbf{y}$, and the last equation holds due to the gradient formula of vector-valued functions, i.e., $\nabla\frac{f(\mathbf{x})}{g(\mathbf{x})} = \frac{\nabla f(\mathbf{x})}{g(\mathbf{x})} - \frac{f(\mathbf{x})\nabla g(\mathbf{x})^{\top}}{g(\mathbf{x})^2}$.

Firstly, we compute $I_1$:

$$I_1 = \nabla\bigg(\int_{\mathbb{R}^d} p_{\text{data}}(\mathbf{y})(\mathbf{x}-f(t)\mathbf{y})\phi\big(\frac{\mathbf{x}-f(t)\mathbf{y}}{g(t)}\big)\mathrm{d}\mathbf{y}\bigg)$$

$$= \int_{\mathbb{R}^d} p_{\text{data}}(\mathbf{y})\bigg(\phi\big(\frac{\mathbf{x}-f(t)\mathbf{y}}{g(t)}\big)\mathbf{I} + (\mathbf{x}-f(t)\mathbf{y})\nabla_{\mathbf{x}}\big[\phi\big(\frac{\mathbf{x}-f(t)\mathbf{y}}{g(t)}\big)\big]^{\top}\bigg)\mathrm{d}\mathbf{y}$$

$$\overset{(i)}{=} \int_{\mathbb{R}^d} p_{\text{data}}(\mathbf{y})\bigg(\phi\mathbf{I} + (\mathbf{x}-f(t)\mathbf{y})\big(-\phi\frac{\mathbf{x}-f(t)\mathbf{y}}{g(t)^2}\big)^{\top}\bigg)\mathrm{d}\mathbf{y}$$

$$= \int_{\mathbb{R}^d} p_{\text{data}}(\mathbf{y}) \Big( \mathbf{I} - \frac{(\mathbf{x} - f(t)\mathbf{y})(\mathbf{x} - f(t)\mathbf{y})^\top}{g(t)^2} \Big) \phi \mathrm{d}\mathbf{y}$$

$$= \underbrace{\int_{\mathbb{R}^d} p_{\text{data}}(\mathbf{y}) \phi \mathrm{d}\mathbf{y} \cdot \mathbf{I} - \frac{\mathbf{x}\mathbf{x}^\top}{g(t)^2} \int_{\mathbb{R}^d} p_{\text{data}}(\mathbf{y}) \phi \mathrm{d}\mathbf{y}}_{I_{1,1}} + \underbrace{\frac{f(t)}{g(t)^2} \mathbf{x} \int_{\mathbb{R}^d} p_{\text{data}}(\mathbf{y}) \mathbf{y}^\top \phi \mathrm{d}\mathbf{y}}_{I_{1,2}}$$

$$+ \underbrace{\frac{f(t)}{g(t)^2} \Big( \int_{\mathbb{R}^d} p_{\text{data}}(\mathbf{y}) \mathbf{y} \phi \mathrm{d}\mathbf{y} \Big) \mathbf{x}^\top}_{I_{1,3}} - \frac{f(t)^2}{g(t)^2} \int_{\mathbb{R}^d} p_{\text{data}}(\mathbf{y}) \mathbf{y}\mathbf{y}^\top \phi \mathrm{d}\mathbf{y}, \tag{H.6}$$

where we use the shorthand expression $\phi = \phi\big(\frac{\mathbf{x} - f(t)\mathbf{y}}{g(t)}\big)$ and $(i)$ holds due to (H.3).

Next, we compute $I_2$:

$$I_2 = \Big[ \int_{\mathbb{R}^d} p_{\text{data}}(\mathbf{y})(\mathbf{x} - f(t)\mathbf{y}) \phi\Big(\frac{\mathbf{x} - f(t)\mathbf{y}}{g(t)}\Big) \mathrm{d}\mathbf{y} \Big] \Big[ \nabla \int_{\mathbb{R}^d} p_{\text{data}}(\mathbf{y}) \phi\Big(\frac{\mathbf{x} - f(t)\mathbf{y}}{g(t)}\Big) \mathrm{d}\mathbf{y} \Big]^\top$$

$$= \Big[ \int_{\mathbb{R}^d} p_{\text{data}}(\mathbf{y})(\mathbf{x} - f(t)\mathbf{y}) \phi\Big(\frac{\mathbf{x} - f(t)\mathbf{y}}{g(t)}\Big) \mathrm{d}\mathbf{y} \Big] \Big[ \int_{\mathbb{R}^d} p_{\text{data}}(\mathbf{y}) \Big( - \frac{\mathbf{x} - f(t)\mathbf{y}}{g(t)^2} \Big) \phi\Big(\frac{\mathbf{x} - f(t)\mathbf{y}}{g(t)}\Big) \mathrm{d}\mathbf{y} \Big]^\top$$

$$= -\frac{1}{g(t)^2} \Big[ \int_{\mathbb{R}^d} p_{\text{data}}(\mathbf{y})(\mathbf{x} - f(t)\mathbf{y}) \phi \mathrm{d}\mathbf{y} \Big] \Big[ \int_{\mathbb{R}^d} p_{\text{data}}(\mathbf{y})(\mathbf{x} - f(t)\mathbf{y})^\top \phi \mathrm{d}\mathbf{y} \Big]$$

$$= \underbrace{-\frac{\mathbf{x}\mathbf{x}^\top}{g(t)^2} \int_{\mathbb{R}^d} p_{\text{data}}(\mathbf{y}) \phi \mathrm{d}\mathbf{y} \cdot \int_{\mathbb{R}^d} p_{\text{data}}(\mathbf{y}) \phi \mathrm{d}\mathbf{y}}_{I_{2,1}} + \underbrace{\frac{f(t)}{g(t)^2} \mathbf{x} \int_{\mathbb{R}^d} p_{\text{data}}(\mathbf{y}) \mathbf{y}^\top \phi \mathrm{d}\mathbf{y} \cdot \int_{\mathbb{R}^d} p_{\text{data}}(\mathbf{y}) \phi \mathrm{d}\mathbf{y}}_{I_{2,2}}$$

$$+ \underbrace{\frac{f(t)}{g(t)^2} \Big( \int_{\mathbb{R}^d} p_{\text{data}}(\mathbf{y}) \mathbf{y} \phi \mathrm{d}\mathbf{y} \Big) \mathbf{x}^\top \int_{\mathbb{R}^d} p_{\text{data}}(\mathbf{y}) \phi \mathrm{d}\mathbf{y}}_{I_{2,3}} - \frac{f(t)^2}{g(t)^2} \int_{\mathbb{R}^d} p_{\text{data}}(\mathbf{y}) \mathbf{y} \phi \mathrm{d}\mathbf{y} \int_{\mathbb{R}^d} p_{\text{data}}(\mathbf{y}) \mathbf{y}^\top \phi \mathrm{d}\mathbf{y},$$

$$\tag{H.7}$$

where the second inequality holds due to (H.3), the third inequality holds due to the shorthand expression $\phi = \phi\big(\frac{\mathbf{x} - f(t)\mathbf{y}}{g(t)}\big)$. Substituting (H.6) and (H.7) into (H.5), notice that $\Phi(\mathbf{x}) = \int_{\mathbb{R}^d} p_{\text{data}}(\mathbf{y}) \phi\big(\frac{\mathbf{x} - f(t)\mathbf{y}}{g(t)}\big) \mathrm{d}\mathbf{y}$, $I_{1,1} = I_{2,1}$, $I_{1,2} = I_{2,2}$ and $I_{1,3} = I_{2,3}$. Thus, we have the following equation:

$$\nabla^2 \log q(t, \mathbf{x}) = -\frac{1}{g(t)^2} \Big[ \mathbf{I} - \frac{f(t)^2}{g(t)^2} \Big( \frac{\int_{\mathbb{R}^d} p_{\text{data}}(\mathbf{y}) \mathbf{y}\mathbf{y}^\top \phi \mathrm{d}\mathbf{y}}{\int_{\mathbb{R}^d} p_{\text{data}}(\mathbf{y}) \phi \mathrm{d}\mathbf{y}}$$
$$- \frac{\big[ \int_{\mathbb{R}^d} p_{\text{data}}(\mathbf{y}) \mathbf{y} \phi \mathrm{d}\mathbf{y} \big] \big[ \int_{\mathbb{R}^d} p_{\text{data}}(\mathbf{y}) \mathbf{y}^\top \phi \mathrm{d}\mathbf{y} \big]}{\big( \int_{\mathbb{R}^d} p_{\text{data}}(\mathbf{y}) \phi \mathrm{d}\mathbf{y} \big)^2} \Big) \Big]. \tag{H.8}$$

Using Assumption 5.1, when $p_{\text{data}}(\mathbf{y}) \neq 0$, we have $\|\mathbf{y}\mathbf{y}^\top\|_2 = \|\mathbf{y}\|^2 \leq R^2$. Moreover, we have

$$\Big\| \int_{\mathbb{R}^d} p_{\text{data}}(\mathbf{y}) \mathbf{y} \phi \mathrm{d}\mathbf{y} \cdot \int_{\mathbb{R}^d} p_{\text{data}}(\mathbf{y}) \mathbf{y}^\top \phi \mathrm{d}\mathbf{y} \Big\|_2 = \Big\| \int_{\mathbb{R}^d} p_{\text{data}}(\mathbf{y}) \mathbf{y} \phi \mathrm{d}\mathbf{y} \Big\|^2$$

$$\leq \Big( \int_{\mathbb{R}^d} p_{\text{data}}(\mathbf{y}) \|\mathbf{y}\| \phi \mathrm{d}\mathbf{y} \Big)^2$$

$$\leq R^2 \Big( \int_{\mathbb{R}^d} p_{\text{data}}(\mathbf{y}) \phi \mathrm{d}\mathbf{y} \Big)^2.$$

Substituting into (H.8), we have

$$\big\| \nabla^2 \log q(t, \mathbf{x}) \big\|_2 \leq \frac{1}{g(t)^2} \Big( 1 + \frac{f(t)^2}{g(t)^2} 2R^2 \Big).$$

This implies that $\nabla \log q(t, \mathbf{x})$ is Lipschitz.

**Derivatives of the Divergence of the Score Function.**

Next, we consider the divergence of the score function, i.e.,

$$\nabla \cdot [\nabla \log q(t, \mathbf{x})] = \text{tr}\big( \nabla^2 \log q(t, \mathbf{x}) \big).$$

To start with, using (H.8), we have:

$$\text{tr}\big(\nabla^2 \log q(t, \mathbf{x})\big) = -\frac{d}{g(t)^2} + \frac{f(t)^2}{g(t)^4} \left( \frac{\int_{\mathbb{R}^d} p_{\text{data}}(\mathbf{y})\|\mathbf{y}\|^2 \phi \mathrm{d}\mathbf{y}}{\int_{\mathbb{R}^d} p_{\text{data}}(\mathbf{y})\phi \mathrm{d}\mathbf{y}} - \frac{\|\int_{\mathbb{R}^d} p_{\text{data}}(\mathbf{y})\mathbf{y}\phi \mathrm{d}\mathbf{y}\|^2}{\left(\int_{\mathbb{R}^d} p_{\text{data}}(\mathbf{y})\phi \mathrm{d}\mathbf{y}\right)^2} \right), \quad \text{(H.9)}$$

where we use the fact $\text{tr}(\mathbf{x}\mathbf{x}^\top) = \|\mathbf{x}\|^2$. Directly computing the gradient, we have

$$\nabla \text{tr}\big(\nabla^2 \log q(t, \mathbf{x})\big) = \frac{f(t)^2}{g(t)^4} \bigg[ \underbrace{\nabla \frac{\int_{\mathbb{R}^d} p_{\text{data}}(\mathbf{y})\|\mathbf{y}\|^2 \phi \mathrm{d}\mathbf{y}}{\int_{\mathbb{R}^d} p_{\text{data}}(\mathbf{y})\phi \mathrm{d}\mathbf{y}}}_{J_1} - \underbrace{\nabla \frac{\|\int_{\mathbb{R}^d} p_{\text{data}}(\mathbf{y})\mathbf{y}\phi \mathrm{d}\mathbf{y}\|^2}{\left(\int_{\mathbb{R}^d} p_{\text{data}}(\mathbf{y})\phi \mathrm{d}\mathbf{y}\right)^2}}_{J_2} \bigg].$$

Firstly, for $J_1$, we have

$$J_1 = \frac{\nabla \int_{\mathbb{R}^d} p_{\text{data}}(\mathbf{y})\|\mathbf{y}\|^2 \phi \mathrm{d}\mathbf{y} \cdot \int_{\mathbb{R}^d} p_{\text{data}}(\mathbf{y})\phi \mathrm{d}\mathbf{y} - \nabla \int_{\mathbb{R}^d} p_{\text{data}}(\mathbf{y})\phi \mathrm{d}\mathbf{y} \cdot \int_{\mathbb{R}^d} p_{\text{data}}(\mathbf{y})\|\mathbf{y}\|^2 \phi \mathrm{d}\mathbf{y}}{\left(\int_{\mathbb{R}^d} p_{\text{data}}(\mathbf{y})\phi \mathrm{d}\mathbf{y}\right)^2}.$$
$$\text{(H.10)}$$

We calculate the two gradients separately. First, we have

$$\begin{aligned}
\nabla \int_{\mathbb{R}^d} p_{\text{data}}(\mathbf{y})\|\mathbf{y}\|^2 \phi \mathrm{d}\mathbf{y} &= \int_{\mathbb{R}^d} p_{\text{data}}(\mathbf{y})\|\mathbf{y}\|^2 \nabla \phi \mathrm{d}\mathbf{y} \\
&= \int_{\mathbb{R}^d} p_{\text{data}}(\mathbf{y})\|\mathbf{y}\|^2 \Big( -\frac{\mathbf{x} - f(t)\mathbf{y}}{g(t)^2} \Big) \phi \mathrm{d}\mathbf{y} \\
&= -\frac{\mathbf{x}}{g(t)^2} \int_{\mathbb{R}^d} p_{\text{data}}(\mathbf{y})\|\mathbf{y}\|^2 \phi \mathrm{d}\mathbf{y} + \frac{f(t)}{g(t)^2} \int_{\mathbb{R}^d} p_{\text{data}}(\mathbf{y})\|\mathbf{y}\|^2 \mathbf{y}\phi \mathrm{d}\mathbf{y},
\end{aligned}$$
$$\text{(H.11)}$$

where the second equality holds due to (H.3). Next, we have

$$\begin{aligned}
\nabla \int_{\mathbb{R}^d} p_{\text{data}}(\mathbf{y})\phi \mathrm{d}\mathbf{y} &= \int_{\mathbb{R}^d} p_{\text{data}}(\mathbf{y})\nabla \phi \mathrm{d}\mathbf{y} \\
&= \int_{\mathbb{R}^d} p_{\text{data}}(\mathbf{y}) \Big( -\frac{\mathbf{x} - f(t)\mathbf{y}}{g(t)^2} \Big) \phi \mathrm{d}\mathbf{y} \\
&= -\frac{\mathbf{x}}{g(t)^2} \int_{\mathbb{R}^d} p_{\text{data}}(\mathbf{y})\phi \mathrm{d}\mathbf{y} + \frac{f(t)}{g(t)^2} \int_{\mathbb{R}^d} p_{\text{data}}(\mathbf{y})\mathbf{y}\phi \mathrm{d}\mathbf{y}.
\end{aligned}$$
$$\text{(H.12)}$$

Substituting (H.11) and (H.12) into (H.10), we have

$$\begin{aligned}
J_1 &= \frac{\Big( -\frac{\mathbf{x}}{g(t)^2} \int_{\mathbb{R}^d} p_{\text{data}}(\mathbf{y})\|\mathbf{y}\|^2 \phi \mathrm{d}\mathbf{y} + \frac{f(t)}{g(t)^2} \int_{\mathbb{R}^d} p_{\text{data}}(\mathbf{y})\|\mathbf{y}\|^2 \mathbf{y}\phi \mathrm{d}\mathbf{y} \Big) \cdot \int_{\mathbb{R}^d} p_{\text{data}}(\mathbf{y})\phi \mathrm{d}\mathbf{y}}{\left(\int_{\mathbb{R}^d} p_{\text{data}}(\mathbf{y})\phi \mathrm{d}\mathbf{y}\right)^2} \\
&\quad - \frac{\Big( -\frac{\mathbf{x}}{g(t)^2} \int_{\mathbb{R}^d} p_{\text{data}}(\mathbf{y})\phi \mathrm{d}\mathbf{y} + \frac{f(t)}{g(t)^2} \int_{\mathbb{R}^d} p_{\text{data}}(\mathbf{y})\mathbf{y}\phi \mathrm{d}\mathbf{y} \Big) \cdot \int_{\mathbb{R}^d} p_{\text{data}}(\mathbf{y})\|\mathbf{y}\|^2 \phi \mathrm{d}\mathbf{y}}{\left(\int_{\mathbb{R}^d} p_{\text{data}}(\mathbf{y})\phi \mathrm{d}\mathbf{y}\right)^2} \\
&= \frac{f(t)}{g(t)^2} \left( \frac{\int_{\mathbb{R}^d} p_{\text{data}}(\mathbf{y})\|\mathbf{y}\|^2 \mathbf{y}\phi \mathrm{d}\mathbf{y}}{\int_{\mathbb{R}^d} p_{\text{data}}(\mathbf{y})\phi \mathrm{d}\mathbf{y}} - \frac{\int_{\mathbb{R}^d} p_{\text{data}}(\mathbf{y})\mathbf{y}\phi \mathrm{d}\mathbf{y} \int_{\mathbb{R}^d} p_{\text{data}}(\mathbf{y})\|\mathbf{y}\|^2 \phi \mathrm{d}\mathbf{y}}{\left(\int_{\mathbb{R}^d} p_{\text{data}}(\mathbf{y})\phi \mathrm{d}\mathbf{y}\right)^2} \right).
\end{aligned}$$

Next, for $J_2$, we have:

$$J_2 = \frac{\nabla \|\int_{\mathbb{R}^d} p_{\text{data}}(\mathbf{y})\mathbf{y}\phi \mathrm{d}\mathbf{y}\|^2 \cdot \left(\int_{\mathbb{R}^d} p_{\text{data}}(\mathbf{y})\phi \mathrm{d}\mathbf{y}\right)^2 - \nabla\left(\int_{\mathbb{R}^d} p_{\text{data}}(\mathbf{y})\phi \mathrm{d}\mathbf{y}\right)^2 \cdot \|\int_{\mathbb{R}^d} p_{\text{data}}(\mathbf{y})\mathbf{y}\phi \mathrm{d}\mathbf{y}\|^2}{\left(\int_{\mathbb{R}^d} p_{\text{data}}(\mathbf{y})\phi \mathrm{d}\mathbf{y}\right)^4}.$$
$$\text{(H.13)}$$

Again, we calculate the two gradients as follows. First, we have:

$$\nabla \left\| \int_{\mathbb{R}^d} p_{\text{data}}(\mathbf{y})\mathbf{y}\phi \mathrm{d}\mathbf{y} \right\|^2 = \nabla \sum_{i=1}^d \left( \int_{\mathbb{R}^d} p_{\text{data}}(\mathbf{y})y_i \phi \mathrm{d}\mathbf{y} \right)^2$$

$$= \sum_{i=1}^{d} 2 \int_{\mathbb{R}^d} p_{\text{data}}(\mathbf{y}) y_i \phi \mathrm{d}\mathbf{y} \cdot \int_{\mathbb{R}^d} p_{\text{data}}(\mathbf{y}) y_i \nabla \phi \mathrm{d}\mathbf{y}$$

$$\stackrel{(i)}{=} \sum_{i=1}^{d} 2 \int_{\mathbb{R}^d} p_{\text{data}}(\mathbf{y}) y_i \phi \mathrm{d}\mathbf{y} \cdot \int_{\mathbb{R}^d} p_{\text{data}}(\mathbf{y}) y_i \left( -\frac{\mathbf{x} - f(t)\mathbf{y}}{g(t)^2} \right) \phi \mathrm{d}\mathbf{y}$$

$$= -\frac{2\mathbf{x}}{g(t)^2} \sum_{i=1}^{d} \left( \int_{\mathbb{R}^d} p_{\text{data}}(\mathbf{y}) y_i \phi \mathrm{d}\mathbf{y} \right)^2 + \frac{2f(t)}{g(t)^2} \sum_{i=1}^{d} \int_{\mathbb{R}^d} p_{\text{data}}(\mathbf{y}) y_i \phi \mathrm{d}\mathbf{y} \int_{\mathbb{R}^d} p_{\text{data}}(\mathbf{y}) y_i \mathbf{y} \phi \mathrm{d}\mathbf{y}$$

$$\stackrel{(ii)}{=} -\frac{2\mathbf{x}}{g(t)^2} \left\| \int_{\mathbb{R}^d} p_{\text{data}}(\mathbf{y}) \mathbf{y} \phi \mathrm{d}\mathbf{y} \right\|^2 + \frac{2f(t)}{g(t)^2} \int_{\mathbb{R}^d} p_{\text{data}}(\mathbf{y}) \mathbf{y} \mathbf{y}^\top \phi \mathrm{d}\mathbf{y} \cdot \int_{\mathbb{R}^d} p_{\text{data}}(\mathbf{y}) \mathbf{y} \phi \mathrm{d}\mathbf{y}, \quad \text{(H.14)}$$

where $(i)$ holds due to (H.3), and $(ii)$ holds due to the coordinate expression of matrix multiplication. Moreover, we have

$$\nabla \left( \int_{\mathbb{R}^d} p_{\text{data}}(\mathbf{y}) \phi \mathrm{d}\mathbf{y} \right)^2 = 2 \int_{\mathbb{R}^d} p_{\text{data}}(\mathbf{y}) \phi \mathrm{d}\mathbf{y} \cdot \nabla \int_{\mathbb{R}^d} p_{\text{data}}(\mathbf{y}) \phi \mathrm{d}\mathbf{y}$$

$$\stackrel{(i)}{=} 2 \int_{\mathbb{R}^d} p_{\text{data}}(\mathbf{y}) \phi \mathrm{d}\mathbf{y} \cdot \left( -\frac{\mathbf{x}}{g(t)^2} \int_{\mathbb{R}^d} p_{\text{data}}(\mathbf{y}) \phi \mathrm{d}\mathbf{y} + \frac{f(t)}{g(t)^2} \int_{\mathbb{R}^d} p_{\text{data}}(\mathbf{y}) \mathbf{y} \phi \mathrm{d}\mathbf{y} \right)$$

$$= -\frac{2\mathbf{x}}{g(t)^2} \left( \int_{\mathbb{R}^d} p_{\text{data}}(\mathbf{y}) \phi \mathrm{d}\mathbf{y} \right)^2 + \frac{2f(t)}{g(t)^2} \int_{\mathbb{R}^d} p_{\text{data}}(\mathbf{y}) \phi \mathrm{d}\mathbf{y} \cdot \int_{\mathbb{R}^d} p_{\text{data}}(\mathbf{y}) \mathbf{y} \phi \mathrm{d}\mathbf{y},$$
$$\text{(H.15)}$$

where $(i)$ holds due to (H.12). Substituting (H.14) and (H.15) into (H.13), we have

$$J_2 = \frac{\left( -\frac{2\mathbf{x}}{g(t)^2} \| \int_{\mathbb{R}^d} p_{\text{data}}(\mathbf{y}) \mathbf{y} \phi \mathrm{d}\mathbf{y} \|^2 + \frac{2f(t)}{g(t)^2} \int_{\mathbb{R}^d} p_{\text{data}}(\mathbf{y}) \mathbf{y} \mathbf{y}^\top \phi \mathrm{d}\mathbf{y} \int_{\mathbb{R}^d} p_{\text{data}}(\mathbf{y}) \mathbf{y} \phi \mathrm{d}\mathbf{y} \right) \left( \int_{\mathbb{R}^d} p_{\text{data}}(\mathbf{y}) \phi \mathrm{d}\mathbf{y} \right)^2}{\left( \int_{\mathbb{R}^d} p_{\text{data}}(\mathbf{y}) \phi \mathrm{d}\mathbf{y} \right)^4}$$

$$- \frac{\left( -\frac{2\mathbf{x}}{g(t)^2} \left( \int_{\mathbb{R}^d} p_{\text{data}}(\mathbf{y}) \phi \mathrm{d}\mathbf{y} \right)^2 + \frac{2f(t)}{g(t)^2} \int_{\mathbb{R}^d} p_{\text{data}}(\mathbf{y}) \phi \mathrm{d}\mathbf{y} \int_{\mathbb{R}^d} p_{\text{data}}(\mathbf{y}) \mathbf{y} \phi \mathrm{d}\mathbf{y} \right) \cdot \| \int_{\mathbb{R}^d} p_{\text{data}}(\mathbf{y}) \mathbf{y} \phi \mathrm{d}\mathbf{y} \|^2}{\left( \int_{\mathbb{R}^d} p_{\text{data}}(\mathbf{y}) \phi \mathrm{d}\mathbf{y} \right)^4}$$

$$= \frac{2f(t)}{g(t)^2} \left( \frac{\int_{\mathbb{R}^d} p_{\text{data}}(\mathbf{y}) \mathbf{y} \mathbf{y}^\top \phi \mathrm{d}\mathbf{y} \int_{\mathbb{R}^d} p_{\text{data}}(\mathbf{y}) \mathbf{y} \phi \mathrm{d}\mathbf{y}}{\left( \int_{\mathbb{R}^d} p_{\text{data}}(\mathbf{y}) \phi \mathrm{d}\mathbf{y} \right)^2} + \frac{\int_{\mathbb{R}^d} p_{\text{data}}(\mathbf{y}) \mathbf{y} \phi \mathrm{d}\mathbf{y} \cdot \| \int_{\mathbb{R}^d} p_{\text{data}}(\mathbf{y}) \mathbf{y} \phi \mathrm{d}\mathbf{y} \|^2}{\left( \int_{\mathbb{R}^d} p_{\text{data}}(\mathbf{y}) \phi \mathrm{d}\mathbf{y} \right)^3} \right).$$

We can conclude with

$$\|J_1\| \leq \frac{f(t)}{g(t)^2} \left( \frac{\int_{\mathbb{R}^d} p_{\text{data}}(\mathbf{y}) \|\mathbf{y}\|^2 \|\mathbf{y}\| \phi \mathrm{d}\mathbf{y}}{\int_{\mathbb{R}^d} p_{\text{data}}(\mathbf{y}) \phi \mathrm{d}\mathbf{y}} - \frac{\int_{\mathbb{R}^d} p_{\text{data}}(\mathbf{y}) \|\mathbf{y}\| \phi \mathrm{d}\mathbf{y} \int_{\mathbb{R}^d} p_{\text{data}}(\mathbf{y}) \|\mathbf{y}\|^2 \phi \mathrm{d}\mathbf{y}}{\left( \int_{\mathbb{R}^d} p_{\text{data}}(\mathbf{y}) \phi \mathrm{d}\mathbf{y} \right)^2} \right)$$

$$\leq \frac{f(t)}{g(t)^2} \left( R^3 + R \cdot R^2 \right)$$

$$= \frac{2f(t)}{g(t)^2} R^3.$$

Moreover, we have

$$\|J_2\| \leq \frac{2f(t)}{g(t)^2} \left( \frac{\| \int_{\mathbb{R}^d} p_{\text{data}}(\mathbf{y}) \mathbf{y} \mathbf{y}^\top \phi \mathrm{d}\mathbf{y} \|_2 \| \int_{\mathbb{R}^d} p_{\text{data}}(\mathbf{y}) \mathbf{y} \phi \mathrm{d}\mathbf{y} \|}{\left( \int_{\mathbb{R}^d} p_{\text{data}}(\mathbf{y}) \phi \mathrm{d}\mathbf{y} \right)^2} \right.$$

$$+ \left. \frac{\int_{\mathbb{R}^d} p_{\text{data}}(\mathbf{y}) \|\mathbf{y}\| \phi \mathrm{d}\mathbf{y} \cdot \| \int_{\mathbb{R}^d} p_{\text{data}}(\mathbf{y}) \mathbf{y} \phi \mathrm{d}\mathbf{y} \|^2}{\left( \int_{\mathbb{R}^d} p_{\text{data}}(\mathbf{y}) \phi \mathrm{d}\mathbf{y} \right)^3} \right)$$

$$\leq \frac{2f(t)}{g(t)^2} \left( \frac{\int_{\mathbb{R}^d} p_{\text{data}}(\mathbf{y}) \|\mathbf{y} \mathbf{y}^\top\|_2 \phi \mathrm{d}\mathbf{y} \int_{\mathbb{R}^d} p_{\text{data}}(\mathbf{y}) \|\mathbf{y}\| \phi \mathrm{d}\mathbf{y}}{\left( \int_{\mathbb{R}^d} p_{\text{data}}(\mathbf{y}) \phi \mathrm{d}\mathbf{y} \right)^2} \right.$$

$$+ \left. \frac{\int_{\mathbb{R}^d} p_{\text{data}}(\mathbf{y}) \|\mathbf{y}\| \phi \mathrm{d}\mathbf{y} \cdot \left( \int_{\mathbb{R}^d} p_{\text{data}}(\mathbf{y}) \|\mathbf{y}\| \phi \mathrm{d}\mathbf{y} \right)^2}{\left( \int_{\mathbb{R}^d} p_{\text{data}}(\mathbf{y}) \phi \mathrm{d}\mathbf{y} \right)^3} \right)$$

$$\overset{(i)}{=} \frac{2f(t)}{g(t)^2}\left( \frac{\int_{\mathbb{R}^d} p_{\text{data}}(\mathbf{y})\|\mathbf{y}\|^2\phi \mathrm{d}\mathbf{y} \int_{\mathbb{R}^d} p_{\text{data}}(\mathbf{y})\|\mathbf{y}\|\phi \mathrm{d}\mathbf{y}}{\left( \int_{\mathbb{R}^d} p_{\text{data}}(\mathbf{y})\phi \mathrm{d}\mathbf{y} \right)^2} \right.$$

$$\left. + \frac{\int_{\mathbb{R}^d} p_{\text{data}}(\mathbf{y})\|\mathbf{y}\|\phi \mathrm{d}\mathbf{y} \cdot \left( \int_{\mathbb{R}^d} p_{\text{data}}(\mathbf{y})\|\mathbf{y}\|\phi \mathrm{d}\mathbf{y} \right)^2}{\left( \int_{\mathbb{R}^d} p_{\text{data}}(\mathbf{y})\phi \mathrm{d}\mathbf{y} \right)^3} \right)$$

$$\overset{(i)}{\leq} \frac{2f(t)}{g(t)^2}\left( R^2 \cdot R + R \cdot R^2 \right) = \frac{4f(t)}{g(t)^2}R^3,$$

where $(i)$ holds due to the fact $\|\mathbf{y}\mathbf{y}^\top\|_2 = \|\mathbf{y}\|^2$ for any $\mathbf{y} \in \mathbb{R}^d$, and $(ii)$ holds due to Assumption 5.1. Putting everything together, we know that

$$\left\| \nabla \text{tr}\left( \nabla^2 \log q(t,\mathbf{x}) \right) \right\| \leq \frac{6f(t)^3}{g(t)^6}R^3.$$

This completes the proof of Lemma H.1. $\qquad\square$

The following lemma considers the upper bounds for the time-derivative of our score function. We use similar techniques to as we obtain in the proof of Lemma H.1.

**Lemma H.2.** Suppose Assumption 5.1 holds. Let $X_t$ be the forward process defined in (H.1), and its distribution is denoted by $q(t,\mathbf{x})$. Then we have:

$$\left\| \frac{\partial}{\partial t}\left( \nabla \log q(t,\mathbf{x}) \right) \right\| \leq \frac{2g'(t)}{g(t)^3}\left( \|\mathbf{x}\| + |f(t)|R \right) + \frac{|f'(t)|R}{g(t)^2}$$

$$+ \frac{2g(t)|f'(t)| \cdot \left( \|\mathbf{x}\|R + f(t)R^2 \right)^2 + 2|g'(t)| \cdot \left( \|\mathbf{x}\| + f(t)R \right)^3}{g(t)^5},$$

$$\left| \frac{\partial}{\partial t}\left( \text{tr}\left( \nabla^2 \log q(t,\mathbf{x}) \right) \right) \right| \leq \frac{2d \cdot |g'(t)|}{g(t)^3} + \frac{4f(t) \cdot \left| f'(t)g(t) - 2f(t)g'(t) \right|}{g(t)^5}R^2$$

$$+ 6R^2 \frac{f(t)^2}{g(t)^4}\left( \frac{g(t)|f'(t)| \cdot \left( \|\mathbf{x}\|R + f(t)R^2 \right) + |g'(t)| \cdot \left( \|\mathbf{x}\| + f(t)R \right)^2}{g(t)^3} \right).$$

*Proof of Lemma H.2.* To start with, we first compute the time-derivative of $\phi(\frac{\mathbf{x}-f(t)\mathbf{y}}{g(t)})$. Using the chain rule, we have

$$\frac{\partial}{\partial t}\phi\left( \frac{\mathbf{x}-f(t)\mathbf{y}}{g(t)} \right) = \frac{\partial}{\partial t}\left( \frac{1}{(\sqrt{2\pi})^d}e^{-\frac{\left\|\frac{\mathbf{x}-f(t)\mathbf{y}}{g(t)}\right\|^2}{2}} \right)$$

$$= \frac{1}{(\sqrt{2\pi})^d}e^{-\frac{\left\|\frac{\mathbf{x}-f(t)\mathbf{y}}{g(t)}\right\|^2}{2}}\left( -\frac{1}{2} \right)\frac{\partial}{\partial t}\left\| \frac{\mathbf{x}-f(t)\mathbf{y}}{g(t)} \right\|^2. \tag{H.16}$$

Since we know that:

$$\frac{\partial}{\partial t}\left\| \frac{\mathbf{x}-f(t)\mathbf{y}}{g(t)} \right\|^2 = \frac{\partial}{\partial t}\frac{\|\mathbf{x}\|^2 - 2f(t)\mathbf{x}^\top\mathbf{y} + f(t)^2\|\mathbf{y}\|^2}{g(t)^2}$$

$$= \frac{\left( -2f'(t)\mathbf{x}^\top\mathbf{y} + 2f(t)f'(t)\|\mathbf{y}\|^2 \right)g(t)^2 - 2\|\mathbf{x}-f(t)\mathbf{y}\|^2 g(t)g'(t)}{g(t)^4}$$

$$= -\frac{2}{g(t)^3}\left( g(t)f'(t)\left( \mathbf{x}^\top\mathbf{y} - f(t)\|\mathbf{y}\|^2 \right) + g'(t)\|\mathbf{x}-f(t)\mathbf{y}\|^2 \right). \tag{H.17}$$

Substituting (H.17) into (H.16), and recall our shorthand expression $\phi = \phi(\frac{\mathbf{x}-f(t)\mathbf{y}}{g(t)})$, we know that:

$$\frac{\partial}{\partial t}\phi = \frac{g(t)f'(t)\left( \mathbf{x}^\top\mathbf{y} - f(t)\|\mathbf{y}\|^2 \right) + g'(t)\|\mathbf{x}-f(t)\mathbf{y}\|^2}{g(t)^3}\phi. \tag{H.18}$$

Therefore, assuming $\|\mathbf{y}\| \leq R$, we have

$$\left| \frac{\partial}{\partial t}\phi \right| = \left| \frac{g(t)f'(t)\left( \mathbf{x}^\top\mathbf{y} - f(t)\|\mathbf{y}\|^2 \right) + g'(t)\|\mathbf{x}-f(t)\mathbf{y}\|^2}{g(t)^3} \right|\phi$$

$$\leq \frac{g(t)|f'(t)| \cdot \big(\|\mathbf{x}\|\|\mathbf{y}\| + f(t)\|\mathbf{y}\|^2\big) + |g'(t)| \cdot \|\mathbf{x} - f(t)\mathbf{y}\|^2}{g(t)^3}\phi$$

$$\leq \frac{g(t)|f'(t)| \cdot \big(\|\mathbf{x}\|R + f(t)R^2\big) + |g'(t)| \cdot \big(\|\mathbf{x}\| + f(t)R\big)^2}{g(t)^3}\phi. \tag{H.19}$$

**Time-Derivative of the Score Function.**
By (H.4), we know:

$$\nabla \log q(t, \mathbf{x}) = -\frac{1}{g(t)^2} \frac{\int_{\mathbb{R}^d} p_{\text{data}}(\mathbf{y})\big(\mathbf{x} - f(t)\mathbf{y}\big)\phi\big(\frac{\mathbf{x}-f(t)\mathbf{y}}{g(t)}\big)\mathrm{d}\mathbf{y}}{\int_{\mathbb{R}^d} p_{\text{data}}(\mathbf{y})\phi\big(\frac{\mathbf{x}-f(t)\mathbf{y}}{g(t)}\big)\mathrm{d}\mathbf{y}}.$$

Taking the derivative with respect to $t$, we have:

$$\frac{\partial}{\partial t}\Big(\nabla \log q(t, \mathbf{x})\Big) = \frac{2g'(t)}{g(t)^3} \frac{\int_{\mathbb{R}^d} p_{\text{data}}(\mathbf{y})\big(\mathbf{x} - f(t)\mathbf{y}\big)\phi\mathrm{d}\mathbf{y}}{\int_{\mathbb{R}^d} p_{\text{data}}(\mathbf{y})\phi\mathrm{d}\mathbf{y}}$$

$$- \frac{1}{g(t)^2}\Big(\underbrace{\frac{\partial}{\partial t} \frac{\int_{\mathbb{R}^d} p_{\text{data}}(\mathbf{y})\big(\mathbf{x} - f(t)\mathbf{y}\big)\phi\mathrm{d}\mathbf{y}}{\int_{\mathbb{R}^d} p_{\text{data}}(\mathbf{y})\phi\mathrm{d}\mathbf{y}}}_{L_1}\Big).$$

Using Assumption 5.1, we can easily see that:

$$\left\|\frac{\int_{\mathbb{R}^d} p_{\text{data}}(\mathbf{y})\big(\mathbf{x} - f(t)\mathbf{y}\big)\phi\mathrm{d}\mathbf{y}}{\int_{\mathbb{R}^d} p_{\text{data}}(\mathbf{y})\phi\mathrm{d}\mathbf{y}}\right\| \leq \|\mathbf{x}\| + |f(t)|R. \tag{H.20}$$

For $L_1$, we have:

$$L_1 = \frac{\int_{\mathbb{R}^d} p_{\text{data}}(\mathbf{y})\Big(\big(\mathbf{x} - f(t)\mathbf{y}\big)\phi_t - f'(t)\mathbf{y}\phi\Big)\mathrm{d}\mathbf{y}}{\big(\int_{\mathbb{R}^d} p_{\text{data}}(\mathbf{y})\phi\mathrm{d}\mathbf{y}\big)} - \frac{\int_{\mathbb{R}^d} p_{\text{data}}(\mathbf{y})\big(\mathbf{x} - f(t)\mathbf{y}\big)\phi\mathrm{d}\mathbf{y} \int_{\mathbb{R}^d} p_{\text{data}}(\mathbf{y})\phi_t\mathrm{d}\mathbf{y}}{\big(\int_{\mathbb{R}^d} p_{\text{data}}(\mathbf{y})\phi\mathrm{d}\mathbf{y}\big)^2}.$$

From (H.19) and Assumption 5.1, we know that:

$$\|L_1\| \leq \frac{g(t)|f'(t)| \cdot \big(\|\mathbf{x}\|R + f(t)R^2\big) + |g'(t)| \cdot \big(\|\mathbf{x}\| + f(t)R\big)^2}{g(t)^3}2\Big(\|\mathbf{x}\| + |f(t)|R\Big) + |f'(t)|R. \tag{H.21}$$

Combining (H.20) and (H.21), we have:

$$\left\|\frac{\partial}{\partial t}\Big(\nabla \log q(t, \mathbf{x})\Big)\right\| = \frac{2g'(t)}{g(t)^3}\Big(\|\mathbf{x}\| + |f(t)|R\Big)$$

$$+ \frac{1}{g(t)^2}\Big(\frac{g(t)|f'(t)| \cdot \big(\|\mathbf{x}\|R + f(t)R^2\big) + |g'(t)| \cdot \big(\|\mathbf{x}\| + f(t)R\big)^2}{g(t)^3}2\big(\|\mathbf{x}\| + |f(t)|R\big) + |f'(t)|R\Big)$$

$$= \frac{2g'(t)}{g(t)^3}\Big(\|\mathbf{x}\| + |f(t)|R\Big) + \frac{2g(t)|f'(t)| \cdot \big(\|\mathbf{x}\|R + f(t)R^2\big)^2 + 2|g'(t)| \cdot \big(\|\mathbf{x}\| + f(t)R\big)^3}{g(t)^5}$$

$$+ \frac{|f'(t)|R}{g(t)^2}.$$

**Time-Derivative of the Divergence of the Score Function.**
In this section, we consider the $t$-derivative of the divergence of the score function, i.e.,

$$\frac{\partial}{\partial t}\Big(\nabla \cdot [\nabla \log q(t, \mathbf{x})]\Big) = \frac{\partial}{\partial t}\Big(\text{tr}\big(\nabla^2 \log q(t, \mathbf{x})\big)\Big).$$

To start with, using (H.9), we have:

$$\text{tr}\big(\nabla^2 \log q(t, \mathbf{x})\big) = -\frac{d}{g(t)^2} + \frac{f(t)^2}{g(t)^4}\left(\frac{\int_{\mathbb{R}^d} p_{\text{data}}(\mathbf{y})\|\mathbf{y}\|^2\phi\mathrm{d}\mathbf{y}}{\int_{\mathbb{R}^d} p_{\text{data}}(\mathbf{y})\phi\mathrm{d}\mathbf{y}} - \frac{\|\int_{\mathbb{R}^d} p_{\text{data}}(\mathbf{y})\mathbf{y}\phi\mathrm{d}\mathbf{y}\|^2}{\big(\int_{\mathbb{R}^d} p_{\text{data}}(\mathbf{y})\phi\mathrm{d}\mathbf{y}\big)^2}\right). \tag{H.22}$$

Taking the derivative with respect to $t$, we have:

$$\frac{\partial}{\partial t}\Big(\,\text{tr}\left(\nabla^2 \log q(t,\mathbf{x})\right)\Big) = \frac{2d \cdot g'(t)}{g(t)^3}$$
$$+ \underbrace{\frac{2f(t)f'(t)g(t) - 4f(t)^2 g'(t)}{g(t)^5}\left(\frac{\int_{\mathbb{R}^d} p_{\text{data}}(\mathbf{y})\|\mathbf{y}\|^2 \phi \mathrm{d}\mathbf{y}}{\int_{\mathbb{R}^d} p_{\text{data}}(\mathbf{y})\phi \mathrm{d}\mathbf{y}} - \frac{\|\int_{\mathbb{R}^d} p_{\text{data}}(\mathbf{y})\mathbf{y}\phi \mathrm{d}\mathbf{y}\|^2}{\left(\int_{\mathbb{R}^d} p_{\text{data}}(\mathbf{y})\phi \mathrm{d}\mathbf{y}\right)^2}\right)}_{K_1}$$
$$+ \frac{f(t)^2}{g(t)^4}\Big(\underbrace{\frac{\partial}{\partial t}\frac{\int_{\mathbb{R}^d} p_{\text{data}}(\mathbf{y})\|\mathbf{y}\|^2 \phi \mathrm{d}\mathbf{y}}{\int_{\mathbb{R}^d} p_{\text{data}}(\mathbf{y})\phi \mathrm{d}\mathbf{y}}}_{K_2} - \underbrace{\frac{\partial}{\partial t}\frac{\|\int_{\mathbb{R}^d} p_{\text{data}}(\mathbf{y})\mathbf{y}\phi \mathrm{d}\mathbf{y}\|^2}{\left(\int_{\mathbb{R}^d} p_{\text{data}}(\mathbf{y})\phi \mathrm{d}\mathbf{y}\right)^2}}_{K_3}\Big). \tag{H.23}$$

For $K_1$, using Assumption 5.1, we have

$$|K_1| \leq \left|\frac{2f(t)f'(t)g(t) - 4f(t)^2 g'(t)}{g(t)^5}(R^2 + R^2)\right| = \frac{4f(t)\cdot\left|f'(t)g(t) - 2f(t)g'(t)\right|}{g(t)^5}R^2. \tag{H.24}$$

For $K_2$, Using the definition of $K_2$ in (H.23), we have

$$|K_2| = \left|\frac{\partial}{\partial t}\frac{\int_{\mathbb{R}^d} p_{\text{data}}(\mathbf{y})\|\mathbf{y}\|^2 \phi \mathrm{d}\mathbf{y}}{\int_{\mathbb{R}^d} p_{\text{data}}(\mathbf{y})\phi \mathrm{d}\mathbf{y}}\right|$$
$$= \left|\frac{\frac{\partial}{\partial t}\int_{\mathbb{R}^d} p_{\text{data}}(\mathbf{y})\|\mathbf{y}\|^2 \phi \mathrm{d}\mathbf{y}}{\int_{\mathbb{R}^d} p_{\text{data}}(\mathbf{y})\phi \mathrm{d}\mathbf{y}} - \frac{\int_{\mathbb{R}^d} p_{\text{data}}(\mathbf{y})\|\mathbf{y}\|^2 \phi \mathrm{d}\mathbf{y}\cdot\left(\frac{\partial}{\partial t}\int_{\mathbb{R}^d} p_{\text{data}}(\mathbf{y})\phi \mathrm{d}\mathbf{y}\right)}{\left(\int_{\mathbb{R}^d} p_{\text{data}}(\mathbf{y})\phi \mathrm{d}\mathbf{y}\right)^2}\right|$$
$$\leq \left|\frac{\int_{\mathbb{R}^d} p_{\text{data}}(\mathbf{y})\|\mathbf{y}\|^2 \frac{\partial}{\partial t}\phi \mathrm{d}\mathbf{y}}{\int_{\mathbb{R}^d} p_{\text{data}}(\mathbf{y})\phi \mathrm{d}\mathbf{y}}\right| + \left|\frac{\int_{\mathbb{R}^d} p_{\text{data}}(\mathbf{y})\|\mathbf{y}\|^2 \phi \mathrm{d}\mathbf{y}\cdot\left(\int_{\mathbb{R}^d} p_{\text{data}}(\mathbf{y})\frac{\partial}{\partial t}\phi \mathrm{d}\mathbf{y}\right)}{\left(\int_{\mathbb{R}^d} p_{\text{data}}(\mathbf{y})\phi \mathrm{d}\mathbf{y}\right)^2}\right|$$
$$\leq 2R^2 \frac{g(t)|f'(t)|\cdot\left(\|\mathbf{x}\|R + f(t)R^2\right) + |g'(t)|\cdot\left(\|\mathbf{x}\| + f(t)R\right)^2}{g(t)^3}, \tag{H.25}$$

where the last inequality holds due to (H.19) and Assumption 5.1.
For $K_3$, we have

$$|K_3| = \left|\frac{\partial}{\partial t}\frac{\|\int_{\mathbb{R}^d} p_{\text{data}}(\mathbf{y})\mathbf{y}\phi \mathrm{d}\mathbf{y}\|^2}{(\int_{\mathbb{R}^d} p_{\text{data}}(\mathbf{y})\phi \mathrm{d}\mathbf{y})^2}\right|$$
$$\leq \left|\frac{\frac{\partial}{\partial t}\|\int_{\mathbb{R}^d} p_{\text{data}}(\mathbf{y})\mathbf{y}\phi \mathrm{d}\mathbf{y}\|^2}{(\int_{\mathbb{R}^d} p_{\text{data}}(\mathbf{y})\phi \mathrm{d}\mathbf{y})^2} - \frac{\left\|\int_{\mathbb{R}^d} p_{\text{data}}(\mathbf{y})\mathbf{y}\phi \mathrm{d}\mathbf{y}\right\|^2 \cdot\frac{\partial}{\partial t}\left(\int_{\mathbb{R}^d} p_{\text{data}}(\mathbf{y})\phi \mathrm{d}\mathbf{y}\right)^2}{(\int_{\mathbb{R}^d} p_{\text{data}}(\mathbf{y})\phi \mathrm{d}\mathbf{y})^4}\right|$$
$$\leq \left|\frac{\frac{\partial}{\partial t}\|\int_{\mathbb{R}^d} p_{\text{data}}(\mathbf{y})\mathbf{y}\phi \mathrm{d}\mathbf{y}\|^2}{(\int_{\mathbb{R}^d} p_{\text{data}}(\mathbf{y})\phi \mathrm{d}\mathbf{y})^2}\right| + \left|\frac{\left\|\int_{\mathbb{R}^d} p_{\text{data}}(\mathbf{y})\mathbf{y}\phi \mathrm{d}\mathbf{y}\right\|^2 \cdot\frac{\partial}{\partial t}\left(\int_{\mathbb{R}^d} p_{\text{data}}(\mathbf{y})\phi \mathrm{d}\mathbf{y}\right)^2}{(\int_{\mathbb{R}^d} p_{\text{data}}(\mathbf{y})\phi \mathrm{d}\mathbf{y})^4}\right|. \tag{H.26}$$

Moreover, we have:

$$\left|\frac{\partial}{\partial t}\Big\|\int_{\mathbb{R}^d} p_{\text{data}}(\mathbf{y})\mathbf{y}\phi \mathrm{d}\mathbf{y}\Big\|^2\right| = \left|\frac{\partial}{\partial t}\sum_{i=1}^d \Big(\int_{\mathbb{R}^d} p_{\text{data}}(\mathbf{y})y_i \phi \mathrm{d}\mathbf{y}\Big)^2\right|$$
$$= \left|\sum_{i=1}^d 2\int_{\mathbb{R}^d} p_{\text{data}}(\mathbf{y})y_i \phi \mathrm{d}\mathbf{y}\int_{\mathbb{R}^d} p_{\text{data}}(\mathbf{y})y_i \frac{\partial}{\partial t}\phi \mathrm{d}\mathbf{y}\right|$$
$$\leq \sum_{i=1}^d 2\int_{\mathbb{R}^d} p_{\text{data}}(\mathbf{y})|y_i|\phi \mathrm{d}\mathbf{y}\cdot\int_{\mathbb{R}^d} p_{\text{data}}(\mathbf{y})|y_i|\Big|\frac{\partial}{\partial t}\phi\Big|\mathrm{d}\mathbf{y}$$
$$\leq 2\left(\frac{g(t)|f'(t)|\cdot\left(\|\mathbf{x}\|R + f(t)R^2\right) + |g'(t)|\cdot\left(\|\mathbf{x}\| + f(t)R\right)^2}{g(t)^3}\right)\sum_{i=1}^d\left(\int_{\mathbb{R}^d} p_{\text{data}}(\mathbf{y})|y_i|\phi \mathrm{d}\mathbf{y}\right)^2.$$

Using the Cauchy-Schwarz inequality, we have

$$\sum_{i=1}^d\left(\int_{\mathbb{R}^d} p_{\text{data}}(\mathbf{y})|y_i|\phi \mathrm{d}\mathbf{y}\right)^2 \leq \sum_{i=1}^d\left(\int_{\mathbb{R}^d} p_{\text{data}}(\mathbf{y})\phi \mathrm{d}\mathbf{y}\right)\left(\int_{\mathbb{R}^d} p_{\text{data}}(\mathbf{y})|y_i|^2 \phi \mathrm{d}\mathbf{y}\right)$$

$$= \left( \int_{\mathbb{R}^d} p_{\text{data}}(\mathbf{y})\phi\mathrm{d}\mathbf{y} \right)\left( \int_{\mathbb{R}^d} p_{\text{data}}(\mathbf{y})\|\mathbf{y}\|^2\phi\mathrm{d}\mathbf{y} \right)$$

$$\leq R^2 \left( \int_{\mathbb{R}^d} p_{\text{data}}(\mathbf{y})\phi\mathrm{d}\mathbf{y} \right)^2,$$

where the last inequality holds due to Assumption 5.1. Therefore, we have

$$\left| \frac{\partial}{\partial t}\Big\| \int_{\mathbb{R}^d} p_{\text{data}}(\mathbf{y})\mathbf{y}\phi\mathrm{d}\mathbf{y} \Big\|^2 \right|$$

$$\leq 2R^2 \left( \frac{g(t)|f'(t)| \cdot \left( \|\mathbf{x}\|R + f(t)R^2 \right) + |g'(t)| \cdot \left( \|\mathbf{x}\| + f(t)R \right)^2}{g(t)^3} \right)\left( \int_{\mathbb{R}^d} p_{\text{data}}(\mathbf{y})\phi\mathrm{d}\mathbf{y} \right)^2. \tag{H.27}$$

Moreover, we have:

$$\left| \frac{\partial}{\partial t}\Big( \int_{\mathbb{R}^d} p_{\text{data}}(\mathbf{y})\phi\mathrm{d}\mathbf{y} \Big)^2 \right| = 2\left| \int_{\mathbb{R}^d} p_{\text{data}}(\mathbf{y})\phi\mathrm{d}\mathbf{y} \cdot \frac{\partial}{\partial t}\int_{\mathbb{R}^d} p_{\text{data}}(\mathbf{y})\phi\mathrm{d}\mathbf{y} \right|$$

$$\leq 2\left| \int_{\mathbb{R}^d} p_{\text{data}}(\mathbf{y})\phi\mathrm{d}\mathbf{y} \right| \cdot \int_{\mathbb{R}^d} p_{\text{data}}(\mathbf{y})\Big| \frac{\partial}{\partial t}\phi \Big|\mathrm{d}\mathbf{y}$$

$$\leq 2\frac{g(t)|f'(t)| \cdot \left( \|\mathbf{x}\|R + f(t)R^2 \right) + |g'(t)| \cdot \left( \|\mathbf{x}\| + f(t)R \right)^2}{g(t)^3}\left( \int_{\mathbb{R}^d} p_{\text{data}}(\mathbf{y})\phi\mathrm{d}\mathbf{y} \right)^2, \tag{H.28}$$

where the last inequality holds due to (H.18). Substituting (H.27) and (H.28) into (H.26), and using Assumption 5.1, we have:

$$|K_3| \leq 4R^2 \left( \frac{g(t)|f'(t)| \cdot \left( \|\mathbf{x}\|R + f(t)R^2 \right) + |g'(t)| \cdot \left( \|\mathbf{x}\| + f(t)R \right)^2}{g(t)^3} \right). \tag{H.29}$$

Combining (H.23), (H.24), (H.25) and (H.29), we have the following inequality:

$$\left| \frac{\partial}{\partial t}\Big( \text{tr}\left( \nabla^2 \log q(t, \mathbf{x}) \right) \Big) \right| \leq \frac{2d \cdot |g'(t)|}{g(t)^3} + \frac{4f(t) \cdot \left| f'(t)g(t) - 2f(t)g'(t) \right|}{g(t)^5}R^2$$

$$+ 6R^2 \frac{f(t)^2}{g(t)^4}\left( \frac{g(t)|f'(t)| \cdot \left( \|\mathbf{x}\|R + f(t)R^2 \right) + |g'(t)| \cdot \left( \|\mathbf{x}\| + f(t)R \right)^2}{g(t)^3} \right). \tag{H.30}$$

This completes the proof of Lemma H.2. $\qquad\square$

## H.2 PROOF OF LEMMAS F.3 AND F.4

*Proof of Lemma F.3.* Since $\boldsymbol{X}_t = \mathrm{e}^{-t}\boldsymbol{X}_0 + N\big(0, (1 - \mathrm{e}^{-2t})\mathbf{I}_d\big)$, we have $f(t) = \mathrm{e}^{-t}$ and $g(t) = \sqrt{1 - \mathrm{e}^{-2t}}$ in this case. When $t < 1$, we know that $f(t) = \Theta(1)$, $f'(t) = \Theta(1)$, $g(t) = \Theta(\sqrt{t})$, $g'(t) = \Theta(\frac{1}{\sqrt{t}})$. Hence by Lemma H.2, we have:

$$\left| \frac{\partial}{\partial t}\Big( \text{tr}\left( \nabla^2 \log q(t, \mathbf{x}) \right) \Big) \right| \lesssim \frac{d}{t^2} + \frac{R^2}{t^3} + R^2 \frac{1}{t^2}\frac{\sqrt{t}(\|\mathbf{x}\| + R)R + \frac{1}{\sqrt{t}}(\|\mathbf{x}\| + R)^2}{\sqrt{t}^3}$$

$$= \frac{d}{t^2} + \frac{R^2}{t^3} + \frac{R^3(\|\mathbf{x}\| + R)}{t^3} + \frac{R^2(\|\mathbf{x}\| + R)^2}{t^4}$$

$$\overset{(i)}{\lesssim} \frac{d}{t^2} + \frac{R^2(\|\mathbf{x}\| + R)^2}{t^4}, \quad t < 1.$$

Here in $(i)$ we use $2\frac{R^2}{t^3} \leq \frac{1}{t^2} + \frac{R^4}{t^4} \leq \frac{d}{t^2} + \frac{R^2(\|\mathbf{x}\|+R)^2}{t^4}$. We next consider the case $t \geq 1$:
Denote $\mathrm{e}^{-t}$ by $a$, then we know that $f(t) = \Theta(a)$, $f'(t) = \Theta(a)$, $g(t) = \Theta(1)$, $g'(t) = \Theta(a^2)$. By, Lemma H.2, we have:

$$\left| \frac{\partial}{\partial t}\Big( \text{tr}\left( \nabla^2 \log q(t, \mathbf{x}) \right) \Big) \right| \lesssim a^2d + R^2a^2 + R^2a^2\Big( a(\|\mathbf{x}\| + aR)R + a^2(\|\mathbf{x}\| + aR)^2 \Big)$$

$$\overset{(ii)}{\leq} a^2 d + R^2 a^2 + R^2 a^2 \left( (\|\mathbf{x}\| + R)R + (\|\mathbf{x}\| + R)^2 \right)$$

$$\lesssim a^2 d + a^2 R^2 + a^2 R^2 (\|\mathbf{x}\| + R)^2$$

$$\overset{(iii)}{\lesssim} a^2 d + a^2 R^2 (\|\mathbf{x}\| + R)^2$$

$$\leq d + R^2 (\|\mathbf{x}\| + R)^2, \quad t \geq 1.$$

Here $(ii)$ is due to $a < 1$ and $(iii)$ is because $2R^2 \leq 1 + R^4 \leq d + R^2 (\|\mathbf{x}\| + R)^2$. This proves the first inequality of Lemma F.3.

Same as above, When $t < 1$, we know that $f(t) = \Theta(1)$, $f'(t) = \Theta(1)$, $g(t) = \Theta(\sqrt{t})$, $g'(t) = \Theta(\frac{1}{\sqrt{t}})$. Hence by Lemma H.2, we have:

$$\left\| \frac{\partial}{\partial t} \left( \nabla \log q(t, \mathbf{x}) \right) \right\| \lesssim \frac{\|\mathbf{x}\| + R}{t^2} + \frac{(\|\mathbf{x}\| + R)^2 R^2}{t^2} + \frac{(\|\mathbf{x}\| + R)^3}{t^3} + \frac{R}{t}$$

$$\lesssim \frac{\|\mathbf{x}\| + R}{t^2} + \frac{(\|\mathbf{x}\| + R)^2 R^2}{t^2} + \frac{(\|\mathbf{x}\| + R)^3}{t^3}, \quad t < 1.$$

We next consider the case $t \geq 1$. Denote $\mathrm{e}^{-t}$ by $a$, then we know that $f(t) = \Theta(a)$, $f'(t) = \Theta(a)$, $g(t) = \Theta(1)$, $g'(t) = \Theta(a^2)$. By, Lemma H.2, we have:

$$\left\| \frac{\partial}{\partial t} \left( \nabla \log q(t, \mathbf{x}) \right) \right\| \lesssim a^2 (\|\mathbf{x}\| + aR) + a(\|\mathbf{x}\| + aR)^2 R^2 + a^2 (\|\mathbf{x}\| + aR)^3 + aR$$

$$\overset{(i)}{\lesssim} a(\|\mathbf{x}\| + R) + a(\|\mathbf{x}\| + R)^2 R^2 + a(\|\mathbf{x}\| + R)^3$$

$$\leq (\|\mathbf{x}\| + R) + (\|\mathbf{x}\| + R)^2 R^2 + (\|\mathbf{x}\| + R)^3, \quad t \geq 1.$$

Here $(i)$ is due to $a = \mathrm{e}^{-t} < 1$ given $t \geq 1$. This completes the proof of Lemma F.3. $\qquad\square$

*Proof of Lemma F.4.* Because $\boldsymbol{X}_t = \mathrm{e}^{-t} \boldsymbol{X}_0 + N\left(0, (1 - \mathrm{e}^{-2t})\mathbf{I}_d\right)$, we have $f(t) = \mathrm{e}^{-t}$ and $g(t) = \sqrt{1 - \mathrm{e}^{-2t}}$ in this case. When $t < 1$, we know that

$$f(t) = \Theta(1), f'(t) = \Theta(1), g(t) = \Theta(\sqrt{t}), g'(t) = \Theta(\frac{1}{\sqrt{t}}).$$

Then by Lemma H.1, we can easily show the three inequalities are valid when $t < 1$.
Moreover, when $t \geq 1$, we have

$$f(t) = \Theta(\mathrm{e}^{-t}), f'(t) = \Theta(\mathrm{e}^{-t}), g(t) = \Theta(1), g'(t) = \Theta(\mathrm{e}^{-2t}).$$

Then by Lemma H.1, we can easily show the three inequalities are valid when $t \geq 1$. This completes the proof of Lemma F.4. $\qquad\square$

## H.3 Proof of Lemmas G.3 and G.4

*Proof of Lemma G.3.* Since $\boldsymbol{X}_t = \boldsymbol{X}_0 + N\left(0, \sqrt{t}\mathbf{I}_d\right)$, we have $f(t) = 1$ and $g(t) = \sqrt{t}$ in this case. Hence $f'(t) = 0$ and $g'(t) = \Theta(\frac{1}{\sqrt{t}})$. By Lemma H.2, we have:

$$\left\| \frac{\partial}{\partial t} \left( \nabla \log q(t, \mathbf{x}) \right) \right\| \lesssim \frac{\|\mathbf{x}\| + R}{t^2} + \frac{(\|\mathbf{x}\| + R)^3}{t^3}.$$

$$\left| \frac{\partial}{\partial t} \left( \mathrm{tr} \left( \nabla^2 \log q(t, \mathbf{x}) \right) \right) \right| \lesssim \frac{d}{t^2} + \frac{R^2}{t^3} + \frac{R^2 (\|\mathbf{x}\| + R)^2}{t^4}$$

$$\overset{(i)}{\lesssim} \frac{d}{t^2} + \frac{R^2 (\|\mathbf{x}\| + R)^2}{t^4},$$

here $(i)$ is due to $2\frac{R^2}{t^3} \leq \frac{1}{t^2} + \frac{R^4}{t^4}$. This completes our proof. $\qquad\square$

*Proof of Lemma G.4.* Since $\boldsymbol{X}_t = \boldsymbol{X}_0 + N\left(0, \sqrt{t}\mathbf{I}_d\right)$, we have $f(t) = 1$ and $g(t) = \sqrt{t}$ in this case. Hence $f'(t) = 0$ and $g'(t) = \Theta(\frac{1}{\sqrt{t}})$. Substituting into Lemma H.1, we completes our proof. $\qquad\square$

# I    PROOF OF REMAINING LEMMAS IN SECTIONS F AND G

## I.1    PROOF OF LEMMAS F.1 AND F.2

We first present the following two technical lemmas.

**Lemma I.1.** Let $Y_t$ and $F_t(\mathbf{z})$ be defined in (F.2) and (F.3). Then for any $0 \leq k \leq N - 1$, $t \in [t_k, t_{k+1}]$ and $\mathbf{x} \in \mathbb{R}^d$, we have:

$$\frac{p_{\mathbf{Y}_t}(\mathbf{x})}{p_{F_t(\mathbf{Y}_{t_k})}(\mathbf{x})} \leq \frac{\mathrm{e} \left| \nabla F_t\big(F_t^{-1}(\mathbf{x})\big) \right|}{a} \cdot \Big( \frac{1 - \mathrm{e}^{-2(T-t)} + \frac{a^2 - 1}{1 + 2/d}}{1 - \mathrm{e}^{-2(T-t)}} \Big)^{d/2} \cdot \mathrm{e}^{\frac{(1+d/2)\|\mathbf{g}(\mathbf{x})\|_2^2}{2(a^2 - 1)}},$$

where $a = \mathrm{e}^{t - t_k}$ and $\mathbf{g}(\mathbf{x}) = a F_t^{-1}(\mathbf{x}) - \mathbf{x}$.

**Lemma I.2.** Suppose $\mathbf{Y}_t = \mathrm{e}^{-(T-t)} \mathbf{X}_{\mathrm{data}} + N\big(0, (1 - \mathrm{e}^{-2(T-t)})I_d\big)$ and $p(\|\mathbf{X}_{\mathrm{data}}\|_2 < R) = 1$. Then for $\lambda < \min\{\frac{1}{d}, \frac{1}{2}\}$, we have:

$$\mathbb{E}\mathrm{e}^{\lambda \|\mathbf{Y}_t\|_2^2} \leq \mathrm{e}^{\lambda R^2 - 1}.$$

Using these lemmas, we can start the proof of Lemma F.1:

*Proof of Lemma F.1.* To start with, we have the following lemma about the ratio: Firstly, using the expression of the interpolation operator (F.3), we have

$$\nabla F_t(\mathbf{x}) = \mathrm{e}^{t - t_k} \mathbf{I}_d + \big(\mathrm{e}^{t - t_k} - 1\big) \nabla \mathbf{s}_\theta(T - t_k, \mathbf{x}).$$

The operator norm of $\nabla F_t(\mathbf{x})$ can be bounded by

$$\begin{aligned} \|\nabla F_t(\mathbf{x})\|_2 &\leq \mathrm{e}^{t - t_k} + \big(\mathrm{e}^{t - t_k} - 1\big)L \\ &= a + (a - 1)L \end{aligned}$$

Using the fact that $|\mathbf{A}| \leq \|\mathbf{A}\|^d$, we know that:

$$\begin{aligned} \left| \nabla F_t\big(F_t^{-1}(\mathbf{x})\big) \right| &\leq \big(a + (a - 1)L\big)^d \\ &\leq \big(1 + 2\eta + 2\eta L\big)^d \\ &\leq \mathrm{e} \end{aligned}$$

Here the last inequality is because $\eta \leq \frac{1}{2(L+1)d}$. With our time schedule, by (2.6), we have

$$t - t_k \leq t_{k+1} - t_k \leq \eta \min\{1, T - t\}.$$

We consider two cases when $T - t \geq 1$ and $T - t < 1$.
First, when $T - t \geq 1$, we have $t - t_k \leq \eta$. Therefore, we have

$$\begin{aligned} \Big( \frac{1 - \mathrm{e}^{-2(T-t)} + \frac{a^2 - 1}{1 + 2/d}}{1 - \mathrm{e}^{-2(T-t)}} \Big)^{d/2} &\leq \Big( 1 + \frac{\mathrm{e}^{2(t - t_k)} - 1}{1 - \mathrm{e}^{-2(T-t)}} \Big)^{d/2} \\ &\leq \Big( 1 + \frac{\mathrm{e}^{2\eta} - 1}{1 - \mathrm{e}^{-2}} \Big)^{d/2}. \end{aligned}$$

Since $\mathrm{e}^x - 1 \leq 2x$ when $x \leq 1$ and $1 - \mathrm{e}^{-2} > \frac{1}{2}$, when $\eta \leq 1/(8d)$, we have:

$$\Big( 1 + \frac{\mathrm{e}^{2\eta} - 1}{1 - \mathrm{e}^{-2}} \Big)^{d/2} \leq \Big( 1 + \frac{1}{d} \Big)^{d/2} \leq \sqrt{\mathrm{e}}.$$

Secondly, when $T - t < 1$, since $1 - \mathrm{e}^{-2x} > x/4$ and $\mathrm{e}^x - 1 \leq 2x$ when $0 < x < 1$, when $\eta \leq 1/16d$, we know that:

$$\begin{aligned} \Big( \frac{1 - \mathrm{e}^{-2(T-t)} + \frac{a^2 - 1}{1 + 2/d}}{1 - \mathrm{e}^{-2(T-t)}} \Big)^{d/2} &\leq \Big( 1 + 4\frac{\mathrm{e}^{2(t - t_k)} - 1}{T - t} \Big)^{d/2} \\ &\leq \Big( 1 + 4\frac{4(t - t_k)}{T - t} \Big)^{d/2} \end{aligned}$$

$$\leq (1+16\eta)^{d/2} \leq \sqrt{e}.$$

Combining the two cases, we can summarize that when $\eta \leq 1/16d$, we have

$$\left(\frac{1 - e^{-2(T-t)} + \frac{a^2-1}{1+2/d}}{1 - e^{-2(T-t)}}\right)^{d/2} \leq \sqrt{e}.$$

Since $g(\mathbf{x}) = aF_t^{-1}(\mathbf{x}) - \mathbf{x}$, we have:

$$\begin{aligned}
g(\mathbf{x}) &= aF_t^{-1}(\mathbf{x}) - \mathbf{x} \\
&= aF_t^{-1}(\mathbf{x}) - aF_t^{-1}(\mathbf{x}) - (a-1)\mathbf{s}_\theta(T - t_k, F_t^{-1}(\mathbf{x})) \\
&= -(a-1)\mathbf{s}_\theta(T - t_k, F_t^{-1}(\mathbf{x})).
\end{aligned}$$

Hence,

$$\frac{(1+d/2)\|\mathbf{g}(\mathbf{x})\|_2^2}{2(a^2-1)} = \frac{(1+d/2)(a-1)\|\mathbf{s}_\theta(T - t_k, F_t^{-1}(\mathbf{x}))\|^2}{2(a+1)}.$$

Using the assumption on $\mathbf{s}_\theta$ and inequality (F.4), we have that:

$$\begin{aligned}
\frac{(1+d/2)(a-1)\|\mathbf{s}_\theta(T - t_k, F_t^{-1}(\mathbf{x}))\|^2}{2(a+1)} &\leq \frac{(1+d/2)(a-1)}{2(a+1)}(L\|F_t^{-1}(\mathbf{x})\| + c)^2 \\
&\leq \frac{(1+d/2)(a-1)}{2(a+1)}(2L\|\mathbf{x}\| + L + c)^2 \\
&\leq \frac{(1+d/2)(a-1)}{2(a+1)} \\
&\leq \frac{(1+d/2)(a-1)}{a+1}4L^2\|\mathbf{x}\|^2 + \frac{(1+d/2)(a-1)}{a+1}(L+c)^2 \\
&\leq (t - t_k)\underbrace{(1+d/2)4L^2}_{c_1}\|\mathbf{x}\|^2 + (t - t_k)\underbrace{(1+d/2)(L+c)^2}_{c_2},
\end{aligned}$$

where the 1st inequality holds due to $a \geq 1$ and $a - 1 \leq 2(t - t_k)$. Thus, we know that:

$$\frac{p_{\mathbf{Y}_t}(\mathbf{x})}{p_{F_t(\mathbf{Y}_{t_k})}(\mathbf{x})} \lesssim e^{(t-t_k)c_1\|\mathbf{x}\|^2 + (t-t_k)c_2}.$$

This proves the first part of Lemma F.1.

Next we prove the second part, we know that:

$$\begin{aligned}
\int_{\Omega_t} \left(\frac{p_{\mathbf{Y}_t}(\mathbf{x})}{p_{F_t(\mathbf{Y}_{t_k})}(\mathbf{x})}\right)^2 p_{F_t(\mathbf{Y}_{t_k})}(\mathbf{x})d\mathbf{x} &\lesssim \int_{\Omega_t} e^{(t-t_k)c_1\|\mathbf{x}\|^2 + (t-t_k)c_2} p_{\mathbf{Y}_t}(\mathbf{x})d\mathbf{x} \\
&\leq e^{(t-t_k)c_2}\mathbb{E}e^{(t-t_k)c_1\|\mathbf{Y}_t\|^2} \\
&\overset{(i)}{\leq} e^{(t-t_k)c_2}e^{(t-t_k)c_1R^2-1} \\
&\overset{(ii)}{\lesssim} 1,
\end{aligned}$$

where the $(i)$ holds due to Lemma I.2 since our $t - t_k \leq \eta \leq \min\{\frac{1}{c_1d}, \frac{1}{2c_1}\}$. And $(ii)$ holds due to $t - t_k \leq \eta \leq \min\{\frac{1}{c_2}, \frac{1}{R^2c_1}\}$

By Lemma I.1, we know that:

$$\begin{aligned}
\frac{p_{\mathbf{Y}_t}(\mathbf{x})}{p_{F_t(\mathbf{Y}_{t_k})}(\mathbf{x})} &\leq \frac{e\left|\nabla F_t\left(F_t^{-1}(\mathbf{x})\right)\right|}{a}\left(\frac{1 - e^{-2(T-t)} + \frac{a^2-1}{1+2/d}}{1 - e^{-2(T-t)}}\right)^{d/2} \cdot e^{\frac{(1+d/2)\|\mathbf{g}(\mathbf{x})\|_2^2}{2(a^2-1)}} \\
&\lesssim 1.
\end{aligned}$$

here $a = e^{t-t_k}$ and $\mathbf{g}(\mathbf{x}) = aF_t^{-1}(\mathbf{x}) - \mathbf{x}$. This completes the proof of Lemma F.1. $\qquad\square$

Next, we begin our proof of Lemma F.2.

*Proof of Lemma F.2.* To start with, using Lemma F.1, we know that:

$$\frac{p_{\boldsymbol{Y}_t}(\mathbf{x})}{p_{F_t(\boldsymbol{Y}_{t_k})}(\mathbf{x})} \lesssim \mathrm{e}^{(t-t_k)c_1\|\mathbf{x}\|^2+(t-t_k)c_2},$$

where $c_1 = (1+d/2)4L^2$ and $c_2 = (1+d/2)(L+c)^2$.
Thus, we have:

$$\begin{aligned}
\int_{\Omega_t} \left\|\frac{\nabla q(T-t,\mathbf{x})}{p_{F_t(\boldsymbol{Y}_{t_k})}(\mathbf{x})}\right\|^2 p_{F_t(\boldsymbol{Y}_{t_k})}(\mathbf{x})\mathrm{d}\mathbf{x} &= \int_{\Omega_t}\left\|\nabla\log q(T-t,\mathbf{x})\right\|^2 \frac{p_{\boldsymbol{Y}_t}(\mathbf{x})}{p_{F_t(\boldsymbol{Y}_{t_k})}(\mathbf{x})}p_{\boldsymbol{Y}_t}(\mathbf{x})\mathrm{d}\mathbf{x}\\
&\lesssim \int_{\Omega_t}\left\|\nabla\log q(T-t,\mathbf{x})\right\|^2 \mathrm{e}^{\eta(c_1\|\mathbf{x}\|^2+c_2)}p_{\boldsymbol{Y}_t}(\mathbf{x})\mathrm{d}\mathbf{x}\\
&\lesssim \mathbb{E}_Q\Big[\left\|\nabla\log q(T-t,\boldsymbol{Y}_t)\right\|^2\mathrm{e}^{\eta c_1\|\boldsymbol{Y}_t\|^2}\Big],
\end{aligned}$$

where the last inequality holds due to $\eta \leq \frac{1}{c_2}$. Next, we select a constant $M = 2R$. We have the following inequality:

$$\begin{aligned}
&\mathbb{E}_Q\Big[\left\|\nabla\log q(T-t,\boldsymbol{Y}_t)\right\|^2\mathrm{e}^{\eta c_1\|\boldsymbol{Y}_t\|^2}\Big]\\
&= \mathbb{E}_Q\Big[\left\|\nabla\log q(T-t,\boldsymbol{Y}_t)\right\|^2\mathrm{e}^{\eta c_1\|\boldsymbol{Y}_t\|^2}1_{\|\boldsymbol{Y}_t\|<M}\Big] + \mathbb{E}_Q\Big[\left\|\nabla\log q(T-t,\boldsymbol{Y}_t)\right\|^2\mathrm{e}^{\eta c_1\|\boldsymbol{Y}_t\|^2}1_{\|\boldsymbol{Y}_t\|\geq M}\Big]\\
&\lesssim \underbrace{\mathbb{E}_Q\left\|\nabla\log q(T-t,\boldsymbol{Y}_t)\right\|^2}_{I_1} + \underbrace{\mathbb{E}_Q\Big[\left\|\nabla\log q(T-t,\boldsymbol{Y}_t)\right\|^2\mathrm{e}^{\eta c_1\|\boldsymbol{Y}_t\|^2}1_{\|\boldsymbol{Y}_t\|\geq M}\Big]}_{I_2},
\end{aligned}$$

where the last inequality holds due to $\eta \leq \frac{1}{4R^2 c_1} = \frac{1}{c_1 M^2}$. For $I_1$, using Lemma K.2, we have:

$$\begin{aligned}
I_1 &= \mathbb{E}_Q\|\nabla\log q(T-t,\boldsymbol{Y}_t)\|^2\\
&\leq d\frac{1}{1-\mathrm{e}^{-2(T-t)}}\\
&\overset{(i)}{\lesssim} \frac{d}{\min\{T-t,1\}},
\end{aligned} \tag{I.1}$$

where $(i)$ holds because $1-\mathrm{e}^{-2x} > x/2$ when $x < 1$ and $1-\mathrm{e}^{-2x} > 1/2$ when $x \geq 1$.
For $I_2$, using Lemma H.1 with $f(t) = \mathrm{e}^{-(T-t)}$, $g(t) = \sqrt{1-\mathrm{e}^{-2(T-t)}}$, we have:

$$\begin{aligned}
\|\nabla\log q(T-t,\mathbf{x})\| &\leq \frac{\|\mathbf{x}\|+R}{1-\mathrm{e}^{-2(T-t)}}\\
&= \frac{\|\mathbf{x}\|+R}{\sigma_{T-t}},
\end{aligned}$$

here we denote $1-\mathrm{e}^{-2(T-t)}$ by $\sigma_{T-t}$. Therefore, we have:

$$I_2 \lesssim \mathbb{E}_Q\Big[\Big(\frac{\|\boldsymbol{Y}_t\|+R}{\sigma_{T-t}}\Big)^2\mathrm{e}^{\eta c_1\|\boldsymbol{Y}_t\|^2}1_{\|\boldsymbol{Y}_t\|\geq M}\Big]. \tag{I.2}$$

Let $\alpha = \mathrm{e}^{-(T-t)}$, we have $\boldsymbol{Y}_t = \boldsymbol{X}_{T-t} = \alpha\boldsymbol{X}_0 + \sqrt{1-\alpha^2}\boldsymbol{Z}$. Thus, $\|\boldsymbol{Y}_t\| \leq R + \sqrt{1-\alpha^2}\|\boldsymbol{Z}\|$.
Hence, (I.2) becomes

$$\begin{aligned}
I_2 &\leq \frac{1}{(\sigma_{T-t})^2}\mathbb{E}_Q\Big[\big(2R+\sqrt{1-\alpha^2}\|\boldsymbol{Z}\|\big)^2\mathrm{e}^{\eta c_1(R+\sqrt{1-\alpha^2}\|\boldsymbol{Z}\|)^2}1_{\|\boldsymbol{Z}\|\geq\frac{M-R}{\sqrt{1-\alpha^2}}}\Big]\\
&\lesssim \frac{1}{(\sigma_{T-t})^2}\mathbb{E}_Q\Big[\big(R^2+(1-\alpha^2)\|\boldsymbol{Z}\|^2\big)\mathrm{e}^{2\eta c_1(1-\alpha^2)\|\boldsymbol{Z}\|^2}1_{\|\boldsymbol{Z}\|\geq\frac{M-R}{\sqrt{1-\alpha^2}}}\Big],
\end{aligned} \tag{I.3}$$

where the last inequality holds due to $(a+b)^2 \leq 2a^2+2b^2$, and $\eta \leq 1/c_1 R^2$. Let $\lambda = 2\eta c_1(1-\alpha^2)$. Then we have

$$I_2 \lesssim \frac{R^2}{(\sigma_{T-t})^2}\underbrace{\int_{\mathbb{R}^d}\frac{1}{(\sqrt{2\pi})^d}\mathrm{e}^{-\frac{\|\mathbf{z}\|^2}{2}}\mathrm{e}^{\lambda\|\mathbf{z}\|^2}1_{\|\mathbf{z}\|\geq\frac{M-R}{\sqrt{1-\alpha^2}}}\mathrm{d}\mathbf{z}}_{K_1}$$

$$+ \frac{2}{\sigma_{T-t}} \underbrace{\int_{\mathbb{R}^d} \frac{1}{\sqrt{2\pi}^d} e^{-\frac{\|\mathbf{z}\|^2}{2}} \|\mathbf{z}\|^2 e^{\lambda\|\mathbf{z}\|^2} 1_{\|\mathbf{z}\| \geq \frac{M-R}{\sqrt{1-\alpha^2}}} d\mathbf{z}}_{K_2}, \tag{I.4}$$

here we use that $1 - \alpha^2 = 1 - e^{-2(T-t)} = \sigma_{T-t}$. Let $\phi_{\sigma^2}(\mathbf{x}) = \exp(-\|\mathbf{x}\|^2/2\sigma^2)/(2\pi\sigma^2)^{d/2}$. When $\lambda$ smaller then $\frac{1}{2}$, we have $e^{\lambda\|\mathbf{z}\|^2}\phi_1(\mathbf{z}) = \phi_{\frac{1}{1-2\lambda}}(\mathbf{z})(\frac{1}{1-2\lambda})^{\frac{d}{2}}$. Therefore, we know that:

$$
\begin{aligned}
K_1 &= \left(\frac{1}{1-2\lambda}\right)^{\frac{d}{2}} \int_{\mathbb{R}^d} \phi_{\frac{1}{1-2\lambda}}(\mathbf{z}) 1_{\|\mathbf{z}\| \geq \frac{M-R}{\sqrt{1-\alpha^2}}} d\mathbf{z} \\
&= \left(\frac{1}{1-2\lambda}\right)^{\frac{d}{2}} \mathbb{P}\left[\|\mathbf{Z}'\| \geq \frac{M-R}{\sqrt{1-\alpha^2}} \Big| \mathbf{Z}' \sim N\left(0, \frac{1}{1-2\lambda}\mathbf{I}_d\right)\right] \\
&= \left(\frac{1}{1-2\lambda}\right)^{\frac{d}{2}} \mathbb{P}\left[\|\mathbf{Z}\| \geq \frac{\sqrt{1-2\lambda}(M-R)}{\sqrt{1-\alpha^2}} \Big| \mathbf{Z} \sim N(0, \mathbf{I}_d)\right] \\
&\overset{(i)}{\leq} \left(\frac{1}{1-2\lambda}\right)^{\frac{d}{2}} \frac{1-\alpha^2}{(1-2\lambda)(M-R)^2} \mathbb{E}\|\mathbf{Z}\|^2 \\
&= \left(\frac{1}{1-2\lambda}\right)^{\frac{d}{2}+1} \frac{\sigma_{T-t}d}{(M-R)^2} \\
&\overset{(i)}{\lesssim} \frac{\sigma_{T-t}d}{R^2}.
\end{aligned}
\tag{I.5}
$$

where $(i)$ holds due to the Markov's inequality. Since $\eta \leq \frac{1}{2c_1d} \leq \frac{1}{2\sigma_{T-t}c_1d}$, we have $\lambda = 2\eta c_1(1-\alpha^2) \leq 1/(d+2)$. This implies $(1/(1-2\lambda))^{d/2} \leq e$, thus $(ii)$ holds. Moreover, for $K_2$ we have:

$$
\begin{aligned}
K_2 &= \left(\frac{1}{1-2\lambda}\right)^{\frac{d}{2}} \int_{\mathbb{R}^d} \phi_{\frac{1}{1-2\lambda}}(\mathbf{z}) \|\mathbf{z}\|^2 1_{\|\mathbf{z}\| \geq \frac{M-R}{\sqrt{1-\alpha^2}}} d\mathbf{z} \\
&\leq \left(\frac{1}{1-2\lambda}\right)^{\frac{d}{2}} \mathbb{E}_{\mathbf{Z}' \sim N(0, \frac{1}{1-2\lambda}\mathbf{I}_d)} \|\mathbf{Z}'\|^2 \\
&= \left(\frac{1}{1-2\lambda}\right)^{\frac{d}{2}} \frac{d}{1-2\lambda} \\
&\lesssim d.
\end{aligned}
\tag{I.6}
$$

The last inequality holds due to $(1/(1-2\lambda))^{d/2} \leq e$ when $\lambda \leq 1/(d+2)$. Substituting (I.5) and (I.6) into (I.2), we have

$$
\begin{aligned}
I_2 &\lesssim \frac{d}{\sigma_{T-t}} \\
&\lesssim \frac{d}{\min\{T-t, 1\}}.
\end{aligned}
\tag{I.7}
$$

Combining I.1 and I.7, we have:

$$\int_{\Omega_t} \left\|\frac{\nabla q(T-t, \mathbf{x})}{p_{F_t(\mathbf{Y}_{t_k})}(\mathbf{x})}\right\|^2 p_{F_t(\mathbf{Y}_{t_k})}(\mathbf{x}) d\mathbf{x} \lesssim \frac{d}{\min\{T-t, 1\}},$$

which completes the proof of Lemma F.2. $\qquad\square$

## I.2 PROOF OF LEMMAS G.1 AND G.2

We first present three technical lemmas.

**Lemma I.3.** Let $\mathbf{Y}_t$ and $F_t(\mathbf{z})$ be defined in (G.2) and (G.5). For any $k, t \in [t_k, t_{k+1}]$ and $\mathbf{x} \in \mathbb{R}^d$, we have:

$$\frac{p_{\mathbf{Y}_t}(\mathbf{x})}{p_{F_t(\mathbf{Y}_{t_k})}(\mathbf{x})} \leq e \left|\nabla F_t\left(F_t^{-1}(\mathbf{x})\right)\right| \cdot \left(\frac{T-t+\frac{t-t_k}{1+2/d}}{T-t}\right)^{d/2} \cdot e^{\frac{(1+d/2)\|F_t^{-1}(\mathbf{x})-\mathbf{x}\|_2^2}{2(t-t_k)}},$$

**Lemma I.4.** Suppose $\mathbf{Y}_t = \mathbf{X}_0 + N\left(0, (T-t)I_d\right)$ and $p(\|\mathbf{X}_0\|_2 < R) = 1$. Then for $\lambda < \min\{\frac{1}{4d(T-t)}, \frac{1}{R^2}\}$, we have:

$$\mathbb{E}e^{\lambda\|\mathbf{Y}_t\|_2^2} \lesssim 1$$

**Lemma I.5.** Recall that $\boldsymbol{X}_t$ is our forward process defined in G.1 and we use $q(t, \mathbf{x})$ to denote its law, under Assumption 5.1, we have:

$$\mathbb{E}\|\nabla \log q(t, \boldsymbol{X}_t)\|^2 \le \frac{d}{t}.$$

*Proof of lemma G.1.* By Lemma I.3, we know that:

$$\frac{p_{\boldsymbol{Y}_t}(\mathbf{x})}{p_{F_t(\boldsymbol{Y}_{t_k})}(\mathbf{x})} \le \mathrm{e} \left| \nabla F_t\big(F_t^{-1}(\mathbf{x})\big) \right| \cdot \mathrm{e}^{\frac{(1+d/2)\|F_t^{-1}(\mathbf{x}) - \mathbf{x}\|_2^2}{2(t-t_k)}} \cdot \left( \frac{T - t + \frac{t-t_k}{1+2/d}}{T - t} \right)^{d/2},$$

Firstly, using the expression of the interpolation operator (G.5), we have

$$\nabla F_t(\mathbf{x}) = \mathbf{I}_d + (t - t_k) c_l \nabla \mathbf{s}_\theta(T - t_k, \mathbf{x}).$$

The operator norm of $\nabla F_t(\mathbf{x})$ can be bounded by

$$\|\nabla F_t(\mathbf{x})\|_2 \le 1 + (t - t_k) c_l L$$
$$\le 1 + (t - t_k) L,$$

Using the fact that $|\mathbf{A}| \le \|\mathbf{A}\|^d$, we know that:

$$\left| \nabla F_t\big(F_t^{-1}(\mathbf{x})\big) \right| \le \big(1 + (t - t_k) L\big)^d.$$

Since $t - t_k \le \eta < 1$, when $\eta \le 1/(Ld)$, we have

$$\left| \nabla F_t\big(F_t^{-1}(\mathbf{x})\big) \right| \le \big(1 + \eta L\big)^d \le \mathrm{e}.$$

With our time schedule, by (2.6), we have

$$t - t_k \le t_{k+1} - t_k \le \eta \min\{1, T - t\}.$$

We have $t - t_k \le \eta$. Therefore, we have

$$\left( \frac{T - t + \frac{t-t_k}{1+2/d}}{T - t} \right)^{d/2} \le \left( 1 + \frac{t - t_k}{T - t} \right)^{d/2}$$
$$\le (1 + \eta)^{d/2}.$$

When $\eta \le \frac{1}{d}$, we have:

$$(1 + \eta)^{d/2} \le (1 + \frac{1}{d})^{d/2} \le \sqrt{\mathrm{e}}.$$

Then we have:

$$\frac{(1 + d/2)\|F_t^{-1}(\mathbf{x}) - \mathbf{x}\|_2^2}{2(t - t_k)} = \frac{(1 + d/2)(t - t_k)}{2} c_l^2 \|\mathbf{s}_\theta\big(T - t_k, F_t^{-1}(\mathbf{x})\big)\|^2$$

$$\le \frac{(1 + d/2)(t - t_k)}{2} \big(L \|F_t^{-1}(\mathbf{x})\| + c\big)^2$$

$$\le \frac{(1 + d/2)(t - t_k)}{2} \big(2L \|\mathbf{x}\| + L + c\big)^2$$

$$\le (t - t_k) \Big( \underbrace{(1 + d/2)4L^2}_{c_1} \|\mathbf{x}\|^2 + \underbrace{(1 + d/2)(L + c)^2}_{c_2} \Big).$$

Putting together, we know that:

$$\frac{p_{\boldsymbol{Y}_t}(\mathbf{x})}{p_{F_t(\boldsymbol{Y}_{t_k})}(\mathbf{x})} \lesssim \mathrm{e}^{(t-t_k)c_1 \|\mathbf{x}\|^2 + (t-t_k)c_2}.$$

Moreover, since $t - t_k \le \eta \le 1/c_2$, we have:

$$\int_{\Omega_t} \left( \frac{p_{\boldsymbol{Y}_t}(\mathbf{x})}{p_{F_t(\boldsymbol{Y}_{t_k})}(\mathbf{x})} \right)^2 p_{F_t(\boldsymbol{Y}_{t_k})}(\mathbf{x}) \mathrm{d}\mathbf{x} = \int_{\Omega_t} \frac{p_{\boldsymbol{Y}_t}(\mathbf{x})}{p_{F_t(\boldsymbol{Y}_{t_k})}(\mathbf{x})}) p_{\boldsymbol{Y}_t}(\mathbf{x}) \mathrm{d}\mathbf{x}$$

$$\lesssim \mathbb{E} \mathrm{e}^{(t-t_k)c_1 \|\boldsymbol{Y}_t\|^2}$$

$$\overset{(i)}{\lesssim} 1,$$

where $(i)$ holds due to Lemma I.4 (we have $(t - t_k)c_1 \le \min\{\frac{1}{4d(T-t)}, \frac{1}{R^2}\}$). $\qquad \square$

*Proof of Lemma G.2.* To start with, recall that we assume that $L \geq 1$, we can easily verify that $\eta$ satisfies Lemma G.1's condition. Using Lemma G.1, we know that:

$$\frac{p_{\boldsymbol{Y}_t}(\mathbf{x})}{p_{F_t(\boldsymbol{Y}_{t_k})}(\mathbf{x})} \lesssim e^{(t-t_k)c_1\|\mathbf{x}\|^2 + (t-t_k)c_2}$$

$$\leq e^{\eta\left(c_1\|\mathbf{x}\|^2 + c_2\right)},$$

where $c_1 = (1+d/2)4L^2$ and $c_2 = (1+d/2)(L+c)^2$. Thus we have:

$$\int_{\Omega_t} \left\|\frac{\nabla q(T-t,\mathbf{x})}{p_{F_t(\boldsymbol{Y}_{t_k})}(\mathbf{x})}\right\|^2 p_{F_t(\boldsymbol{Y}_{t_k})}(\mathbf{x})\mathrm{d}\mathbf{x} \leq \int_{\Omega_t}\left\|\nabla \log q(T-t,\mathbf{x})\right\|^2 \frac{p_{\boldsymbol{Y}_t}(\mathbf{x})}{p_{F_t(\boldsymbol{Y}_{t_k})}(\mathbf{x})}p_{\boldsymbol{Y}_t}(\mathbf{x})\mathrm{d}\mathbf{x}$$

$$\lesssim \int_{\Omega_t}\left\|\nabla \log q(T-t,\mathbf{x})\right\|^2 e^{\eta(c_1\|\mathbf{x}\|^2 + c_2)}p_{\boldsymbol{Y}_t}(\mathbf{x})\mathrm{d}\mathbf{x}$$

$$\lesssim \mathbb{E}_Q\left[\left\|\nabla \log q(T-t,\boldsymbol{Y}_t)\right\|^2 e^{\eta c_1\|\boldsymbol{Y}_t\|^2}\right],$$

where the last inequality holds due to $\eta \leq \frac{1}{c_2}$. Next, we select a constant $M = 2R$. We have the following inequality:

$$\mathbb{E}_Q\left[\left\|\nabla \log q(T-t,\boldsymbol{Y}_t)\right\|^2 e^{\eta c_1\|\boldsymbol{Y}_t\|^2}\right]$$

$$= \mathbb{E}_Q\left[\left\|\nabla \log q(T-t,\boldsymbol{Y}_t)\right\|^2 e^{\eta c_1\|\boldsymbol{Y}_t\|^2}\mathbb{1}_{\|\boldsymbol{Y}_t\|<M}\right] + \mathbb{E}_Q\left[\left\|\nabla \log q(T-t,\boldsymbol{Y}_t)\right\|^2 e^{\eta c_1\|\boldsymbol{Y}_t\|^2}\mathbb{1}_{\|\boldsymbol{Y}_t\|\geq M}\right]$$

$$\lesssim \underbrace{\mathbb{E}_Q\left\|\nabla \log q(T-t,\boldsymbol{Y}_t)\right\|^2}_{I_1} + \underbrace{\mathbb{E}_Q\left[\left\|\nabla \log q(T-t,\boldsymbol{Y}_t)\right\|^2 e^{\eta c_1\|\boldsymbol{Y}_t\|^2}\mathbb{1}_{\|\boldsymbol{Y}_t\|\geq M}\right]}_{I_2},$$

where the last inequality holds due to $\eta \leq \frac{1}{c_1 R^2}$. For $I_1$, using Lemma I.5, we have:

$$I_1 = \mathbb{E}_Q\|\nabla \log q(T-t,\boldsymbol{Y}_t)\|^2$$

$$\leq \frac{d}{T-t}, \tag{I.8}$$

For $I_2$, using Lemma H.1 with $f(t) = 1$, $g(t) = \sqrt{T-t}$, we have:

$$\|\nabla \log q(T-t,\mathbf{x})\| \leq \frac{\|\mathbf{x}\| + R}{T-t},$$

Therefore, we have:

$$I_2 \leq \mathbb{E}_Q\left[\left(\frac{\|\boldsymbol{Y}_t\| + R}{T-t}\right)^2 e^{\eta c_1\|\boldsymbol{Y}_t\|^2}\mathbb{1}_{\|\boldsymbol{Y}_t\|\geq M}\right]. \tag{I.9}$$

We have $\boldsymbol{Y}_t = \boldsymbol{X}_{T-t} = \boldsymbol{X}_0 + \sqrt{T-t}\boldsymbol{Z}$. Thus, $\|\boldsymbol{Y}_t\| \leq R + \sqrt{T-t}\|\boldsymbol{Z}\|$. Hence, (I.9) becomes

$$I_2 \leq \frac{1}{(T-t)^2}\mathbb{E}_Q\left[\left(2R + \sqrt{T-t}\|\boldsymbol{Z}\|\right)^2 e^{\eta c_1(R+\sqrt{T-t}\|\boldsymbol{Z}\|)^2}\mathbb{1}_{\|\boldsymbol{Z}\|\geq\frac{M-R}{\sqrt{T-t}}}\right]$$

$$\lesssim \frac{1}{(T-t)^2}\mathbb{E}_Q\left[\left(R^2 + (T-t)\|\boldsymbol{Z}\|^2\right)e^{2\eta c_1(T-t)\|\boldsymbol{Z}\|^2}\mathbb{1}_{\|\boldsymbol{Z}\|\geq\frac{M-R}{\sqrt{T-t}}}\right], \tag{I.10}$$

where the last inequality holds due to $(a+b)^2 \leq 2a^2 + 2b^2$, and $\eta \leq \frac{1}{c_1 R^2}$. Let $\lambda = 2\eta c_1(T-t)$. Then we have

$$I_2 \lesssim \frac{R^2}{(T-t)^2}\underbrace{\int_{\mathbb{R}^d}\frac{1}{(\sqrt{2\pi})^d}e^{-\frac{\|\mathbf{z}\|^2}{2}}e^{\lambda\|\mathbf{z}\|^2}\mathbb{1}_{\|\mathbf{z}\|\geq\frac{M-R}{\sqrt{T-t}}}\mathrm{d}\mathbf{z}}_{K_1}$$

$$+ \frac{2}{T-t}\underbrace{\int_{\mathbb{R}^d}\frac{1}{\sqrt{2\pi}^d}e^{-\frac{\|\mathbf{z}\|^2}{2}}\|\mathbf{z}\|^2 e^{\lambda\|\mathbf{z}\|^2}\mathbb{1}_{\|\mathbf{z}\|\geq\frac{M-R}{\sqrt{T-t}}}\mathrm{d}\mathbf{z}}_{K_2}, \tag{I.11}$$

Let $\phi_{\sigma^2}(\mathbf{x}) = \exp(-\|\mathbf{x}\|^2/2\sigma^2)/(2\pi\sigma^2)^{d/2}$. When $\lambda$ smaller then $\frac{1}{2}$, we have $e^{\lambda\|\mathbf{z}\|^2}\phi_1(\mathbf{z}) = \phi_{\frac{1}{1-2\lambda}}(\mathbf{z})(\frac{1}{1-2\lambda})^{\frac{d}{2}}$. Therefore, we know that:

$$
\begin{aligned}
K_1 &= \left(\frac{1}{1-2\lambda}\right)^{\frac{d}{2}} \int_{\mathbb{R}^d} \phi_{\frac{1}{1-2\lambda}}(\mathbf{z}) 1_{\|\mathbf{z}\| \geq \frac{M-R}{\sqrt{T-t}}} \, d\mathbf{z} \\
&= \left(\frac{1}{1-2\lambda}\right)^{\frac{d}{2}} \mathbb{P}\left[\|\mathbf{Z}'\| \geq \frac{M-R}{\sqrt{T-t}} \Big| \mathbf{Z}' \sim N\left(0, \frac{1}{1-2\lambda}\mathbf{I}_d\right)\right] \\
&= \left(\frac{1}{1-2\lambda}\right)^{\frac{d}{2}} \mathbb{P}\left[\|\mathbf{Z}\| \geq \frac{\sqrt{1-2\lambda}(M-R)}{\sqrt{T-t}} \Big| \mathbf{Z} \sim N(0, \mathbf{I}_d)\right] \\
&\overset{(i)}{\leq} \left(\frac{1}{1-2\lambda}\right)^{\frac{d}{2}} \frac{T-t}{(1-2\lambda)(M-R)^2} \mathbb{E}\|\mathbf{Z}\|^2 \\
&\leq \left(\frac{1}{1-2\lambda}\right)^{\frac{d}{2}+1} \frac{(T-t)d}{(M-R)^2} \\
&\overset{(i)}{\lesssim} \frac{(T-t)d}{R^2}.
\end{aligned}
\tag{I.12}
$$

where $(i)$ holds due to the Markov's inequality. $(ii)$ holds because $\eta \leq \frac{1}{2(d+2)c_1(T-t)}$, thus $\lambda \leq 1/(d+2)$, then we know that $(1/(1-2\lambda))^{d/2} \leq e$. Moreover, for $K_2$ we have:

$$
\begin{aligned}
K_2 &= \left(\frac{1}{1-2\lambda}\right)^{\frac{d}{2}} \int_{\mathbb{R}^d} \phi_{\frac{1}{1-2\lambda}}(\mathbf{z})\|\mathbf{z}\|^2 1_{\|\mathbf{z}\| \geq \frac{M-R}{\sqrt{T-t}}} \, d\mathbf{z} \\
&\leq \left(\frac{1}{1-2\lambda}\right)^{\frac{d}{2}} \mathbb{E}_{\mathbf{Z}' \sim N(0, \frac{1}{1-2\lambda}\mathbf{I}_d)}\|\mathbf{Z}'\|^2 \\
&= \left(\frac{1}{1-2\lambda}\right)^{\frac{d}{2}} \frac{d}{1-2\lambda} \\
&\lesssim d.
\end{aligned}
\tag{I.13}
$$

The last inequality holds due to $(1/(1-2\lambda))^{d/2} \leq e$. Substituting (I.12) and (I.13) into (I.11), we have

$$
I_2 \lesssim \frac{d}{T-t}.
\tag{I.14}
$$

Combining I.8 and I.14, we have:

$$
\int_{\Omega_t} \left\|\frac{\nabla q(T-t, \mathbf{x})}{p_{F_t(\mathbf{Y}_{t_k})}(\mathbf{x})}\right\|^2 p_{F_t(\mathbf{Y}_{t_k})}(\mathbf{x}) \, d\mathbf{x} \lesssim \frac{d}{T-t},
$$

which completes the proof of Lemma G.2. □

## J    PROOF OF LEMMAS IN SECTION I

*Proof of Lemma I.1.* Using the Jacobian transformation of probability densities, we have

$$
p_{F_t(\mathbf{Y}_{t_k})}(\mathbf{x}) = p_{\mathbf{Y}_{t_k}}\left(F_t^{-1}(\mathbf{x})\right)\left|\nabla\left(F_t^{-1}(\mathbf{x})\right)\right|.
\tag{J.1}
$$

Moreover, the forward process indicates $\mathbf{Y}_{t_k} = e^{-(t-t_k)}\mathbf{Y}_t + \mathbf{Z}_{t,t_k}$, where $\mathbf{Z}_{t,t_k} \sim N(0, 1 - e^{-2(t-t_k)})$ is independent of $\mathbf{Y}_t$. Using the Jacobian transformation of probability densities, we have

$$
\begin{aligned}
p_{\mathbf{Y}_{t_k}}\left(F_t^{-1}(\mathbf{x})\right) &= p_{\mathbf{Y}_t + e^{t-t_k}\mathbf{Z}_{t,t_k}}\left(e^{t-t_k}F_t^{-1}(\mathbf{x})\right) \cdot e^{t-t_k} \\
&= e^{t-t_k} \int_{\mathbb{R}^d} q(t, \mathbf{y})\phi_{e^{2(t-t_k)}-1}\left(e^{t-t_k}F_t^{-1}(\mathbf{x}) - \mathbf{y}\right) d\mathbf{y},
\end{aligned}
$$

where $\phi_{\sigma^2}(\mathbf{x}) = \exp(-\|\mathbf{x}\|^2/2\sigma^2)/(2\pi\sigma^2)^{d/2}$ is the probability density function of Gaussian distribution with variance $\sigma^2$. The last equality holds due to the formula for the sum of independent

variables. For simplicity, denote $a = e^{t-t_k}$ and we ignore the dependency of $t$ and $t_k$ when it will not cause any confusion. Then we have

$$p_{\boldsymbol{Y}_{t_k}}\left(F_t^{-1}(\mathbf{x})\right) = a \int_{\mathbb{R}^d} q(T - t, \mathbf{y})\phi_{a^2-1}\left(aF_t^{-1}(\mathbf{x}) - \mathbf{y}\right)\mathrm{d}\mathbf{y}. \tag{J.2}$$

Since $aF_t^{-1}(\mathbf{x}) - \mathbf{y} = \mathbf{x} - \mathbf{y} + (aF_t^{-1}(\mathbf{x}) - \mathbf{x})$, with the shorthand notation $g(\mathbf{x}) = aF_t^{-1}(\mathbf{x}) - \mathbf{x}$, we have

$$\begin{aligned}
\phi_{a^2-1}\left(aF_t^{-1}(\mathbf{x}) - \mathbf{y}\right) &= \phi_{a^2-1}\left(\mathbf{x} - \mathbf{y} + \mathbf{g}(\mathbf{x})\right) \\
&= \frac{1}{(2\pi(a^2-1))^{d/2}}e^{-\frac{\|\mathbf{x}-\mathbf{y}+\mathbf{g}(\mathbf{x})\|_2^2}{2(a^2-1)}} \\
&= \frac{1}{(2\pi(a^2-1))^{d/2}}e^{-\frac{\|\mathbf{x}-\mathbf{y}\|^2+\|\mathbf{g}(\mathbf{x})\|_2^2+2\langle\mathbf{x}-\mathbf{y},\mathbf{g}(\mathbf{x})\rangle}{2(a^2-1)}} \\
&\geq \frac{1}{(2\pi(a^2-1))^{d/2}}e^{-\frac{(1+2/d)\|\mathbf{x}-\mathbf{y}\|_2^2+(1+d/2)\|\mathbf{g}(\mathbf{x})\|_2^2}{2(a^2-1)}} \\
&= \frac{1}{(1+2/d)^{d/2}} \cdot \phi_{\frac{a^2-1}{1+2/d}}(\mathbf{x}-\mathbf{y}) \cdot e^{-\frac{(1+d/2)\|\mathbf{g}(\mathbf{x})\|_2^2}{2(a^2-1)}}, \tag{J.3}
\end{aligned}$$

where the first inequality holds due to the Young's inequality. In the last equality, we use the definition of $\phi_{\sigma^2}(\cdot)$ with $\sigma^2 = (a^2-1)/(1+2/d)$. Combining (J.1), (J.2) and (J.3), we have

$$\begin{aligned}
p_{F_t(\boldsymbol{Y}_{t_k})}(\mathbf{x}) &\geq \left|\nabla\left(F_t^{-1}(\mathbf{x})\right)\right| \cdot a \cdot \int_{\mathbb{R}^d} q(t, \mathbf{y})\frac{1}{(1+2/d)^{d/2}}\phi_{\frac{a^2-1}{1+2/d}}(\mathbf{x}-\mathbf{y})e^{-\frac{(1+d/2)\|\mathbf{g}(\mathbf{x})\|_2^2}{2(a^2-1)}}\mathrm{d}\mathbf{y} \\
&\geq \frac{a}{e\left|\nabla F_t\left(F_t^{-1}(\mathbf{x})\right)\right|}p_{\boldsymbol{Y}_t+N(0,\frac{a^2-1}{1+2/d})}(\mathbf{x}) \cdot e^{-\frac{(1+d/2)\|\mathbf{g}(\mathbf{x})\|_2^2}{2(a^2-1)}},
\end{aligned}$$

where we use $(1+2/d)^{d/2} \leq e$, the fact that $\nabla\left(F_t^{-1}(\mathbf{x})\right) = \left[\nabla F_t\left(F_t^{-1}(\mathbf{x})\right)\right]^{-1}$ and the formula for the distribution of the sum of independent random variables. Then we have

$$\frac{p_{\boldsymbol{Y}_t}(\mathbf{x})}{p_{F_t(\boldsymbol{Y}_{t_k})}(\mathbf{x})} \leq \frac{e\left|\nabla F_t\left(F_t^{-1}(\mathbf{x})\right)\right|}{a} \cdot e^{\frac{(1+d/2)\|\mathbf{g}(\mathbf{x})\|_2^2}{2(a^2-1)}} \cdot \frac{p_{\boldsymbol{Y}_t}(\mathbf{x})}{p_{\boldsymbol{Y}_t+N(0,\frac{a^2-1}{1+2/d})}(\mathbf{x})}$$

Since $\boldsymbol{Y}_t = e^{-(T-t)}X_{\text{data}} + N(0, 1 - e^{-2(T-t)})$, by Lemma K.3, we know that

$$\frac{p_{\boldsymbol{Y}_t}(\mathbf{x})}{p_{\boldsymbol{Y}_t+N(0,\frac{t-t_k}{1+2/d})}(\mathbf{x})} \leq \left(\frac{1 - e^{-2(T-t)} + \frac{a^2-1}{1+2/d}}{1 - e^{-2(T-t)}}\right)^{d/2}$$

Combining the above two inequalities and we can complete the proof of Lemma I.1 $\qquad\square$

*Proof of lemma I.2.* Since $\boldsymbol{Y}_t = e^{-(T-t)}\boldsymbol{X}_{\text{data}} + N\left(0, (1 - e^{-2(T-t)})I_d\right)$, denote $e^{-(T-t)}$ by $c$ and use $\boldsymbol{Z}$ to represent standard normal distribution, we know that:

$$\begin{aligned}
\mathbb{E}e^{\lambda\|\boldsymbol{Y}_t\|_2^2} &= \mathbb{E}e^{\lambda\|c\boldsymbol{X}_{\text{data}}+\sqrt{1-c^2}\boldsymbol{Z}\|_2^2} \\
&\overset{(i)}{\leq} \mathbb{E}e^{\lambda\|\boldsymbol{X}_{\text{data}}\|_2^2+\lambda\|\boldsymbol{Z}\|_2^2} \\
&\overset{(ii)}{=} \mathbb{E}e^{\lambda\|\boldsymbol{X}_{\text{data}}\|_2^2} \cdot \mathbb{E}e^{\lambda\|\boldsymbol{Z}\|_2^2}.
\end{aligned}$$

where $(i)$ holds due to the Cauchy-Schwartz inequality. $(ii)$ holds due to the independency of $\boldsymbol{X}_{\text{data}}$ and $\boldsymbol{Z}$.

Since $p(\|\boldsymbol{X}_{\text{data}}\|_2 < R) = 1$, then $\mathbb{E}e^{\lambda\|\boldsymbol{X}_{\text{data}}\|_2^2} \leq e^{\lambda R^2}$. For $\lambda < \frac{1}{2}$, we know that:

$$\begin{aligned}
\mathbb{E}e^{\lambda\|\boldsymbol{Z}\|_2^2} &= \int_{\mathbb{R}^d} e^{\lambda\|\mathbf{x}\|_2^2}(2\pi)^{-\frac{d}{2}}e^{-\frac{\|\mathbf{x}\|_2^2}{2}}\mathrm{d}\mathbf{x} \\
&= \int_{\mathbb{R}^d} (2\pi\frac{1}{1-2\lambda})^{-\frac{d}{2}}(1-2\lambda)^{\frac{d}{2}}e^{-\frac{\|\mathbf{x}\|_2^2}{2\frac{1}{1-2\lambda}}}\mathrm{d}\mathbf{x}
\end{aligned}$$

$$= (1 - 2\lambda)^{\frac{d}{2}}.$$

When $\lambda < \frac{1}{d}$, $(1 - 2\lambda)^{-\frac{d}{2}} < \frac{1}{e}$, thus we have:

$$\mathbb{E}e^{\lambda \|\boldsymbol{Y}_t\|_2^2} \le e^{\lambda R^2 - 1}.$$

$\square$

*Proof of Lemma I.3.* Using the Jacobian transformation of probability densities, we have

$$p_{F_t(\boldsymbol{Y}_{t_k})}(\mathbf{x}) = p_{\boldsymbol{Y}_{t_k}}\big(F_t^{-1}(\mathbf{x})\big)\big|\nabla\big(F_t^{-1}(\mathbf{x})\big)\big|. \tag{J.4}$$

Moreover, the forward process indicates $\boldsymbol{Y}_{t_k} = \boldsymbol{Y}_t + \boldsymbol{Z}_{t,t_k}$, where $\boldsymbol{Z}_{t,t_k} \sim N(0, t - t_k)$ is independent of $\boldsymbol{Y}_t$. Using the Jacobian transformation of probability densities, we have

$$p_{\boldsymbol{Y}_{t_k}}\big(F_t^{-1}(\mathbf{x})\big) = p_{\boldsymbol{Y}_t + \boldsymbol{Z}_{t,t_k}}\big(F_t^{-1}(\mathbf{x})\big)$$

$$= \int_{\mathbb{R}^d} q(T - t, \mathbf{y})\phi_{t-t_k}\big(F_t^{-1}(\mathbf{x}) - \mathbf{y}\big)\mathrm{d}\mathbf{y},$$

where $\phi_{\sigma^2}(\mathbf{x}) = \exp(-\|\mathbf{x}\|^2/2\sigma^2)/(2\pi\sigma^2)^{d/2}$ is the probability density function of Gaussian distribution with variance $\sigma^2$. The last equality holds due to the formula for the sum of independent variables. We have

$$\phi_{t-t_k}\big(F_t^{-1}(\mathbf{x}) - \mathbf{y}\big) = \phi_{t-t_k}\big(\mathbf{x} - \mathbf{y} + F_t^{-1}(\mathbf{x}) - \mathbf{x}\big)$$

$$= \frac{1}{(2\pi(t - t_k))^{d/2}}e^{-\frac{\|\mathbf{x}-\mathbf{y}+F_t^{-1}(\mathbf{x})-\mathbf{x}\|_2^2}{2(t-t_k)}}$$

$$= \frac{1}{(2\pi(t - t_k))^{d/2}}e^{-\frac{\|\mathbf{x}-\mathbf{y}\|^2+\|F_t^{-1}(\mathbf{x})-\mathbf{x}\|_2^2+2\langle\mathbf{x}-\mathbf{y},F_t^{-1}(\mathbf{x})-\mathbf{x}\rangle}{2(t-t_k)}}$$

$$\ge \frac{1}{(2\pi(t - t_k))^{d/2}}e^{-\frac{(1+2/d)\|\mathbf{x}-\mathbf{y}\|_2^2+(1+d/2)\|F_t^{-1}(\mathbf{x})-\mathbf{x}\|_2^2}{2(t-t_k)}}$$

$$= \frac{1}{(1 + 2/d)^{d/2}} \cdot \phi_{\frac{t-t_k}{1+2/d}}(\mathbf{x} - \mathbf{y}) \cdot e^{-\frac{(1+d/2)\|F_t^{-1}(\mathbf{x})-\mathbf{x}\|_2^2}{2(t-t_k)}}, \tag{J.5}$$

where the first inequality holds due to the Young's inequality. In the last equality, we use the definition of $\phi_{\sigma^2}(\cdot)$ with $\sigma^2 = (t - t_k)/(1 + 2/d)$. Combining (J.4) and (J.5), we have

$$p_{F_t(\boldsymbol{Y}_{t_k})}(\mathbf{x}) \ge \big|\nabla\big(F_t^{-1}(\mathbf{x})\big)\big| \cdot \int_{\mathbb{R}^d} q(T - t, \mathbf{y})\frac{1}{(1 + 2/d)^{d/2}}\phi_{\frac{t-t_k}{1+2/d}}(\mathbf{x} - \mathbf{y})e^{-\frac{(1+d/2)\|F_t^{-1}(\mathbf{x})-\mathbf{x}\|_2^2}{2(t-t_k)}}\mathrm{d}\mathbf{y}$$

$$\ge \frac{1}{e\big|\nabla F_t\big(F_t^{-1}(\mathbf{x})\big)\big|}p_{\boldsymbol{Y}_t+N(0,\frac{t-t_k}{1+2/d})}(\mathbf{x}) \cdot e^{-\frac{(1+d/2)\|F_t^{-1}(\mathbf{x})-\mathbf{x}\|_2^2}{2(t-t_k)}},$$

where we use $(1 + 2/d)^{d/2} \le e$, the fact that $\nabla\big(F_t^{-1}(\mathbf{x})\big) = \big[\nabla F_t\big(F_t^{-1}(\mathbf{x})\big)\big]^{-1}$ and the formula for the distribution of the sum of independent random variables. Then we have

$$\frac{p_{\boldsymbol{Y}_t}(\mathbf{x})}{p_{F_t(\boldsymbol{Y}_{t_k})}(\mathbf{x})} \le e\big|\nabla F_t\big(F_t^{-1}(\mathbf{x})\big)\big| \cdot e^{\frac{(1+d/2)\|F_t^{-1}(\mathbf{x})-\mathbf{x}\|_2^2}{2(t-t_k)}} \cdot \frac{p_{\boldsymbol{Y}_t}(\mathbf{x})}{p_{\boldsymbol{Y}_t+N(0,\frac{t-t_k}{1+2/d})}(\mathbf{x})}$$

Since $\boldsymbol{Y}_t = \boldsymbol{X}_0 + N(0, T - t)$, by Lemma K.3, we know that

$$\frac{p_{\boldsymbol{Y}_t}(\mathbf{x})}{p_{\boldsymbol{Y}_t+N(0,\frac{t-t_k}{1+2/d})}(\mathbf{x})} \le \Big(\frac{T - t + \frac{t-t_k}{1+2/d}}{T - t}\Big)^{d/2}$$

Combining the above two inequalities and we can complete the proof of Lemma I.3 $\square$

*Proof of lemma I.4.* Since $\boldsymbol{Y}_t = \boldsymbol{X}_0 + \sqrt{T - t}\boldsymbol{Z}$, we know that:

$$\mathbb{E}e^{\lambda\|\boldsymbol{Y}_t\|_2^2} \le \mathbb{E}e^{2\lambda\|\boldsymbol{X}_0\|^2} \cdot \mathbb{E}e^{2\lambda(T-t)\|\boldsymbol{Z}\|^2}$$

$$\le e^{2\lambda R^2} \cdot e^{2\lambda(T-t)\|\boldsymbol{Z}\|^2}$$

$$\overset{(i)}{\lesssim} e^{2\lambda(T-t)\|\boldsymbol{Z}\|^2},$$

here $(i)$ is because we assumed $\lambda < 1/R^2$. Moreover, we have $\lambda \le \frac{1}{4d(T-t)}$, this implies $2\lambda(T - t) \le \frac{1}{2}$, thus we have

$$
\begin{aligned}
\mathbb{E}e^{2\lambda(T-t)\|\boldsymbol{Z}\|_2^2} &= \int_{\mathbb{R}^d} e^{2\lambda(T-t)\|\mathbf{x}\|_2^2}(2\pi)^{-\frac{d}{2}}e^{-\frac{\|\mathbf{x}\|_2^2}{2}}\,\mathrm{d}\mathbf{x} \\
&= \int_{\mathbb{R}^d} (2\pi\frac{1}{1-4\lambda(T-t)})^{-\frac{d}{2}}(1-4\lambda(T-t))^{-\frac{d}{2}}e^{-\frac{\|\mathbf{x}\|_2^2}{2\frac{1}{1-4\lambda(T-t)}}}\,\mathrm{d}\mathbf{x} \\
&= (1-4\lambda(T-t))^{-\frac{d}{2}} \\
&\lesssim 1.
\end{aligned}
$$

This completes the proof of Lemma I.4 $\qquad\qquad\square$

*Proof of Lemma I.5.* By tweedie's formula we know that:

$$\nabla \log q(t, \boldsymbol{X}_t) = -\frac{1}{t}\Big(\boldsymbol{X}_t - \mathbb{E}_{q_{0|t}(\cdot|\boldsymbol{X}_t)}\boldsymbol{X}_0\Big).$$

$$
\begin{aligned}
\mathbb{E}\|\nabla \log q_t(\boldsymbol{X}_t)\|^2 &= \frac{1}{t^2}\mathbb{E}\|\boldsymbol{X}_t - \mathbb{E}[\boldsymbol{X}_0|\boldsymbol{X}_t]\|^2 \\
&= \frac{1}{t^2}\left[\mathbb{E}\|\boldsymbol{X}_t\|^2 - 2\mathbb{E}\boldsymbol{X}_t \cdot [\boldsymbol{X}_0|\boldsymbol{X}_t] + \mathbb{E}\|\mathbb{E}[\boldsymbol{X}_0|\boldsymbol{X}_t]\|^2\right].
\end{aligned}
$$

Since $\boldsymbol{X}_t = \boldsymbol{X}_0 + \sqrt{t}\boldsymbol{Z}$, we have $\mathbb{E}\|\boldsymbol{X}_t\|^2 = \mathbb{E}\|\boldsymbol{X}_0\|^2 + td$. Here the second order momentum is finite because our bounded support assumption. Moreover, we know that:

$$\mathbb{E}\boldsymbol{X}_t \cdot \mathbb{E}[\boldsymbol{X}_0|\boldsymbol{X}_t] = \mathbb{E}\boldsymbol{X}_t \cdot \boldsymbol{X}_0 = \mathbb{E}\left(\boldsymbol{X}_0 + \sqrt{t}\boldsymbol{Z}\right) \cdot \boldsymbol{X}_0 = \mathbb{E}\|\boldsymbol{X}_0\|^2.$$

We next consider the trace of the covariance matrix of $\boldsymbol{X}_0$ given $\boldsymbol{X}_t$:

$$\mathrm{tr}\left(\mathrm{Cov}_{q_{0|t}(\cdot|\boldsymbol{X}_t)}(\boldsymbol{X}_0)\right) = \mathbb{E}\left[\|\boldsymbol{X}_0\|^2|\boldsymbol{X}_t\right] - \|\mathbb{E}[\boldsymbol{X}_0|\boldsymbol{X}_t]\|^2,$$

Thus we know that:

$$
\begin{aligned}
\mathbb{E}\left[\|\mathbb{E}[\boldsymbol{X}_0|\boldsymbol{X}_t]\|^2\right] &= \mathbb{E}\|\boldsymbol{X}_0\|^2 - \mathbb{E}\,\mathrm{tr}\left(\mathrm{Cov}_{q_{0|t}(\cdot|\boldsymbol{X}_t)}(\boldsymbol{X}_0)\right) \\
&\le \mathbb{E}\|\boldsymbol{X}_0\|^2.
\end{aligned}
$$

Putting together, we know that:

$$\mathbb{E}\|\nabla \log q(t, \boldsymbol{X}_t)\|^2 \le \frac{d}{t}.$$

$\qquad\qquad\square$

# K  AUXILIARY LEMMAS

**Theorem K.1** (Reynolds Transport Theorem Leal 2007)**.** For a function $F(t, \mathbf{x}) : \mathbb{R} \times \mathbb{R}^d \to \mathbb{R}$ that is continuously differentiable with respect to both $\mathbf{x}$ and $t$, the following equality holds:

$$\frac{\partial}{\partial t}\int_{\Omega_t} F(t, \mathbf{x})\,\mathrm{d}\mathbf{x} = \int_{\Omega_t}\frac{\partial}{\partial t}F(t, \mathbf{x})\,\mathrm{d}\mathbf{x} + \int_{\partial\Omega_t} F(t, \mathbf{x})\,\mathbf{v}(t, \mathbf{x}) \cdot \mathbf{n}(t, \mathbf{x})\,\mathrm{d}S,$$

where $\mathbf{n}$ is the outward-pointing unit normal vector, $\mathbf{v}$ is the velocity of the area element, and $\mathrm{d}S$ is area element.

**Lemma K.2** (Lemma 6 in Benton et al. 2024)**.** Let $\boldsymbol{X}_t$ be the OU forward process defined in (F.1). When $\boldsymbol{X}_0$ has finite second moments, we have:

$$\mathbb{E}\|\nabla \log q(t, \boldsymbol{X}_t)\|^2 \le \frac{d}{1 - e^{-2t}}.$$

*Proof of Lemma K.2.* This lemma is the same as Lemma 6 in Benton et al. (2024). We provide a proof for the completeness of our paper.

Denote $1 - \mathrm{e}^{-2t}$ by $\sigma_t$. By Tweedie's formula we know that:

$$\nabla \log q(t, \boldsymbol{X}_t) = \frac{1}{\sigma_t}\Big( -\boldsymbol{X}_t + \mathrm{e}^{-t}\mathbb{E}_{q_{0|t}(\cdot|\boldsymbol{X}_t)}\boldsymbol{X}_0\Big).$$

Taking the expectation of the square, we have

$$
\begin{aligned}
\mathbb{E}\|\nabla \log q_t(\boldsymbol{X}_t)\|^2 &= \sigma_t^{-2}\mathbb{E}\|\mathrm{e}^{-t}\mathbb{E}[\boldsymbol{X}_0|\boldsymbol{X}_t] - \boldsymbol{X}_t\|^2 \\
&= \sigma_t^{-2}\left[\mathbb{E}\|\boldsymbol{X}_t\|^2 - 2\mathrm{e}^{-t}\mathbb{E}\boldsymbol{X}_t \cdot [\boldsymbol{X}_0|\boldsymbol{X}_t] + \mathrm{e}^{-2t}\mathbb{E}\|\mathbb{E}[\boldsymbol{X}_0|\boldsymbol{X}_t]\|^2\right].
\end{aligned}
$$

Since $\boldsymbol{X}_t = \mathrm{e}^{-t}\boldsymbol{X}_0 + \sqrt{\sigma_t}\boldsymbol{Z}$, we have $\mathbb{E}\|\boldsymbol{X}_t\|^2 = \mathrm{e}^{-2t}\mathbb{E}\|\boldsymbol{X}_0\|^2 + \sigma_t d$. Recall that we assume that $\boldsymbol{X}_0$ has finite second moments. Moreover, we know that:

$$\mathbb{E}\boldsymbol{X}_t \cdot \mathbb{E}[\boldsymbol{X}_0|\boldsymbol{X}_t] = \mathbb{E}\boldsymbol{X}_t \cdot \boldsymbol{X}_0 = \mathbb{E}\left(\mathrm{e}^{-t}\boldsymbol{X}_0 + \sqrt{\sigma_t}\boldsymbol{Z}\right) \cdot \boldsymbol{X}_0 = \mathrm{e}^{-t}\mathbb{E}\|\boldsymbol{X}_0\|^2.$$

We next consider the trace of the covariance matrix of $\boldsymbol{X}_0$ given $\boldsymbol{X}_t$:

$$\mathrm{tr}\Big(\mathrm{Cov}_{q_{0|t}(\cdot|\boldsymbol{X}_t)}(\boldsymbol{X}_0)\Big) = \mathbb{E}\left[\|\boldsymbol{X}_0\|^2|\boldsymbol{X}_t\right] - \|\mathbb{E}[\boldsymbol{X}_0|\boldsymbol{X}_t]\|^2,$$

Thus we know that:

$$
\begin{aligned}
\mathbb{E}\left[\|\mathbb{E}[\boldsymbol{X}_0|\boldsymbol{X}_t]\|^2\right] &= \mathbb{E}\|\boldsymbol{X}_0\|^2 - \mathbb{E}\,\mathrm{tr}\Big(\mathrm{Cov}_{q_{0|t}(\cdot|\boldsymbol{X}_t)}(\boldsymbol{X}_0)\Big) \\
&\le \mathbb{E}\|\boldsymbol{X}_0\|^2.
\end{aligned}
$$

Putting together, we know that:

$$\mathbb{E}\|\nabla \log q(t, \boldsymbol{X}_t)\|^2 \le \frac{d}{\sigma_t}.$$

$\square$

**Lemma K.3.** For any data distribution $p_{\mathrm{data}}$ and positive parameters $\delta$ and $h$. Let $\boldsymbol{X}, \boldsymbol{Z}_\delta, \boldsymbol{Z}_h$ be three independent random variables in $\mathbb{R}^d$ satisfying $\boldsymbol{X} \sim p_{\mathrm{data}}$, $\boldsymbol{Z}_\delta \sim N(0, \delta)$ and $\boldsymbol{Z}_h \sim N(0, h)$. Then we have:

$$\frac{p_{\boldsymbol{X}+\boldsymbol{Z}_\delta}(\mathbf{x})}{p_{\boldsymbol{X}+\boldsymbol{Z}_\delta+\boldsymbol{Z}_h}(\mathbf{x})} \le \Big(\frac{\delta+h}{\delta}\Big)^{d/2}.$$

where we use $p_{\boldsymbol{Y}}(\boldsymbol{x})$ to denote the probability density function of random variable $\boldsymbol{Y}$.

*Proof of Lemma K.3.* We define $\phi_{\sigma^2}(\mathbf{x}) := \frac{1}{(2\pi\sigma^2)^{d/2}}\mathrm{e}^{-\frac{\|\mathbf{x}\|^2}{2\sigma^2}}$, which is the probability density function of normal distribution $N(0, \sigma^2\boldsymbol{I}_d)$.

First, we provide an upper bound of $\phi_\delta(\mathbf{x} - \mathbf{y})/\phi_{\delta+h}(\mathbf{x} - \mathbf{y})$:

$$\frac{\phi_\delta(\mathbf{x} - \mathbf{y})}{\phi_{\delta+h}(\mathbf{x} - \mathbf{y})} = \Big(\frac{\delta+h}{\delta}\Big)^{d/2}\mathrm{e}^{\frac{(\mathbf{x}-\mathbf{y})^2}{2(\delta+h)} - \frac{(\mathbf{x}-\mathbf{y})^2}{2\delta}} \le \Big(\frac{\delta+h}{\delta}\Big)^{d/2}. \tag{K.1}$$

Using the independence property, we have

$$p_{\mathbf{X}+\mathbf{z}_\delta}(\mathbf{x}) = \int_{\mathbb{R}^d} p_{\mathrm{data}}(\mathbf{y})\,\phi_\delta(\mathbf{x} - \mathbf{y})\,\mathrm{d}\mathbf{y},$$

$$p_{\mathbf{X}+\mathbf{z}_\delta+\mathbf{z}_h}(\mathbf{x}) = \int_{\mathbb{R}^d} p_{\mathrm{data}}(\mathbf{y})\,\phi_{\delta+h}(\mathbf{x} - \mathbf{y})\,\mathrm{d}\mathbf{y}.$$

Then we have

$$
\begin{aligned}
\int_{\mathbb{R}^d} p_{\mathrm{data}}(\mathbf{y})\phi_\delta(\mathbf{x} - \mathbf{y})\mathrm{d}\mathbf{y} &= \left(\int_{\mathbb{R}^d} p_{\mathrm{data}}(\mathbf{y})\phi_{\delta+h}(\mathbf{x} - \mathbf{y})\frac{\phi_\delta(\mathbf{x} - \mathbf{y})}{\phi_{\delta+h}(\mathbf{x} - \mathbf{y})}\mathrm{d}\mathbf{y}\right) \\
&\le \int_{\mathbb{R}^d} p_{\mathrm{data}}(\mathbf{y})\phi_{\delta+h}(\mathbf{x} - \mathbf{y})\mathrm{d}\mathbf{y}\Big(\frac{\delta+h}{\delta}\Big)^{d/2},
\end{aligned}
$$

which completes the proof of Lemma K.3.

$\square$

