# OpenReview forum: "Unified Convergence Analysis for Score-Based Diffusion Models with Deterministic Samplers"
_ICLR.cc/2025/Conference — ICLR 2025 Poster_

### Official Review · Reviewer_EFQA · 2024-11-02

**Soundness:** 3
**Presentation:** 2
**Contribution:** 3
**Rating:** 6
**Confidence:** 3

**Summary:**

The paper studies the error of generative models based on deterministic samplers, that is when the backward process is represented by an ODE. Error bounds are presented both for the continuous time and for the discretisation schemes, giving a general result that holds for a wide class of schemes. Several instances of samplers are then considered, giving error bounds for each case.

**Strengths:**

The results in the paper are interesting and cover a good range of algorithms that are of practical interest in generative modelling. The results complement well with the existing literature.

The main idea, Lemma 4.2, is to bound the derivative of the total variation distance of the laws of the true and approximate backward processes, giving an error that depends on the score and its divergence. As far as I know this is original in this literature.

The presentation is overall clear enough and the goal and results are simple to understand.

**Weaknesses:**

The paper presents several times inconsistent notation, e.g. lines 134-137, 151-154, 349-350, repeated equations (e.g. lines 204 and 206).

Some aspects should be explained better in the paper, for instance: why is Assumption 6.1 used in the application the specific examples of samplers? Or also, can the authors give a clearer idea of what F can be in lines 345-350 for specific samplers? As it is now, the reader has to trust that this function can be found easily, while it would be much better to have one or two examples.

Several times I had the impression that sentences could be written better, e.g. the statement of 6.5 just to name one, or several times the authors write "Lipschiz" instead of Lipschitz.

**Questions:**

In lines 756-761, why is the TV distance equal to the integral over the entire real line? Can the authors clarify how this is obtained, and also why the absolute value is inside the integral? It cannot be that one of the two densities is larger than the other over the entire domain, else it would not integrate to 1. For this reason I am doubtful of the proof.

---

> ### Author Response · Authors · 2024-11-20
> **Response to Reviewer EFQA**
>
> # Response to Reviewer EFQA
>
> Thanks for your positive feedback. We will address your concerns one by one. Due to the page limit, we moved the related work section to the appendix, which resulted in changes to the numbering of some sections. For ease of review, all new/added content in our revised version is highlighted in blue.
>
> **Q1:**
>
> The paper presents several times inconsistent notation, e.g. lines 134-137, 151-154, 349-350, repeated equations (e.g. lines 204 and 206).
>
> **A1:**
>
> Thanks for pointing these out. We have fixed these notation problems in our revision.
>
> ----
>
> **Q2:**
>
> Why is Assumption 6.1 used in the application the specific examples of samplers?
>
> **A2:**
>
> We make this commonly-used bounded support assumption (see [1], [2], and [3]) to ensure the smoothness of the score function, which is essential when analyzing the discretization error. Under this assumption, we can avoid more restrictive conditions, such as requiring the smoothness of the data distribution. This allows us to consider a broader range of scenarios, including cases where the data distribution lies on a lower-dimensional submanifold and the score function at $t=0$ is undefined. In [3], which analyzes the convergence behavior of stochastic samplers, the authors initially derive their main results under the bounded support condition and then demonstrate that this assumption can be relaxed to a more general tail bound condition. Building on this, we conjecture that the bounded support assumption in our work could also be further relaxed to cover more settings, such as light-tailed distributions, as discussed in Lines 538-539.
>
> ----
>
> **Q3:**
>
> Or also, can the authors give a clearer idea of what F can be in lines 345-350 for specific samplers? As it is now, the reader has to trust that this function can be found easily, while it would be much better to have one or two examples.
>
> **A3:**
>
>  In Section 4.1, we provide methods to compute values from time $t_k$ to $t_{k+1}$ for specific numerical schemes.  As explained in Lines 328-332, by replacing $t_{k+1}$ with $t$ in equations (4.2), (4.3), and (4.4), we can obtain explicit forms of $F_{t_k\rightarrow t}(\cdot)$, or simplfied as $F_t$. Additionally, we provide explicit form of $F$ in Sections 5.1 and 5.2 for specific numerical schemes, Lines 443-444 and Lines 482-483. We have added a guidance sentence to further clarify its definition.
>
> ----
>
> **Q4:**
>
>  Several times I had the impression that sentences could be written better, e.g. the statement of 6.5 just to name one, or several times the authors write "Lipschiz" instead of Lipschitz.
>
> **A4:**
>
> Thank you for your advice. We have rephrased Statements 5.5 and 5.9, and corrected our typos.
>
> ----
>
> **Q5:**
>
>  In lines 756-761, why is the TV distance equal to the integral over the entire real line? Can the authors clarify how this is obtained, and also why the absolute value is inside the integral? It cannot be that one of the two densities is larger than the other over the entire domain, else it would not integrate to 1. For this reason I am doubtful of the proof.
>
> **A5:**
>
> This is another common definition of TV distance. To obtain this equation, let's first consider the original definition of TV distance. The TV distance between two probability measures $P$ and $Q$ on a measurable space $(Ω,\mathcal{F})$ , it is defined as:
> $$\text{TV}(P,Q) = \sup_{A \in \mathcal{F}} |P(A) - Q(A)|$$
> If P and Q have density functions p(x) and q(x) , the TV distance can be expressed in terms of the densities. Specifically, consider the set $A=\{x \in \Omega :p(x) \geq q(x)\}$. For this choice of $A$, $∣P(A)−Q(A)∣$ achieves the supremum, thus we have:
> $$\text{TV}(P,Q) = P(A) - Q(A) = \int_{A} [p(x) - q(x)] \mathrm{d}x.$$
> Since both $P$ and $Q$ are probability measures that integrate to 1, we have:
> $$\int_{A^c} q(x) - p(x) \mathrm{d}x = \int_{A} p(x) - q(x) \mathrm{d}x$$
> Therefore:
> $$\text{TV}(P,Q) = \frac{1}{2}\left(\int_{A} [p(x) - q(x)] \mathrm{d}x + \int_{A^c} [q(x) - p(x)] \mathrm{d}x\right) = \frac{1}{2}\int |p(x) - q(x)| \mathrm{d}x ,$$
> because $p(x) \ge q(x)$ on $A$ and $p(x) \le q(x)$ on $A^c$. This explains why the TV distance can be calculated by integrating over the entire real line.
>
> ---
>
> **Reference**
>
> [1] Valentin De Bortoli, Convergence of denoising diffusion models under the manifold hypothesis, TMLR 2022.
>
> [2] Chen et al, Sampling is as easy as learning the score: theory for diffusion models with minimal data assumptions, ICLR 2023
>
> [3] Lee et al, Convergence of score-based generative modeling for general data distributions, NuerIPS 2022

---

> ### Author Response · Authors · 2024-11-26
> **We are looking forward to your feedback!**
>
> Dear Reviewer EFQA,
>
> Thank you once again for your valuable feedback. In our rebuttal, we have fixed the notation issues and improved our presentation. We have clarified the reasons for using the bounded support assumption and provided a detailed derivation for obtaining the integral of the absolute value definition of the TV distance. We believe these responses address your questions, and we look forward to your feedback.
>
> Best,
> Authors

---

### Official Review · Reviewer_Pg4X · 2024-11-02

**Soundness:** 3
**Presentation:** 2
**Contribution:** 3
**Rating:** 6
**Confidence:** 4

**Summary:**

This paper consider the convergence of score-based diffusion with deterministic sampler. The paper starts with a counter-example (using 1d OU process) to show that the problem of convergence may be subtle. The paper then shows a $O(d^2/\varepsilon)$ bound for the convergence, yet another polynomial complexity.

**Strengths:**

The paper provides a systematic/unified study of the convergence of the score-based diffusion models using deterministic or ODE samplers (mostly, EI). The paper is generally well written, and I enjoyed reading it.

**Weaknesses:**

There are several weaknesses of this paper:

1. There have been now a handful of papers, e.g., Chen et al, Li et al., and Huang et al,... providing the polynomial convergence of the diffusion models via ODE/deterministic sampling. It is not too clear how the results in this paper are compared to the existing literature. For presentation, the authors may provide a table to showcase.

2. The authors claimed "polynomial iteration complexity for the VE forward process with DDIM numerical scheme, both of which have been relatively underexplored in the literature. It is known that VE and VP are connected by a reparametrization. In most of cases, it suffices to derive the convergence for VP or VE. This viewpoint is highlighted in the important paper of https://arxiv.org/abs/2206.00364, which the authors fail to cite.

3. The main theorem, Theorem 5.3, basically decomposes the error into three pieces: (1) initialization error, (2) score matching error and (3) discretization error. This idea is not too novel (and of course, the analysis may be involved). The theoretical insight from this paper is not significant, and the contribution of the paper is purely technical.

4. In Theorem 5.3, the authors include a term involving the second order derivatives, which echoes Assumption 6.3. I find this assumption is somewhat too strong.

5. Out of curiosity, I wonder if the authors expect the rate derived in the paper is optimal (especially regarding the dimension $d$).

**Questions:**

See weakness.

---

> ### Author Response · Authors · 2024-11-20
> **Response to Reviewer Pg4X**
>
> # Response to Reviewer Pg4X
>
> Thanks for your valuable feedback. We will address your concerns one by one in the following response. Due to the page limit, we moved the related work section to the appendix, which resulted in changes to the numbering of some sections. For ease of review, all new/added content in our revised version is highlighted in blue.
>
> **Q1:**
>
> There have been now a handful of papers, e.g., Chen et al, Li et al., and Huang et al,... providing the polynomial convergence of the diffusion models via ODE/deterministic sampling. It is not too clear how the results in this paper are compared to the existing literature. For presentation, the authors may provide a table to showcase.
>
> **A1:**
>
> We have discussed the connections with these three works in the Related Work section (see Appendix A), as well as in Remarks 2.2 and 3.5, and Assumptions 5.1 and 5.4. Regarding Huang's work, which is most closely related to ours, we have provided a detailed comparison of our methodology and results in Section A.1. To make the comparison with related works clearer, we have added a comparative table in Appendix A.2.
>
> ----
>
> **Q2:**
>
> The authors claimed "polynomial iteration complexity for the VE forward process with DDIM numerical scheme, both of which have been relatively underexplored in the literature. It is known that VE and VP are connected by a reparametrization. In most of cases, it suffices to derive the convergence for VP or VE. This viewpoint is highlighted in the important paper of https://arxiv.org/abs/2206.00364, which the authors fail to cite.
>
> **A2:**
>
> We acknowledge that [1] introduces a general ODE framework for diffusion models and demonstrates that the probability ODE flow of VP and VE can be connected by scaling. However, this equivalence does not account for initialization error and discretization error, which are critical factors in theoretical analysis and significantly influence the final convergence results.
>
> Due to the variance-exploding property of VE, the distance between the final distribution of the forward process and a Gaussian decreases at a $O(1/\sqrt{T})$ rate, which is much slower compared with the $O(\exp(-T))$ rate for VP. Additionally, the scaling equivalence does not consider the time schedule. In practice, the selection of different time schedules will lead to different discretization errors.  Consequently, although their probability flow ODEs can be connected, the theoretical analysis of VP and VE cannot be simply equated, underscoring the value of our work. Moreover, within the scaling framework of [1], the drift coefficients must be linear in $x$. In contrast, our framework offers greater generality by working with nonlinear $f$.
>
> In our revision, we have cited and discussed the referenced paper in Section 5, helping to clarify the distinctions between our theoretical framework and the one proposed in [1].
>
> ----
>
> **Q3:**
>
> The main theorem, Theorem 5.3, basically decomposes the error into three pieces: (1) initialization error, (2) score matching error and (3) discretization error. This idea is not too novel (and of course, the analysis may be involved). The theoretical insight from this paper is not significant, and the contribution of the paper is purely technical.
>
> **A3:**
>
> We believe the reviewer might have overlooked the novelty and significance of our results. Let us clarify our key contributions:
> As discussed in Section 2.1, initialization error, score matching error and discretization error are three fundamental difficulties in the analysis of diffusion models, arising from algorithm design. Addressing these errors is a common requirement in all theoretical works on diffusion models. In our paper, we first illustrate the challenges in the convergence analysis of ODE flows through a counterexample Theorem 2.1. Drawing insights from this counterexample, we develop a key technical tool Lemma 3.2. Unlike previous works that involve the score of the sampling process, Lemma 3.2 uses integration by parts to avoid the appearance of the sampling scores (as detailed in Lines 230-240). Based on Lemma 3.2, we present a unified framework that applies to general diffusion forward processes and samplers, formulating each error term in its most general form. We consider this unified framework to be our main contribution. Through this unified framework, we can systematically analyze the convergence of various existing diffusion algorithms, and potentially develop new algorithms with theoretical guarantees.
>
> ----
>
> **Reference:**
>
> [1] Karras et al. Elucidating the Design Space of Diffusion-Based Generative Models,  Neurips2022
>
> [2] Li et al, Towards Faster Non-Asymptotic Convergence for Diffusion-Based Generative Models, ICLR2024
>
> [3] Li et al, A Sharp Convergence Theory for The Probability Flow ODEs of Diffusion Models, ArXiv

---

> ### Author Response · Authors · 2024-11-20
> **Response to Reviewer Pg4X**
>
> **Q4:**
>
> In Theorem 5.3, the authors include a term involving the second order derivatives, which echoes Assumption 6.3. I find this assumption is somewhat too strong.
>
> **A4:**
>
> Due to the counterexamples we construct in Theorem 2.1, if we want to directly analyze the convergence of ODE flow in terms of TV or KL divergence, we must impose additional conditions. In this paper, we consider the divergence error as one possible choice - we chose it due to the hints from the key lemma. A similar but different assumption has been made in [2] (Assumption 2). They require controlling over the norm of the difference between Jacobian matrices, while we require the divergence error (the trace of the Jacobian). As adding this assumption can guarantee the convergence of ODE sampler, an interesting future direction would be to design training methods that can obtain both small score error and small divergence error.  One potential approach would be to incorporate regularization terms corresponding to divergence error in the loss function, potentially leading to more effective diffusion model training algorithms. We have added a brief discussion of this in our future work section.
>
> **Q5:**
>
> out of curiosity, I wonder if the authors expect the rate derived in the paper is optimal (especially regarding the dimension d).
>
> **A5:**
>
> The discrepancy between us and current sota result [3] stems from two factors: our different assumption that requires control of the divergence error rather than the full Jacobian error,} and our directly estimated Lipschitz constants in the discretization error analysis. We believe that through more delicate methods for analyzing the discretization error and potentially stronger conditions, we could achieve linear dimension dependence within our unified framework. We have added a brief discussion of this direction in our future work section.
>
> **Reference:**
>
> [1] Karras et al. Elucidating the Design Space of Diffusion-Based Generative Models,  Neurips2022
>
> [2] Li et al, Towards Faster Non-Asymptotic Convergence for Diffusion-Based Generative Models, ICLR2024
>
> [3] Li et al, A Sharp Convergence Theory for The Probability Flow ODEs of Diffusion Models, ArXiv

---

> ### Author Response · Authors · 2024-11-25
> **We are looking forward to your feedback!**
>
> Dear  Reviewer Pg4X,
>
> Thank you once again for your valuable feedback. In our rebuttal, we believe that we have carefully addressed your questions.  We have added a new table in Appendix A to clarify how our work relates to relevant studies. Regarding our primary contribution, we have emphasized the significance of the unified framework we propose. We have also provided detailed discussion on the necessity of our assumption 6.3.
>
> As the discussion period nears its conclusion, we would like to ask if you have any remaining concerns. We look forward to your feedback.
>
> Best,
>
> Authors

---

> > ### Comment · Reviewer_Pg4X · 2024-11-26
> >
> > I would thank the authors for detailed explanations, and will raise to 6.

---

> ### Author Response · Authors · 2024-11-26
>
> Thank you for raising your rating. We're glad that our rebuttal has addressed your questions. Your thoughtful comments and feedback have been invaluable in helping us improve the paper.

---

### Official Review · Reviewer_usTt · 2024-11-03

**Soundness:** 3
**Presentation:** 3
**Contribution:** 3
**Rating:** 6
**Confidence:** 4

**Summary:**

This paper investigates score-based diffusion models that utilize deterministic samplers. It introduces a unified framework for convergence analysis by leveraging divergence estimation of the score function. Additionally, the analysis explores the convergence properties of variance-preserving SDEs employing the exponential integrator scheme, along with variance-exploding SDEs in conjunction with the DDIM sampler.

**Strengths:**

The paper is clearly written and presents a novel unified convergence analysis of ODE flows that integrates divergence estimation of the score function. Furthermore, this framework provides fresh insights into the convergence analysis of variance-preserving forward processes employing the exponential integrator scheme, as well as variance-exploding forward processes utilizing the DDIM sampler.

**Weaknesses:**

There are a few typos and missing definitions. Additionally, regarding the assumption of divergence—particularly the assumption of the divergence estimation error—are there any studies that provide estimators for divergence that support the validity of this assumption? How can practitioners verify assumptions like those in Section 6.3 in real-world applications?

Further details can be found in the Questions section.

**Questions:**

1. **Typos and Missing Definitions**
   a. Line 176: What is the definition of $q_{\delta}$, and how does the distance between $q_{\delta}$ and $q_0$ depend on $\delta$?
   b. Assumption 4.1: What is the definition of $Q$?
   c. Theorem 4.4: What is the definition of $X_{\delta}$, and what is the total variation distance between $X_{\delta}$ and $q_0$?
   d. Equation (5.2): The definition of the step size $\eta_k$ is missing.
   e. Line 334: Should $\eta$ be $\eta_k$?
   f. Line 349:  Should $F$ be $F_t$?

2. **Corollary 6.7**: The paper states, "assuming sufficiently small score estimation error and divergence error." Could you quantify how small $\epsilon_{\text{score}}$ and $\epsilon_{\text{divergence}}$ should be in relation to the target error level $\epsilon$?

3. **Effectiveness of the Framework**: In Section 3, the author discusses results from Section 6.1, suggesting that the dimension dependence is worse than that of the state of the art. Does this arise from limitations within the proposed framework, weaker assumptions, or other factors? Furthermore, the results in Section 6.2 indicate a slow decay for the variance-exploding SDE. Is this limitation due to the proof technique, or does it stem from other reasons? Given these considerations, despite the generality of the proposed framework, could the author provide additional evidence to demonstrate its effectiveness? Specifically, can this framework achieve a sharp convergence rate or match the state-of-the-art convergence rates?

---

> ### Author Response · Authors · 2024-11-20
> **Response to Reviewer usTt**
>
> # Response to Reviewer usTt
>
> Thanks for your positive feedback. We will address your concerns one by one. Due to the page limit, we moved the related work section to the appendix, which resulted in changes to the numbering of some sections. For ease of review, all new/added content in our revised version is highlighted in blue.
>
> **Q1:**
>
> There are a few typos and missing definitions.
>
> a. Line 176: What is the definition of $q_\delta$ and how does the distance between $q_\delta$ and $q_0$ depend on $\delta$?
>
> b. Assumption 4.1: What is the definition of $Q$?
>
> c. Theorem 4.4: What is the definition of $X_\delta$ and what is the total variation distance between $X_\delta$ and $q_0$?
>
> d. Equation (5.2): The definition of the step size $\eta_k$ is missing.
>
> e. Line 334: Should $\eta$ be $\eta_k$?
>
> f. Line 349: Should $F$ be $F_t$?
>
> **A1:**
>
> Thanks for pointing this out. We will address them one by one.
>
> a & c. The definitions of $q_t$ and $X_t$ can be found in Lines 108-109. Here, $X_{\delta}$ is the random variable obtained by running the forward process for time $\delta$ from $X_0 \sim q_0$, and $q_{\delta}$ represents its law. Without additional data smoothness assumptions, we cannot provide estimates of the TV distance between $X_{\delta}$ and $X_0$ (or between $q_{\delta}$ and $q_0$). However, the Wasserstein-p metric is small when $\delta$ is small enough. This shows that it is reasonable to approximate $q_\delta$ instead of $q_0$. We have added this discussion in Lines 150-151.
>
> b&f. We have fixed our notations and typos.
>
> d. The definition of step size $\eta_k$ appears in Line 156. For clarity, we have also included this definition in Section 5 of our revision.
>
> e. The inequality $l \geq 1/\eta$ is correct, where $\eta$ is defined in Line 156. This follows from our time schedule choice in Equation (3.6): $l \geq (T-t_{k+1})/(t_{k+1}-t_k) \geq \text{min} \\{  1, T-t_{k+1} \\}/(t_{k+1}-t_k)\geq \frac{1}{\eta}$. We have added the derivation in Line 318 of this in our revision for clarity.
>
> ----
>
> **Q2:**
>
> Additionally, regarding the assumption of divergence—particularly the assumption of the divergence estimation error—are there any studies that provide estimators for divergence that support the validity of this assumption? How can practitioners verify assumptions like those in Section 6.3 in real-world applications?
>
> **A2:**
>
> Due to the counterexamples we construct in Theorem 2.1, if we want to directly analyze the convergence of ODE flow in terms of TV or KL divergence, we must impose additional conditions. In this paper, we consider the divergence error as one possible choice - a decision motivated by insights from our key lemma. A similar but different assumption has been made in [1] (Assumption 2). They require controlling over the norm of the difference between Jacobian matrices, while we require the divergence error (the trace of the Jacobian).
>
> Since adding this assumption can guarantee the convergence of ODE sampler, an interesting future direction would be to design training methods that can obtain both small score error and small divergence error. One potential approach would be to incorporate regularization terms corresponding to divergence error in the loss function, potentially leading to more effective diffusion model training algorithms. To the best of our knowledge, there has been no prior work specifically focused on controlling divergence error during the training process. We will leave it as future work and have added a brief discussion of this in Section 6.
>
> ----
>
> **Q3:**
>
> Corollary 6.7: The paper states, "assuming sufficiently small score estimation error and divergence error." Could you quantify how small $\epsilon_{\text{score}}$ and $\epsilon_{\text{divergence}}$ should be in relation to the target error level $\epsilon$?
>
> **A3:**
>
> Based on Theorem 5.5, the coefficient for $\epsilon_{\text{divergence}}$ is 1, while $\epsilon_{\text{score}}$ has a coefficient of $\sqrt{d}\sqrt{\eta N}$. In Corollary 5.7, we set $\eta N = T + \log(1/\delta)$ and $T = \log(d/\epsilon^2)/2$. This implies that we require $\tilde{O}(\epsilon/\sqrt{d})$ score estimation error and $O(\epsilon)$ divergence error to achieve the target error level $\epsilon$. To address your question, in our revision, we have explicitly incorporated these bounds for score and divergence error into Corollary 5.7.
>
> **Reference:**
>
> [1] Li et al, Towards Faster Non-Asymptotic Convergence for Diffusion-Based Generative Models, ICLR2024

---

> ### Author Response · Authors · 2024-11-20
> **Response to Reviewer usTt**
>
> **Q4:**
>
> Effectiveness of the Framework: the dimension dependence is worse than that of the state of the art. Does this arise from limitations within the proposed framework, weaker assumptions, or other factors? Specifically, can this framework achieve a sharp convergence rate or match the state-of-the-art convergence rates? Furthermore, the results in Section 6.2 indicate a slow decay for the variance-exploding SDE. Is this limitation due to the proof technique, or does it stem from other reasons?
>
> **A4:**
>
> Regarding the VP process with EI schemes, this discrepancy stems from two factors: our different assumption that requires control of the divergence error rather than the full Jacobian error, and our directly estimated Lipschitz constants in the discretization error analysis. We believe that this discrepancy is not due to a limitation of our framework, and we believe that with more refined methods for analyzing the discretization error and possibly stronger conditions, we could achieve linear dependence on the dimensionality within our unified framework.
>
> Regarding the VE process, the polynomial bounds stem from the slow $1/\sqrt{T}$ decay in the distance between $q_T$ and $\pi_d$. Importantly, this limitation reflects the theoretical challenges in this setting rather than a shortcoming of the DDIM sampler itself. Indeed, applying DDIM to VP processes yields results comparable to the EI scheme.
>
> **Reference:**
>
> [1] Li et al, Towards Faster Non-Asymptotic Convergence for Diffusion-Based Generative Models, ICLR2024

---

> ### Author Response · Authors · 2024-11-26
> **We are looking forward to your feedback!**
>
> Dear Reviewer usTt,
>
> Thank you once again for your valuable feedback. In our rebuttal, we have clarified our notation and quantified how small should $\epsilon_{\text{score}}$ and $\epsilon_{\text{div}}$ be in relation to the target error level $\epsilon$. We have also provided a detailed discussion regarding the divergence error assumption, the effectiveness of our framework (we believe that this discrepancy is not due to a limitation of our framework), and the reasons for the slow decay in the VE setting. We believe these responses address your questions, and we look forward to your feedback.
>
> Best,
> Authors

---

### Official Review · Reviewer_HJsC · 2024-11-04

**Soundness:** 4
**Presentation:** 4
**Contribution:** 3
**Rating:** 8
**Confidence:** 3

**Summary:**

The paper addresses a previously unexplored question: bounding the total variation between the true distribution and the distribution generated by an ODE sampler of a diffusion model. The contributions are twofold: 1) Identifying that standard assumptions are insufficient for constructing tight bounds; 2) Laying out an intuitive framework that builds the bound under sufficiently strong assumptions, accounting for both approximation errors in learning and the ODE solver.

**Strengths:**

The paper tackles a fundamental question regarding ODE solvers for probability flow ODE, starting with a strong motivation by cleverly constructing a counterexample that demonstrates the insufficiency of assumptions on score estimation error to prove a tight bound on total variation. Adding an extra divergence approximation error, the proposed analytical direction is clear and intuitive, resulting in a three-step procedure:

1. Bound the total variation between two ODEs.
2. Define an extended ODE that aligns with the discretization scheme used in an ODE solver.
3. Bound the difference between the extended ODE and the true ODE.

Additionally, the paper showcases the instantiation of this framework through VP-SDE with an exponential integrator and VE-SDE with a DDIM integrator, providing a very natural path to follow in defining the extended ODE for other diffusion models or samplers.

**Weaknesses:**

While the paper is well-written, it would benefit from a discussion of potential applications for this bound. The main extra assumption required is a bound on divergence, which results in an additive term in the resulting bound. However, divergence is not typically controlled during training, so exploring whether this can be managed would be interesting.

Minor Remark:

I am uncertain about the claim that Lemma 4.2 serves as an ODE version of the Girsanov Theorem, as the Girsanov Theorem concerns the Radon-Nikodym derivative. The use of a time derivative of the divergence of the marginal distributions of an ODE is also common (as seen in the proof of Lemma 2.21 in [1]). Although such a bound may not be practical within the context of this paper, it might be worthwhile to cite or discuss these bounds.

**Questions:**

The dependency on the time schedule appears to manifest solely through Equation (3.6) in various Lemma in Appendix F and G. While it may be due to my limited understanding of prior work, are there other schedules that could fulfil the condition? Can a similarly concise bound be proven with a different time schedule?

The construction of the counterexample is pivotal in motivating the paper. Would it be beneficial to discuss how one might construct such a counterexample within the main text?

[1] Albergo, M. S., Boffi, N. M., & Vanden-Eijnden, E. (2023, March 15). Stochastic Interpolants: A Unifying Framework for Flows and Diffusions. arXiv.org. https://arxiv.org/abs/2303.08797

---

> ### Author Response · Authors · 2024-11-20
> **Response to Reviewer HJsC**
>
> # Response to Reviewer HJsC
>
> Thanks for your positive feedback. We will address your concerns one by one. Due to the page limit, we moved the related work section to the appendix, which resulted in changes to the numbering of some sections. For ease of review, all new/added content in our revised version is highlighted in blue.
>
> **Q1:**
>
> The main extra assumption required is a bound on divergence, which results in an additive term in the resulting bound. However, divergence is not typically controlled during training, so exploring whether this can be managed would be interesting.
>
> **A1:**
>
> Due to the counterexample we have constructed in Theorem 2.1, we must impose additional conditions in order to obtain the convergence guarantee of ODE flows in terms of TV distance.  In our paper, we consider the divergence error due to the hints from Lemma 3.2. A similar but different assumption has been made in [2] (Assumption 2). They require controlling over the norm of the difference between Jacobian matrices, while we require the divergence error (the trace of the Jacobian).
> As this additional assumption can guarantee the convergence of ODE sampler, an interesting future direction would be to design training methods that can obtain both small score error and small divergence error. One potential approach would be to incorporate regularization terms corresponding to divergence error in the loss function of the training process. This could lead to more effective training algorithms for diffusion models with deterministic samplers. We have added a brief discussion of this in our future work section.
>
> ----
>
> **Q2:**
>
> I am uncertain about the claim that Lemma 4.2 serves as an ODE version of the Girsanov Theorem, as the Girsanov Theorem concerns the Radon-Nikodym derivative. The use of a time derivative of the divergence of the marginal distributions of an ODE is also common (as seen in the proof of Lemma 2.21 in [1]).
>
> **A2:**
>
> We apologize for this confusion. Our use of the term 'analogy' was intended to highlight that this lemma plays a similar role in our overall proof framework as the Girsanov theorem, as both of them bound the distance between final distributions through differences in drift terms. To address this concern, we have rephrased the relevant statements in the Introduction and Conclusion sections.
> Several works, including [1] and [3], consider the time derivative of KL divergence, deriving expressions which involve scores of the sampling processes $\nabla \log p_t(x)$. However, when considering ODEs, even when the divergence error is sufficiently small, we can construct a counter-example where the score of the sampling process explodes. (See Theorem B.1). In our work, we consider the time derivative of TV distance in Lemma 3.2. The key feature of our lemma is that, using integration by part, we avoid the appearance of the score of the sampling process. In our revision, we have cited [1] and added a more detailed discussion below Lemma 3.2 to discuss the relation with [1] and [3].
>
> ----
>
> **Q3:**
>
> The dependency on the time schedule appears to manifest solely through Equation (3.6) in various Lemma in Appendix F and G. While it may be due to my limited understanding of prior work, are there other schedules that could fulfil the condition? Can a similarly concise bound be proven with a different time schedule?
>
> **A3:**
>
> The unified framework we propose can handle various time schedules, and similar proofs can be readily established for different time schedules. For analyzing the convergence of specific samplers, the choice of time schedule is crucial for obtaining good bounds, especially how to select the time step when $t$ approaches 0. Based on this consideration, following prior works, such as [4][5], we use exponential decay time schedule when applying to specific samplers in Section 5, where smaller time steps are used when $t$ approaches 0. This allows us to provide more refined control over discretization error. For comparison, using a uniform time schedule would result in an iteration complexity that is $1/\delta$ times higher than our current result. In our revision, we have added more discussion in Section 2.1 to clarify the relationship between the time schedule and our unified framework.
>
> ----
>
> **Reference:**
>
> [1] Albergo, M. S., Boffi, N. M., & Vanden-Eijnden, E. (2023, March 15). Stochastic Interpolants: A Unifying Framework for Flows and Diffusions.
>
> [2] Li et al, Towards Faster Non-Asymptotic Convergence for Diffusion-Based Generative Models, ICLR2024
>
> [3] Chen et al, Restoration-Degradation Beyond Linear Diffusions: A Non-Asymptotic Analysis For DDIM-Type Samplers, ICML2023
>
> [4] Chen et al, Improved Analysis of Score-based Generative Modeling: User-Friendly Bounds under Minimal Smoothness Assumptions, ICML2023
>
> [5] Benton et al. Nearly d-Linear Convergence Bounds for Diffusion Models via Stochastic Localization, ICLR2024

---

> ### Author Response · Authors · 2024-11-20
> **Response to Reviewer HJsC**
>
> **Q4:**
>
> The construction of the counterexample is pivotal in motivating the paper. Would it be beneficial to discuss how one might construct such a counterexample within the main text?
>
> **A4:**
>
> Thanks for this valuable feedback. We have added a brief discussion on the key ideas behind the counter-example construction at the end of section 2 (see remark 2.2).  We consider a specific scenario where the data distribution is standard Gaussian, and the OU process maintains the distribution. In this case, adding an arbitrarily small perturbation to the drift term of the reverse ODE can result in a sampling probability density function $\hat{q}(T-t,x) = \frac{1}{\sqrt{2\pi}} \mathrm{e}^{-\frac{x^2}{2}} \big( 1 + \frac{t}{2T} \sin (2n\pi x)\big)$ with severe oscillations (n can be arbitrarily large), instead of a Gaussian distribution $q(t,x) = \frac{1}{\sqrt{2\pi}} \mathrm{e}^{-\frac{x^2}{2}}$ . Applying the definition, the TV distance between these two distributions is always larger than a constant.
>
> **Reference:**
>
> [1] Albergo, M. S., Boffi, N. M., & Vanden-Eijnden, E. (2023, March 15). Stochastic Interpolants: A Unifying Framework for Flows and Diffusions.
>
> [2] Li et al, Towards Faster Non-Asymptotic Convergence for Diffusion-Based Generative Models, ICLR2024
>
> [3] Chen et al, Restoration-Degradation Beyond Linear Diffusions: A Non-Asymptotic Analysis For DDIM-Type Samplers, ICML2023
>
> [4] Chen et al, Improved Analysis of Score-based Generative Modeling: User-Friendly Bounds under Minimal Smoothness Assumptions, ICML2023
>
> [5] Benton et al. Nearly d-Linear Convergence Bounds for Diffusion Models via Stochastic Localization, ICLR2024

---

### Meta-Review · Area_Chair_yRaG · 2024-12-13

**Metareview:**

This paper introduces a unified framework that analyzed several existing first-order ODE-based samplers, leading to an iteration complexity of $O(d^2/\epsilon)$. The unified analysis may benefit future development of convergence rates in a modular manner, and hence received positive evaluation from the reviewers. Nonetheless, the developed bound is not sharp in view of existing results, e.g., "A Sharp Convergence Theory for The Probability Flow ODEs of Diffusion Models" by Li et al that offers a rate of $d/\epsilon$.

**Additional Comments On Reviewer Discussion:**

After rebuttal, all reviewers have converged to an acceptance score.

---

### Decision · Program_Chairs · 2025-01-22

Accept (Poster)